# Nonlinear control of transcription through enhancer–promoter interactions

Jessica Zuin[1,5], Gregory Roth[1,5], Yinxiu Zhan[1], Julie Cramard[1], Josef Redolfi[1], Ewa Piskadlo[1], Pia Mach[1,2], Mariya Kryzhanovska[1], Gergely Tihanyi[1,2], Hubertus Kohler[1], Mathias Eder[3], Christ Leemans[3], Bas van Steensel[3], Peter Meister[4], Sebastien Smallwood[1] & Luca Giorgetti[1✉]

Chromosome structure in mammals is thought to regulate transcription by modulating three-dimensional interactions between enhancers and promoters, notably through CTCF-mediated loops and topologically associating domains (TADs)[1–4]. However, how chromosome interactions are actually translated into transcriptional outputs remains unclear. Here, to address this question, we use an assay to position an enhancer at large numbers of densely spaced chromosomal locations relative to a fixed promoter, and measure promoter output and interactions within a genomic region with minimal regulatory and structural complexity. A quantitative analysis of hundreds of cell lines reveals that the transcriptional effect of an enhancer depends on its contact probabilities with the promoter through a nonlinear relationship. Mathematical modelling suggests that nonlinearity might arise from transient enhancer–promoter interactions being translated into slower promoter bursting dynamics in individual cells, therefore uncoupling the temporal dynamics of interactions from those of transcription. This uncovers a potential mechanism of how distal enhancers act from large genomic distances, and of how topologically associating domain boundaries block distal enhancers. Finally, we show that enhancer strength also determines absolute transcription levels as well as the sensitivity of a promoter to CTCF-mediated transcriptional insulation. Our measurements establish general principles for the context-dependent role of chromosome structure in long-range transcriptional regulation.

Transcriptional control in mammals critically depends on enhancers, which control tissue specificity and developmental timing of many genes[5]. Enhancers are often located hundreds of kilobases away from target promoters and are thought to control gene expression by interacting with the promoters in the three-dimensional space of the nucleus. Chromosome conformation capture (3C) methods[6] revealed that enhancer–promoter interactions predominantly occur within sub-megabase domains known as topologically associating domains (TADs). These mainly arise from nested looping interactions between sites that are bound by the DNA-binding protein CTCF that act as barriers for the loop extrusion activity of cohesin[7].

TAD boundaries and CTCF loops are thought to favour enhancer–promoter communication within specific genomic regions and disfavour it with respect to surrounding sequences[1,3,4,8]. However, this view has recently been challenged by reports that disruption of TAD boundaries[9,10] or depletion of CTCF and cohesin[11,12] do not lead to systematic changes in gene expression, and that some regulatory sequences can act across TAD boundaries[13]. The manipulation of single CTCF sites has also been reported to result in variable effects on gene expression[2,4,10,14–18]. The very notion that physical proximity is required for transcriptional regulation has been questioned by the observed lack of correlation between transcription and proximity in single cells[19,20]. Thus, it is highly debated whether there are indeed general principles that determine how physical interactions enable or prevent enhancer action[21]. Enhancer–promoter genomic distance might also contribute to transcriptional regulation[22,23], but it is unclear whether an enhancer acts uniformly within a TAD[24,25], or whether its effect depends on the genomic distance from a promoter[23,26].

## Enhancer action depends on genomic distance

Addressing these questions requires a quantitative understanding of the relationship between transcription and enhancer–promoter interactions in conditions in which confounding effects by additional regulatory and structural interactions are minimized. Here we provide such a description using an experimental assay in which an enhancer is mobilized from an initial location and reinserted at large numbers of genomic positions with respect to a promoter. This enables the measurement of transcription levels as a function of the enhancer location and, therefore, of enhancer–promoter contact frequencies (Fig. 1a). Specifically, we generated mouse embryonic stem (mES) cells carrying a transgene in which a promoter drives the expression of enhanced green fluorescent protein (eGFP).

[1]Friedrich Miescher Institute for Biomedical Research, Basel, Switzerland. [2]University of Basel, Basel, Switzerland. [3]Division of Gene Regulation and Oncode Institute, Netherlands Cancer Institute, Amsterdam, The Netherlands. [4]University of Bern, Bern, Switzerland. [5]These authors contributed equally: Jessica Zuin, Gregory Roth. ✉e-mail: luca.giorgetti@fmi.ch

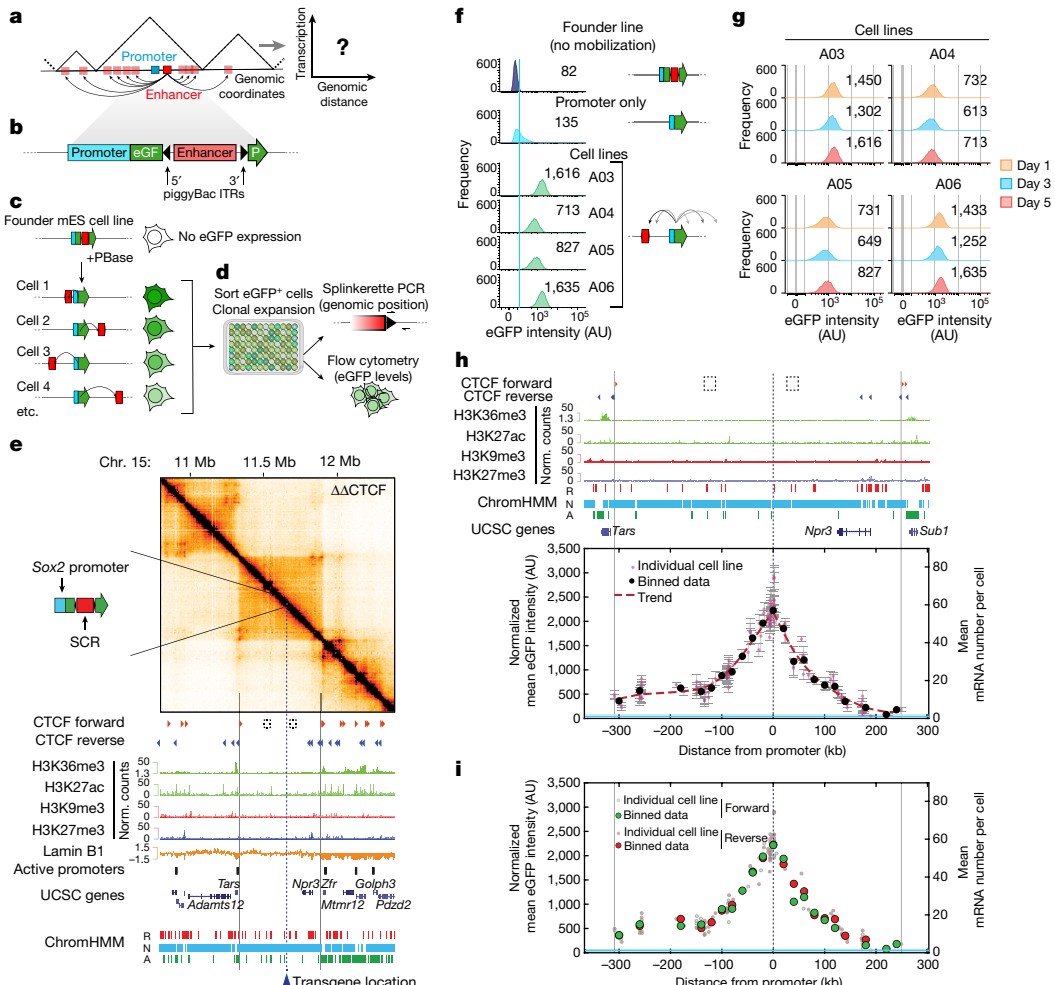

**Fig. 1 | Enhancer action depends on the genomic distance from the promoter and is constrained by TAD boundaries. a**, Mobilization of an enhancer around its target promoter to measure transcription as a function of their genomic distance. **b**, Schematic of the transgene: a promoter drives transcription of an *eGFP* gene split by a piggyBac-enhancer cassette. ITR, inverted terminal repeats. **c**, After expression of PBase, the piggyBac-enhancer cassette is excised and randomly reinserted, occasionally leading to eGFP expression. **d**, Sorting of single eGFP⁺ cells results in cell lines in which the enhancer drives transcription from a single position. Splinkerette PCR and flow cytometry analysis are used to determine the enhancer position and promoter expression levels. **e**, Capture-C (6.4 kb resolution) analysis and genomic datasets in mES cells across 2.6 Mb centred around the selected TAD with both of the internal CTCF motifs deleted (dashed squares; ΔΔCTCF, double CTCF site deletion ). The dashed line indicates the position of the future insertion of the transgene carrying the *Sox2* promoter and SCR. A, active; N, neutral; R,

repressive; Chr, chromosome. **f**, Representative flow cytometry profiles from founder mES cells, a promoter-only control cell line and eGFP⁺ cell lines with mobilized SCR. The light blue line indicates the mean eGFP levels in the promoter-only line. The numbers show the median eGFP intensities. AU, arbitrary units. **g**, eGFP levels in individual eGFP⁺ cell lines over cell passages. The numbers show the median eGFP values. **h**, Normalized mean eGFP intensities in individual eGFP⁺ cell lines as a function of SCR genomic position. The red dots are data from 135 individual cell lines; data are mean ± s.d. *n* = 3 measurements on different days. The black dots show the average values within equally spaced 20 kb bins. The dashed red line shows the spline interpolation of average values. Mean mRNA numbers were inferred using smRNA-FISH calibration (Extended Data Fig. 1h). The light blue area shows the interval between the mean ± s.d. of eGFP levels in three promoter-only cell lines. **i**, Data as in **h**, colour-coded according to SCR genomic orientation.

The *eGFP* transcript is split in two by a piggyBac transposon containing the cognate enhancer of the promoter (Fig. 1b). After expression of the PBase transposase, the transposon is excised and reintegrated randomly into the genome, but preferentially in the vicinity of the initial site[27]. Excision leads to reconstitution of functional eGFP of which the expression is used to isolate clonal cell lines by sorting single eGFP⁺ cells (Fig. 1c, d). This enables the rapid generation of hundreds of cell lines, each with the enhancer in a distinct genomic position. Enhancer position and eGFP expression are then determined in every cell line (Fig. 1d).

To minimize confounding effects, we integrated the transgene within a 560 kb TAD on chromosome 15 carrying minimal regulatory and structural complexity. This TAD does not contain expressed genes or active enhancers, is mostly composed of 'neutral' chromatin[28] except for a repressive ~80 kb region at its 3′ side (Extended Data Fig. 1a), and

displays minimal structure mediated by two internal forward CTCF sites (Extended Data Fig. 1a, b). To further decrease the structural complexity, we deleted the two internal CTCF sites. This led to the loss of the associated loops (Extended Data Fig. 1c) and resulted in a simple homogeneous internal structure, as revealed by capture-C with tiled oligonucleotides spanning 2.9 Mb around the transgene (Fig. 1e and Extended Data Fig. 1c).

We first heterozygously inserted a single copy (Extended Data Fig. 1e) of a version of the transgene carrying the mouse *Sox2* promoter and the essential 4.8 kb region of its distal enhancer known as *Sox2* control region (SCR)[29,30] (Extended Data Fig. 1d and Methods), from which we deleted its single CTCF site, which is not essential for transcriptional regulation at the endogenous locus[17]. Transgene insertion did not lead to substantial structural rearrangements within the TAD besides new moderate interactions with the CTCF sites at the 3′ and 5′ end of the

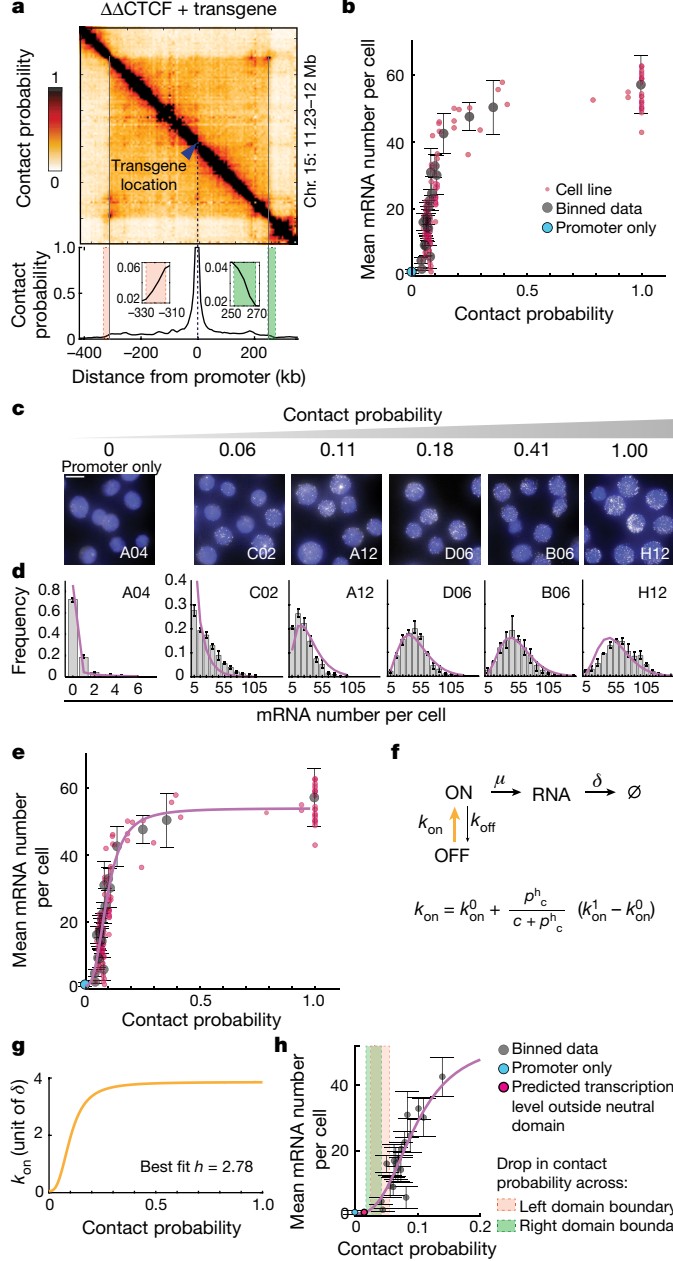

**a** ΔΔCTCF + transgene

Contact probability

Chr. 15: 11.23–12 Mb

Transgene location

Contact probability

0.06 / 0.02 at −330, −310

0.04 / 0.02 at 250, 270

Distance from promoter (kb)

**b**

Mean mRNA number per cell

Contact probability

- Cell line
- Binned data
- Promoter only

**c** Contact probability

0    0.06    0.11    0.18    0.41    1.00

Promoter only

A04    C02    A12    D06    B06    H12

**d**

Frequency

A04    C02    A12    D06    B06    H12

mRNA number per cell

**e**

Mean mRNA number per cell

Contact probability

**f**

$$\text{ON} \xrightarrow{\mu} \text{RNA} \xrightarrow{\delta} \varnothing$$

$k_\text{on} \uparrow\downarrow k_\text{off}$

OFF

$$k_\text{on} = k_\text{on}^0 + \frac{p_\text{c}^h}{c + p_\text{c}^h}(k_\text{on}^1 - k_\text{on}^0)$$

**g**

$k_\text{on}$ (unit of $\delta$)

Best fit $h$ = 2.78

Contact probability

**h**

Mean mRNA number per cell

Contact probability

- Binned data
- Promoter only
- Predicted transcription level outside neutral domain

Drop in contact probability across:
- Left domain boundary
- Right domain boundary

**Fig. 2 | The promoter on rate is a sigmoidal function of enhancer–promoter contact probabilities. a**, Capture-C (6.4 kb resolution) analysis of the founder cell line used for the experiments in Fig. 1 after converting read counts into contact probabilities (top) (Methods). Bottom, cross-section showing contact probabilities from the ectopic *Sox2* transgene. Insets: magnification of contact probability across the TAD boundaries. **b**, Mean *eGFP* mRNA numbers per cell plotted against contact probabilities between the ectopic *Sox2* promoter and SCR insertions. The red dots show individual cell lines. The black dots show the average values within equally spaced 20 kb bins ± s.d. The number of cell lines per bin varies from 1 to 28. **c**, Representative smRNA-FISH images from cell lines in which *eGFP* transcription is driven by the *Sox2* promoter alone (left) or by the SCR located at different distances and contact probabilities (right). Scale bar, 10 μm. **d**, Distributions of mRNA numbers per cell measured in the cell lines shown in **c**. The error bars show the minimum and maximum frequency. *n* = 3 technical replicates. The line shows the best fit of the phenomenological two-state model to the experimental data shown in **b** and **d**. **e**, Best fit to experimental data of **b** and **d**. Best-fit parameters are shown in Extended Data Fig. 3b. **f**, Description of the phenomenological two-state model with a variable on rate. The Hill function describes the dependency of $k_\text{on}$ on contact probability ($p_\text{c}$). $k_\text{on}^0$ and $k_\text{on}^1$ are the minimum and maximum on rates, respectively; $c$ and $h$ are the Hill function critical threshold and the sensitivity parameter, respectively. ∅ symbolizes degraded RNA. **g**, The best-fitting Hill function for $k_\text{on}$ (in units of mRNA lifetime $\delta$), corresponding to a sigmoidal curve. **h**, Close-up of **e**, highlighting the predicted insulation outside the TAD boundaries (red and green shaded areas). Data are presented as in **b**.

tenfold dynamic range in gene expression, from around 5 to 60 mRNAs per cell on average on the basis of smRNA-FISH calibration (Extended Data Fig. 1h). Insertions downstream of the non-transcribed *Npr3* gene generated lower transcription levels (Fig. 1h), possibly because this is a predominantly repressive region. Mild positive and negative deviations from the average decay in transcription levels indeed correlated with local enrichment in active and repressive chromatin states, respectively (Extended Data Fig. 1k). Consistent with the classical notion derived from reporter assays that enhancer activity is independent of genomic orientation[31], enhancers inserted in forward or reverse orientations generated equivalent transcription levels (Fig. 1i). Interestingly, cell-to-cell heterogeneity in eGFP levels (assessed using coefficients of variation (CVs)) showed an opposite trend to mean expression levels and increased with increasing enhancer–promoter genomic distance (Extended Data Fig. 1l; examples of eGFP intensity distributions are provided in Extended Data Fig. 1m). Importantly, these results did not depend on the specific fluorescence gate used to define eGFP⁺ cells (Extended Data Fig. 1n, o). Together, these data show that the range of activity of the enhancer extends to the entire TAD and is delimited by its boundaries. However, transcription levels and their cell-to-cell variability quantitatively depend on enhancer–promoter genomic distance.

## Enhancer contacts modulate burst frequency

We next examined the relationship between transcription levels and contact probabilities. Although reads from the wild-type allele might underemphasize changes introduced by the heterozygous insertion of the transgene, contact patterns detected in capture-C did not change substantially in individual cell lines in which the SCR was mobilized compared to the founder line before piggyBac mobilization (Extended Data Fig. 2a). Thus, the ectopic enhancer and promoter do not create prominent specific interactions, which enabled us to use capture-C data from the founder line (Methods)[32] to infer contact probabilities between promoter and enhancer locations (Fig. 2a). Contact probabilities steeply decayed with increasing genomic distance from the promoter, fell considerably while approaching TAD boundaries (from 1 to around 0.05) and further dropped by a factor of around 3 across boundaries (Fig. 2a). This is consistent with previous estimations[33] confirmed using cross-linking and ligation-free methods[34] and is representative of the contact probabilities

TAD (Extended Data Fig. 1f). Mobilization of the piggyBac-SCR cassette led to random genomic reinsertions with a preference for chromosome 15 itself (Extended Data Fig. 1g). Individual experiments resulted in several tens of cell lines of which the eGFP levels were unimodally distributed (Fig. 1f), generally higher than those detected in control lines in which transcription was driven by the *Sox2* promoter alone (Fig. 1f), and remained stable over cell passages (Fig. 1g). Mean eGFP levels in single cell lines were linearly correlated with average numbers of *eGFP* mRNAs measured using single-molecule RNA fluorescence in situ hybridization (smRNA-FISH) (Extended Data Fig. 1h). We therefore used flow cytometry as a readout of transcriptional activity.

Mapping of piggyBac-SCR positions in more than 300 cell lines revealed that, although in around 15% of them the transposon had not been successfully mobilized, in 99% of those in which it had (262 out of 264), the enhancer reinserted within the initial TAD (Fig. 1h and Extended Data Fig. 1i). In the two cell lines in which the enhancer transposed outside the TAD, eGFP levels were comparable to basal transcription driven by promoter-only control cell lines (Extended Data Fig. 1j). Notably, within the TAD, expression levels decreased with increasing enhancer–promoter genomic distance (Fig. 1h). Genomic distance accounted for a

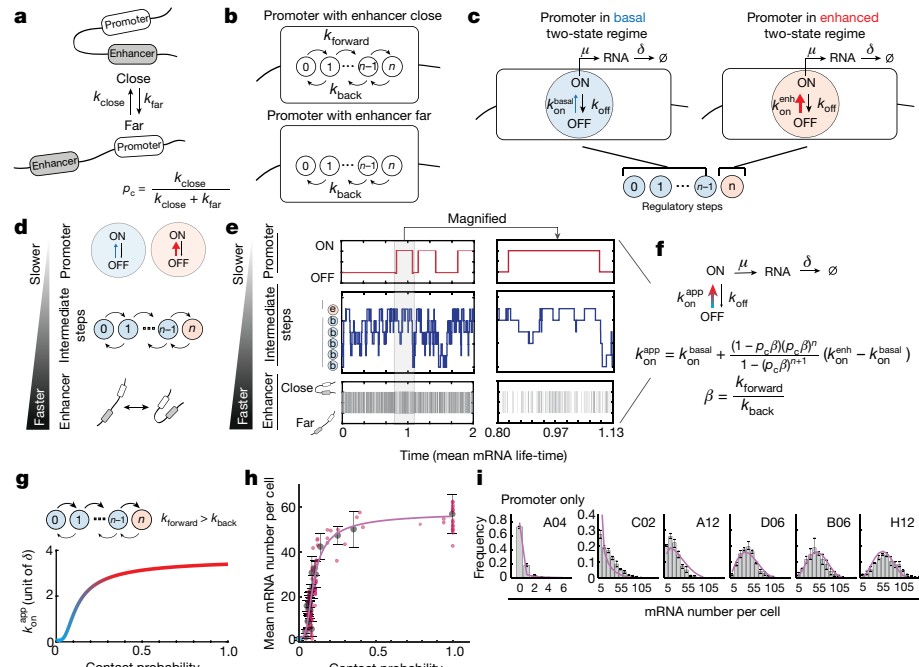

**Fig. 3 | A mechanistic model of enhancer–promoter communication.**
**a**, Stochastic promoter–enhancer interactions occur and disassemble with rates $k_{close}$ and $k_{far}$. **b**, In the close state, the enhancer can trigger $n$ reversible regulatory steps with forward and reverse rates $k_{forward}$ and $k_{back}$. In the far state, regulatory steps can revert only at rate $k_{back}$. **c**, The promoter operates in a basal two-state regime with a small on rate ($k_{on}^{basal}$) unless all $n$ regulatory steps have been completed, in which case it transiently enters an enhanced two-state regime with a higher on rate ($k_{on}^{enh}$). **d**, Schematic of the parameter constraints under which the mechanistic model reduces to an apparent two-state model: $k_{close,far} \gg k_{forward,back} \gg k_{on}^{basal,enh}, k_{off}, \mu$. **e**, Representative single-cell dynamics of enhancer–promoter interactions, promoter regulatory steps and promoter states predicted by the mechanistic model with $n = 5$ and rates satisfying the constraint on timescales described in **d** (time unit, $1/\delta$). **f**, Reduction of the mechanistic model to an apparent two-state model. The equation describes how the apparent on rate $k_{on}^{app}$ depends on contact probability ($p_c$) and other parameters of the mechanistic model. **g**, Dependency of $k_{on}^{app}$ on contact probability, illustrated for the best fitting parameters shown in **h** and **i**. **h**, Best fit of the apparent two-state model to the experimental transcriptional response shown in Fig. 2b. **i**, Best fit of the apparent two-state model to the experimental mRNA distributions shown in Fig. 2c. Best-fit parameters are shown in Extended Data Fig. 4c.

experienced by promoters in mES cells (Extended Data Fig. 2b, c). However, such a trend is at odds with our observation that transcription levels rather mildly decreased inside the TAD and dropped to promoter-only levels outside its boundaries (Fig. 1h and Extended Data Fig. 2d). Interestingly, plotting the mean *eGFP* mRNA numbers as a function of contact probabilities revealed a highly nonlinear relationship (Fig. 2b).

We sought to understand whether such a nonlinear relationship could be related to how enhancer–promoter interactions translate into transcription in individual cells. Transcription occurs in intermittent bursts[35] that give rise to variable mRNA numbers in single cells. smRNA-FISH analysis revealed substantial cell-to-cell variability in *eGFP* mRNA numbers in a panel of cell lines in which promoter–SCR contact probabilities ranged from zero (promoter-only control cell line) to one (Fig. 2c). Similar to eGFP protein distributions (Extended Data Fig. 2e), CVs of mRNA distributions increased with decreasing contact probabilities (Extended Data Fig. 2f). Bursty promoter behaviour can generally be described in terms of a two-state model of gene expression[36] in which the promoter stochastically switches with rates $k_{on}$ and $k_{off}$ between an OFF and an ON state in which transcription can initiate with rate $\mu$. Consistent with this notion, mRNA number distributions (Fig. 2d) and mean transcription levels (Fig. 2e) in individual cell lines could be well approximated by a phenomenological two-state model in which the 'on' rate $k_{on}$ (and therefore the burst frequency) nonlinearly depends on enhancer–promoter contact probability through a Hill function (Fig. 2f and Supplementary Information, model description). Interestingly, the best agreement with experimental data occurred with a Hill coefficient ($h$) of 2.8 (95% confidence interval = 2.4–3.2; Extended Data Fig. 3a, b). This corresponds to a sigmoidal transcriptional response in which the enhancer would be no

longer able to activate the promoter outside the approximately threefold drop in contact probabilities generated by TAD boundaries (Fig. 2g, h). Importantly the sigmoidal behaviour of $k_{on}$ was not an artefact due to systematic errors in estimation of contact probabilities (Extended Data Fig. 3c), confounding effects of CTCF sites and repressive chromatin in the 3′ part of the TAD, or inclusion of promoter-only cell lines in the fit (Extended Data Fig. 3d). Alternative two-state models in which 'off' or initiation rates depend on contact probability rather than the on rate failed to reproduce the observed decrease in CV with contact probabilities (Supplementary Information, model description).

## Mechanistic model of enhancer regulation

We next examined which mechanism could in principle generate such a phenomenological two-state model with sigmoidal modulation of $k_{on}$. Enhancer–promoter contacts are stochastic[32,37,38] and probably dynamic[39] in single cells. Molecular processes that are thought to transmit regulatory information from enhancers to promoters (such as recruitment of transcription factors and coactivators, assembly of the Mediator complex[40]), as well as those that are associated with promoter operation itself (such as pre-initiation complex assembly, RNA polymerase II pausing and release[41,42]) are also stochastic and dynamic[43]. We reasoned that the interplay between the timescales of these processes might generate nonlinear effects, as was recently hypothesized to explain promoter bursting[44]. To investigate this concept in a quantitative manner, we developed a mechanistic model describing the simple hypothesis that, in single cells, the on rate of the promoter is transiently increased after stochastic interactions with an enhancer. We assumed that

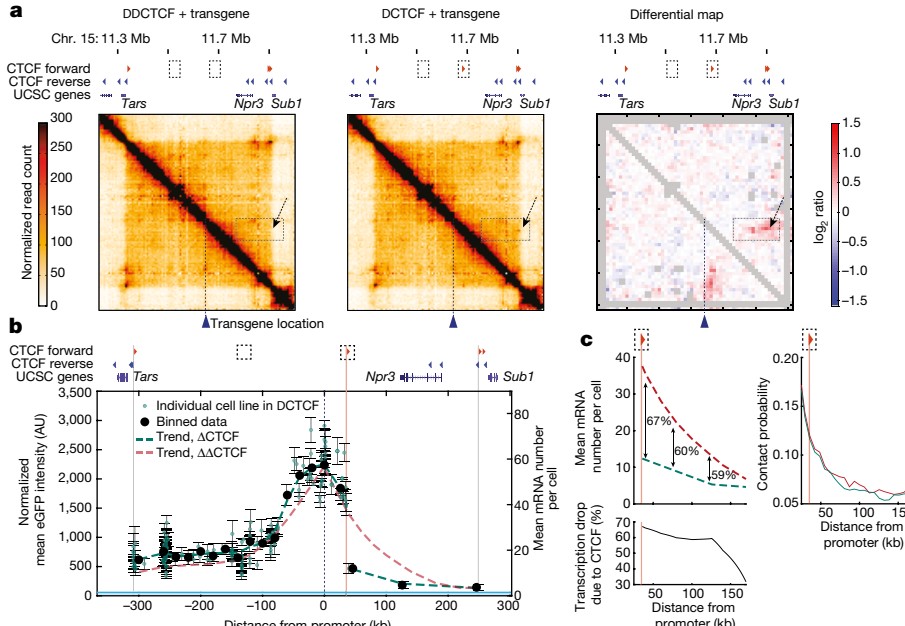

**Fig. 4 | Insulation by a single CTCF site exceeds contact probability changes. a**, Capture-C (6.4 kb resolution) analysis of founder mES cell lines in the absence (ΔΔCTCF) or presence (single CTCF-site deletion, ΔCTCF) of a forward CTCF motif 36 kb downstream of the transgene, and the corresponding differential map. The grey pixels show 'noisy' interactions that did not pass quality control filters (Methods). The dotted boxes and arrows indicate the position of the CTCF site and the structural changes it generates. **b**, The normalized mean ± s.d. eGFP levels in 172 individual eGFP⁺ cell lines following SCR mobilization in ΔCTCF mES cells (green dots); $n = 3$ measurements performed on different days. The black dots show the mean ± s.d. values within equally spaced 20 kb bins. The green dashed line shows the spline interpolation of average eGFP values. The vertical pink line shows the position of the CTCF site at +36 kb. The red dashed line shows the trend of eGFP levels in the ΔΔCTCF background (compare with Fig. 1h). The blue line shows the promoter-only eGFP level as in Fig. 1h. **c**, Magnification of spline interpolants of GFP⁺ cell lines in the absence (ΔΔCTCF, red dashed line) or presence (ΔCTCF, green dashed line) of the CTCF binding site at +36 kb (vertical pink line) (left). The numbers represent the percentage fold changes between trendlines. Bottom, the percentage fold changes as a function of distance from the promoter. Right, contact probabilities from the location of the ectopic *Sox2* transgene in ΔΔCTCF (red line) and ΔCTCF (green line) mES cells.

enhancer–promoter interactions occur and disassemble with rates $k_{close}$ and $k_{far}$, corresponding to a steady-state contact probability of $k_{close}/(k_{close} + k_{far})$ (Fig. 3a). When the enhancer is close to the promoter, it triggers one or more ($n$) reversible regulatory steps that transmit information to the promoter with forward and reverse rates $k_{forward}$ and $k_{back}$ (Fig. 3b). These steps are an abstract representation of any stochastic regulatory processes occurring at the enhancer–promoter interface. When the enhancer is far, no information is transmitted to the promoter and regulatory steps can only revert at rate $k_{back}$ (Fig. 3b). The promoter operates in a basal two-state regime with a small on rate ($k_{on}^{basal}$) (Fig. 3c) unless all regulatory steps have been completed. In this case, the promoter transiently enters an 'enhanced' two-state regime with a higher on rate ($k_{on}^{enh}$), thus transiently increasing its transcriptional activity (Fig. 3c and Supplementary Information, model description). A transient increase in promoter activity therefore requires enhancer interactions that are either long enough (Extended Data Fig. 4a) or frequent enough (Extended Data Fig. 4b) to allow the completion of the $n$ regulatory steps.

This mechanistic model does not generally reproduce the phenomenological two-state behaviour observed in Fig. 2e, f for the ectopic *Sox2* promoter. However, when the timescales of enhancer–promoter interactions are faster than those of intermediate regulatory steps, and both are faster than the promoter's intrinsic bursting dynamics ($k_{close,far} \gg k_{forward,back} \gg k_{on}^{basal,enh}, k_{off}, \mu$) (Fig. 3d, e), the mechanistic model reduces to an apparent two-state model (Fig. 3f and Supplementary Information, model description). If forward transitions through $n > 1$ regulatory steps are favoured over backward reactions ($k_{forward} > k_{back}$), then the on rate of the apparent two-state model ($k_{on}^{app}$) depends sigmoidally on contact probabilities (Fig. 3g). This shows that, in principle, the promoter's phenomenological two-state behaviour with sigmoidal modulation of $k_{on}$ observed in Fig. 2e, f could arise from stochastic enhancer–promoter

interactions being transmitted into slower promoter ON/OFF dynamics through small numbers of intermediate regulatory processes. The resulting sigmoidal transcriptional response would enable an enhancer to act efficiently even when contact probabilities rapidly decay away from the promoter (Extended Data Fig. 2d), and contribute to block enhancer action when small drops in contact probabilities occur across TAD boundaries (Fig. 2h). The mechanistic model also predicts that enhancer–promoter contacts should not correlate with transcription bursts (Fig. 3e), as recently suggested by simultaneous imaging of *Sox2* transcription and genomic locations flanking the endogenous *Sox2* and SCR[20].

Finally, we verified that, when reduced to a two-state model, the mechanistic model could simultaneously fit the experimental transcriptional response to contact probabilities and smRNA-FISH distributions (Fig. 3h, i). Best agreement occurred with five intermediate regulatory steps (95% confidence interval = 3–7; Extended Data Fig. 4c, d and Supplementary Information, model description) and, consistent with previous observations[20], promoter ON/OFF transitions that occur in the timescale of several minutes (considering that the time unit in the model is mRNA lifetime, expected to be around 1.5 h)[45] (Extended Data Fig. 4c, d). Regulatory processes at the interface between enhancers and promoters have been estimated to occur in the order of tens of seconds[41,43,46], consistent with the condition that intermediate regulatory steps should be faster than bursting kinetics (Fig. 3f). The requirement that enhancer–promoter interactions should be even faster (Fig. 3f) therefore predicts that they should occur on a timescale of seconds or less.

## Enhancer strength controls insulation levels

We next set out to examine whether CTCF binding affects the observed nonlinear relationship between transcription and contact probabilities.

To this aim, we repeated the enhancer mobilization assay in mES cells in which only one of the two internal CTCF sites was homozygously deleted. The remaining forward CTCF site is located 36 kb downstream of the transgene and loops onto the reverse CTCF sites at the 3′ end of the domain (Fig. 4a). SCR mobilization in this context resulted in 172 cell lines of which the transcription levels were indistinguishable from those generated in the 'empty' TAD, except across the CTCF site that severely, but not completely, insulated the ectopic *Sox2* promoter from the enhancer (Fig. 4b). Transcription levels across the CTCF site were about 60% lower than those generated in the absence of the CTCF site (Fig. 4c). Strikingly, this occurred in the absence of notable changes in the promoter's interaction probabilities with the region downstream of the CTCF site, at least in the current experimental set-up (capture-C data with 6.4 kb resolution) (Fig. 4c). This suggests that a single CTCF site might exert transcriptional insulation through additional mechanisms beyond simply driving physical insulation, possibly depending on site identity[47] and flanking sequences[16].

The SCR is a strong enhancer that accounts for most of the transcriptional activity of endogenous *Sox2*[29,30]. We reasoned that a weaker enhancer should lead to a different transcriptional response to contact probabilities with the promoter. There are two ways in which the parameters in the model shown in Fig. 3f might change when reducing enhancer strength. The ratio between transition rates through regulatory steps $k_{forward}$ and $k_{back}$ ($\beta$ in Fig. 3h) might decrease, resulting in a slower transmission of regulatory information (Fig. 5a). This would generate a transcriptional response with maximal transcriptional levels that are similar to those generated by the SCR but different sensitivity to changes in contact probabilities (Fig. 5a). Alternatively (although not exclusively), the on rate in the enhanced promoter regime $k_{on}^{enh}$ could decrease (Fig. 5b). This would conserve the shape of the transcriptional response but decrease the maximal transcription level (Fig. 5b). To test these predictions, we performed the enhancer mobilization assay using a truncated version of the SCR (Extended Data Fig. 5a). This contained only one of the two ~1.5 kb subregions that share similar transcription-factor-binding sites[29] and independently operate as weaker enhancers of the *Sox2* promoter in transient reporter assays[29] (Extended Data Fig. 5b). Mobilization of the truncated SCR in mES cells with a forward CTCF site downstream of the promoter (compare with Fig. 4a) led to 74 eGFP⁺ cell lines displaying approximately twofold lower transcription levels compared with those generated by the full-length SCR at comparable genomic distances (Fig. 5c). In contrast to the full-length SCR, the truncated enhancer was completely insulated from the promoter by the CTCF site (Fig. 5c). Thus, the level of functional insulation generated by the same CTCF site depends on the strength of the enhancer. In the region upstream of the CTCF sites, the transcriptional response generated by the truncated SCR (Fig. 5d) was in quantitative agreement with model predictions under the hypothesis that enhancer strength decreases the on rate rather than changing the intermediate regulatory steps (Fig. 5b), and could be predicted using the full-length SCR best-fit parameters with a two-fold decreased $k_{on}^{enh}$. This further strengthens our interpretation that enhancer strength modulates the ability of the promoter to turn on, possibly by regulating chromatin state, transcription factor binding or RNA polymerase II dynamics at the promoter[35,44].

In the nonlinear transcriptional response that we identified, high sensitivity in the low contact probability regime (that is, at long genomic distances) might contribute to secure insulation by TAD boundaries of even strong enhancers such as the SCR. Interestingly, in mES cells, the contact probabilities of most (~75%) active promoters with the nearest TAD boundary are comparable to those experienced by the ectopic *Sox2* promoter in our experiments (lower than 0.2) (Extended Data Fig. 5c). These promoters should therefore experience the same insulation mechanisms. The remaining promoters are closer (or adjacent) to a TAD boundary and therefore experience larger contact probabilities with the boundary, at which the transcriptional response is less sensitive (Extended Data Fig. 5d). However, interestingly, drops in contact

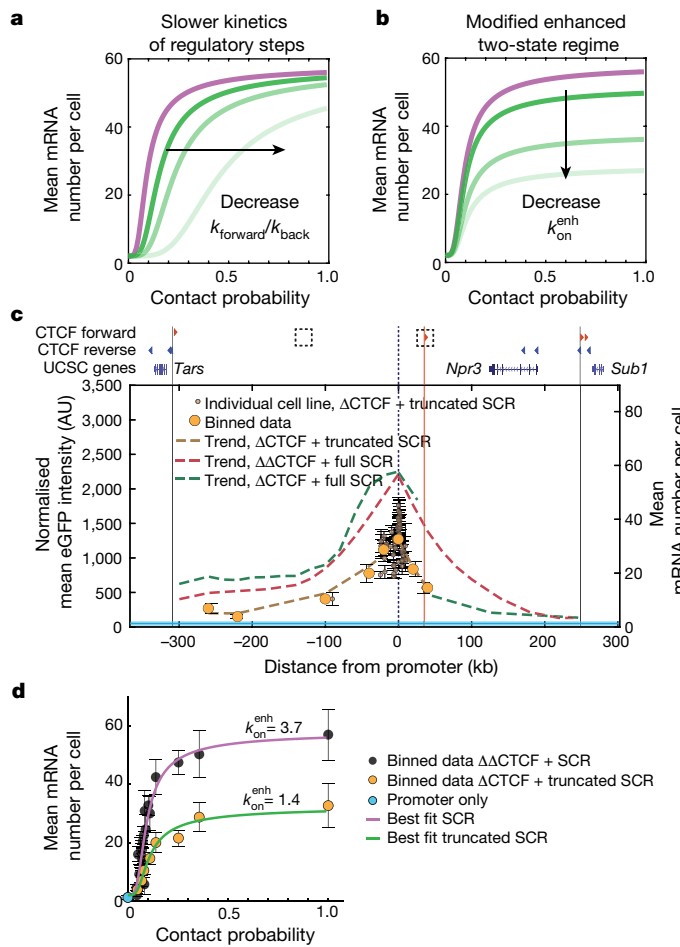

**Fig. 5 | Enhancer strength modulates promoter on rates and determines insulation levels through a CTCF site. a**, Model predictions under the hypothesis that decreasing enhancer strength results in a slower flow of regulatory information to the promoter. **b**, Model predictions as in **a**, under the alternative hypothesis that decreasing enhancer strength modifies the enhanced on rate ($k_{on}^{enh}$). **c**, Normalized eGFP levels in in 74 individual GFP⁺ cell lines (brown dots; the error bars show the s.d. of $n = 3$ measurements performed on different days), binned data (orange dots) and data trend (brown dashed line) after mobilization of the truncated SCR in the ΔCTCF background. Trends of eGFP levels in individual GFP⁺ cell lines in which the SCR was mobilized either in the ΔΔCTCF background (red dashed line; Fig. 1h) or in the ΔCTCF background (green dashed line; Fig. 4b) are shown for comparison. Promoter-only eGFP levels (light blue) are shown as in Fig. 1h. **d**, The transcriptional response of the truncated SCR (green line) can be predicted from the best fit to the full-length SCR (purple line) with a modified enhanced on rate ($k_{on}^{enh}$). Data are mean ± s.d. eGFP values were calculated within equally spaced 20 kb bins as in **c**; the number of cell lines per bin varies from 1 to 56.

probabilities across a boundary increase with decreasing genomic distance from the boundary itself (Extended Data Fig. 5d). This might contribute to the functional insulation of this class of promoters. Boundaries associated with clusters of CTCF sites might also benefit from the fact that insulation from CTCF sites can exceed the changes in contact probabilities that they generate (Fig. 4).

## Discussion

Our study provides unbiased and systematic measurements of promoter output as a function of large numbers of enhancer positions with minimal confounding effects. The analysis of hundreds of cell lines enables us to move beyond locus-specific observations, and establishes a quantitative framework for understanding the role of chromosome

structure in long-range transcriptional regulation. Our data reveal that, within a TAD, absolute transcription levels generated by an enhancer depend on its genomic distance from the promoter and are determined by a nonlinear relationship with their contact probabilities. Minimal regulatory and structural complexities introduce deviations from this behaviour and might therefore confound its detection outside a highly controlled genomic environment, notably when studying regulatory sequences in their endogenous context[23]. Mathematical modelling suggests that the observed nonlinear transcriptional response involves a modulation of the promoter's burst frequency, which could arise from transient enhancer–promoter interactions being translated into slower promoter bursting dynamics in individual cells. In addition to readily explaining the absence of correlation between transcription and physical proximity in single-cell experiments, this argues that the absence of such correlation should not be interpreted as the absence of causality. Although alternative explanations cannot be ruled out (such as cooperative effects through biomolecular condensates[21,48]), our model provides a simple explanatory framework for both population-averaged and single-cell behaviour of enhancer-driven transcription, based on a minimal set of general and realistic hypotheses. Future live-cell imaging experiments with improved spatial and temporal resolution[49] will probably enable the testing of the model's prediction that enhancer–promoter interactions should occur on a timescale of seconds or less, therefore enabling the assessment of the model's premises. Finally, our study reveals that enhancer strength is not only a determinant of absolute transcription levels, but also of the level of insulation provided by CTCF. Our data therefore imply that transcriptional insulation is not an intrinsic absolute property of TAD boundaries or CTCF interactions but, rather, a graded variable depending on enhancer strength, boundary strength and distance from a promoter.

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

## Methods

### Culture of embryonic stem cells

All cell lines are based on E14 mES cells, provided by E. Heard's laboratory. Cells were cultured on gelatin-coated culture plates in Glasgow minimum essential medium (Sigma-Aldrich, G5154) supplemented with 15% fetal calf serum (Eurobio Abcys), 1% L-glutamine (Thermo Fisher Scientific, 25030024), 1% sodium pyruvate MEM (Thermo Fisher Scientific, 11360039), 1% MEM non-essential amino acids (Thermo Fisher Scientific, 11140035) 100 µM β-mercaptoethanol, 20 U ml$^{-1}$ leukaemia inhibitory factor (Miltenyi Biotec, premium grade) in 8% $CO_2$ at 37 °C. Cells were tested for mycoplasma contamination once a month and no contamination was detected. After piggyBac-enhancer transposition, cells were cultured in standard E14 medium supplemented with 2i (1 µM MEK inhibitor PDO35901 (Axon, 1408) and 3 µM GSK3 inhibitor CHIR 99021 (Axon, 1386)).

### Generation of enhancer–promoter piggyBac targeting vectors

Homology arms necessary for the knock-in, the *Sox2* promoter, the SCR and the truncated version of the SCR (Ei) were amplified from E14 mES cell genomic DNA by Phusion High-Fidelity DNA Polymerase (Thermo Fisher Scientific, F549) using primers compatible with Gibson assembly cloning (NEB, E2611). The targeting vector was generated starting from the 3-SB-EF1-PBBAR-SB plasmid[50], gifted by Rob Mitra. To clone homology arms into the vector, BspEI and BclI restriction sites were introduced using Q5 Site-Directed Mutagenesis Kit (NEB, E0554). The left homology arm was cloned using Gibson assembly strategy by linearizing the vector with BspEI (NEB, R0540). The right homology arm was cloned using Gibson assembly strategy by linearizing the vector with BclI (NEB, R0160). The *Sox2* promoter was cloned by first removing the Ef1a promoter from the 3-SB-EF1-PBBAR-SB vector using NdeI (NEB, R0111) and SalI (NEB, R0138) and subsequently using Gibson assembly strategy. The SCR and its truncated version (truncated SCR or Ei) were cloned between the piggyBac transposon-specific inverted terminal repeat sequences (ITR) by linearizing the vector with BamHI (NEB, R3136) and NheI (NEB, R3131). A transcriptional pause sequence from the human alpha2 globin gene and an SV40 poly(A) sequence were inserted at both 5′ and 3′ ends of the enhancers using Gibson assembly strategy. A selection cassette carrying the puromycin resistance gene driven by the PGK promoter and flanked by FRT sites was cloned in front of the *Sox2* promoter by linearizing the piggyBac vector with the AsiSI (NEB, R0630) restriction enzyme. A list of the primers used for cloning is provided in Supplementary Table 1.

### Generation of founder mES cell lines carrying the piggyBac transgene

The gRNA sequence for the knock-in of the piggyBac transgene on chromosome 15 was designed using the online tool (https://eu.idtdna.com/site/order/designtool/index/CRISPR_SEQUENCE) and purchased from Microsynth AG. gRNA sequence was cloned into the PX459 plasmid (Addgene) using the BsaI restriction site. E14 mES cell founder lines carrying the piggyBac transgene were generated using nucleofection with the Amaxa 4D-Nucleofector X-Unit and the P3 Primary Cell 4D-Nucleofector X Kit (Lonza, V4XP-3024 KT). Cells ($2 \times 10^6$) were collected with accutase (Sigma-Aldrich, A6964) and resuspended in 100 µl transfection solution (82 µl primary solution, 18 µl supplement, 1 µg piggyBac targeting vector carrying the SCR, truncated SCR or promoter alone, and 1 µg of PX459 ch15_gRNA/Cas9) and transferred into a single Nucleocuvette (Lonza). Nucleofection was performed using the protocol CG110. Transfected cells were directly seeded in prewarmed 37 °C culture in E14 standard medium. Then, 24 h after transfection, 1 µg ml$^{-1}$ of puromycin (InvivoGen, ant-pr-1) was added to the medium for 3 days to select cells transfected with PX459 gRNA/Cas9 vector. Cells were then cultured in standard E14 medium for an additional 4 days. To select cells with insertion of the piggyBac targeting

vector, a second pulse of puromycin was carried out by culturing cells in standard medium supplemented with 1 µg ml$^{-1}$ of puromycin. After 3 days of selection, single cells were isolated by fluorescence-activated cell sorting (FACS) on 96-well plates. Sorted cells were kept for 2 days in standard E14 medium supplemented with 100 µg µl$^{-1}$ primocin (InvivoGen, ant-pm-1) and 10 µM ROCK inhibitor (STEMCELL Technologies, Y-27632). Cells were then cultured in standard E14 medium with 1 µg ml$^{-1}$ of puromycin. Genomic DNA was extracted by lysing cells with lysis buffer (100 mM Tris-HCl pH 8.0, 5 mM EDTA, 0.2% SDS, 50 mM NaCl, proteinase K and RNase) and subsequent isopropanol precipitation. Individual cell lines were analysed by genotyping PCR to determine heterozygous insertion of the piggyBac donor vector. Cell lines showing the corrected genotyping pattern were selected and expanded. A list of the primers used for genotyping is provided in Supplementary Table 1.

### Puromycin resistance cassette removal

Cells ($1 \times 10^6$) were transfected with 2 µg of a pCAG-FlpO-P2A-HygroR plasmid encoding for the flippase (Flp) recombinase using Lipofectamine 3000 (Thermo Fisher Scientific, L3000008) according to the manufacturer's instructions. Transfected cells were cultured in standard E14 medium for 7 days. Single cells were then isolated using FACS on 96-well plates. Genomic DNA was extracted by lysing cells with lysis buffer (100 mM Tris-HCl pH 8.0, 5 mM EDTA, 0.2% SDS, 50 mM NaCl, proteinase K and RNase) and subsequent isopropanol precipitation. Individual cell lines were analysed by genotyping PCR to verify the deletion of the puromycin resistance cassette. A list of the primers used for genotyping is provided in Supplementary Table 1. Cell lines showing the correct genotyping pattern were selected and expanded. Selected cell lines were processed for targeted Nanopore sequencing with Cas9-guided adapter ligation (nCATS)[51] and only the ones showing unique integration of the piggyBac donor vector were used as founder lines for the enhancer mobilization experiments.

### Mobilization of the piggyBac-enhancer cassette

A mouse codon-optimized version of the piggyBac transposase (PBase) was cloned in frame with the red fluorescent protein tagRFPt (Evrogen) into a pBroad3 vector (pBroad3_hyPBase_IRES_tagRFPt) using Gibson assembly cloning (NEB, E2611). Cells ($2 \times 10^5$) were transfected with 0.5 µg of pBroad3_hyPBase_IRES_tagRFPt using Lipofectamine 3000 (Thermo Fisher Scientific, L3000008) according to the manufacturer's instructions. To increase the probability of enhancer transposition, typically 12 independent PBase transfections were performed at the same time in 24-well plates. Transfection efficiency as well as expression levels of hyPBase_IRES_tagRFPt transposase within the cell population were monitored by flow cytometry analysis. Then, 7 days after transfection with PBase, individual eGFP$^+$ cell lines were isolated using FACS in 96-well plates. Sorted cells were kept for 2 days in standard E14 medium supplemented with 100 µg ml$^{-1}$ primocin (InvivoGen, ant-pm-1) and 10 µM ROCK inhibitor (STEMCELL Technologies, Y-27632). Cells were cultured in E14 standard medium for additional 7 days and triplicated for genomic DNA extraction, flow cytometry analysis and freezing.

### Sample preparation for mapping piggyBac-enhancer insertion sites in individual cell lines

Mapping of enhancer insertion sites in individual cell lines was performed using splinkerette PCR. The protocol was performed as described previously[52] with a small number of modifications. Genomic DNA from individual eGFP$^+$ cell lines was extracted from 96-well plates using the Quick-DNA Universal 96 Kit (Zymo Research, D4071) according to the manufacturer's instructions. Purified genomic DNA was digested by 0.5 µl of Bsp143I restriction enzyme (Thermo Fisher Scientific, FD0784) for 15 min at 37 °C followed by a heat-inactivation step at 65 °C for 20 min. Long (HMSpAa) and short (HMSpBb) splinkerette adapters were first resuspended with 5× NEBuffer 2 (NEB, B7002) to reach a concentration of 50 µM. Then, 50 µl of HMSpA adapter was

mixed with 50 µl of HMSpBb adapter (Aa+Bb) to reach a concentration of 25 µM. The adapter mix was denatured and annealed by heating it to 95 °C for 5 min and then cooling to room temperature. Then, 25 pmol of annealed splinkerette adapters was ligated to the digested genomic DNA using 5 U of T4 DNA ligase (Thermo Fisher Scientific, EL0011) and incubating the samples for 1 h at 22 °C followed by a heat-inactivation step at 65 °C for 10 min. For splinkerette amplifications, PCR 1 was performed combining 2 µl of the splinkerette sample, 1 U of Platinum Taq polymerase (Thermo Fisher Scientific, 10966034), 0.1 µM of HMSp1 and 0.1 µM of PB5-1 (or PB3-1) primer, and splinkerette PCR 2 was performed using 2 µl of PCR 1, 1 U of Platinum Taq polymerase (Thermo Fisher Scientific, 10966034), 0.1 µM of HMSp2 and 0.1 µM of PB5-5 (or PB3-2) primer. The quality of PCR amplification was checked by agarose gel electrophoresis. Samples were sent for Sanger Sequencing (Microsynth AG) using the PB5-2 (or PB3-2) primer. A list of the primers used for splinkerette PCRs and sequencing is provided in Supplementary Table 1. Mapping of enhancer insertion sites in individual cell lines was performed as described in the 'Mapping of piggyBac-enhancer insertion sites in individual cell lines' section.

### Flow cytometry eGFP fluorescence intensity measurements and analysis

eGFP⁺ cell lines were cultured in serum + 2i medium for 2 weeks before flow cytometry measurements. eGFP levels of individual cell lines were measured on the BD LSRII SORP flow cytometer using BD High Throughput Sampler (HTS), which enabled sample acquisition in 96-well plate format. Measurements were repeated three times for each clone. Mean eGFP fluorescence intensities were calculated for each clone using FlowJo and all three replicates were averaged.

### Normalization of mean eGFP fluorescence intensities

Mean eGFP fluorescence levels of each cell line measured in flow cytometry were first corrected by subtracting the mean eGFP fluorescence intensities measured in wild-type E14 mES cells cultured in the same 96-well plate. The resulting mean intensities were then normalized by dividing them by the average mean intensities of all cell lines where the SCR was located within a 40 kb window centred at the promoter location, and multiplied by a common factor.

### Sample preparation for high-throughput sequencing of piggyBac-enhancer insertion sites

Cells ($5 \times 10^5$) were transfected with 2 µg of PBase using Lipofectamine 3000 (Thermo Fisher Scientific, L3000008) according to the manufacturer's instructions. Transfection efficiency as well as expression levels of PBase within the cell population were monitored by flow cytometry analysis. Then, 5 days after transfection with PBase, genomic DNA was purified using the DNeasy Blood & Tissue Kit (Qiagen, 69504). To reduce the contribution from cells in which excision of piggyBac-enhancer did not occur, we depleted eGFP sequences using an in vitro Cas9 digestion strategy. gRNA sequences for eGFP depletion were designed using the online tool (https://eu.idtdna.com/site/order/designtool/index/CRISPR_SEQUENCE) (Supplementary Table 1). Custom-designed Alt-R CRISPR-Cas9 crRNAs containing the gRNA sequences targeting eGFP (gRNA_1_3PRIME and gRNA_2_3PRIME), Alt-R CRISPR-Cas9 tracrRNA (IDT, 1072532) and Alt-R *Streptococcus pyogenes* Cas9 enzyme (IDT, 1081060) were purchased from IDT. In vitro cleavage of the eGFP fragment by Cas9 was performed according to the IDT protocol 'In vitro cleavage of target DNA with ribonucleoprotein complex'. In brief, 100 µM of Alt-R CRISPR–Cas9 crRNA and 100 µM of Alt-R CRISPR–Cas9 tracrRNA were assembled by heating the duplex at 95 °C for 5 min and allowing to cool to room temperature (15–25 °C). To assemble the RNP complex, 10 µM of Alt-R guide RNA (crRNA:tracrRNA) and 10 µM of Alt-R *Sp*Cas9 enzyme were incubated at room temperature for 45 min. To perform in vitro digestion of eGFP, 300 ng of genomic DNA extracted from the pool cells transfected with the PBase was incubated for 2 h with

1 µM Cas9/RNP. After the digestion, 40 µg of proteinase K was added and the digested sample was further incubated at 56 °C for 10 min to release the DNA substrate from the Cas9 endonuclease. After purification using AMPURE beads XP (Beckman Coulter, A63881), genomic DNA was digested by 0.5 µl of Bsp143I restriction enzyme (Thermo Fisher Scientific, FD0784) for 15 min at 37 °C followed by a heat-inactivation step at 65 °C for 20 min. Annealed splinkerette adapters (Aa+Bb; 125 pmol) were then ligated to the digested genomic DNA using 30 U of T4 DNA ligase HC (Thermo Fisher Scientific, EL0013), and the samples were incubated for 1 h at 22 °C followed by a heat-inactivation step at 65 °C for 10 min. For splinkerette amplifications, 96 independent PCR 1 reactions were performed combining 100 ng of the splinkerette sample, 1 U of Platinum Taq polymerase (Thermo Fisher Scientific, 10966034), 0.1 µM of HMSp1 and 0.1 µM of PB3-1 primer, and splinkerette PCR 2 was performed using 4 µl of PCR 1 product, 1 U of Platinum Taq polymerase (Thermo Fisher Scientific, 10966034), 0.1 µM of HMSp2 and 0.1 µM of PB3-2 primer. A list of the primers used for splinkerette PCRs is provided in Supplementary Table 1. Splinkerette amplicon products were processed using the NEB Ultra II kit according to the manufacturer's protocol, using 50 ng of input material. Mapping of genome-wide insertions was performed as described in the 'Mapping of piggyBac-enhancer insertion sites in population-based splinkerette PCR' section.

### Sample preparation for tagmentation-based mapping of PiggyBac insertions

PiggyBac integrations in pools of cells were mapped using a Tn5-transposon-based ITR mapping technique based on ref. [53] with minor alterations. Cells ($2 \times 10^5$) were transfected with 0.5 µg of PBase using Lipofectamine 3000 (Thermo Fisher Scientific, L3000008) according to the manufacturer's instructions in 24-well plates. Eight independent transfections were performed in parallel. Transfection efficiency as well as expression levels of PBase within the cell population were monitored by flow cytometry analysis. Then, 7 days after transfection with PBase, 6 cell pools of 10,000 cells from low GFP values (gates low 1 and low 2) and 6 cell pools of 337 cells of high GFP values (gate high) were sorted in a 24-well plate. Sorted cells were kept for 2 days in standard E14 medium supplemented with 100 µg ml⁻¹ primocin (InvivoGen, ant-pm-1) and 10 µM ROCK inhibitor (StemCell Technologies, Y-27632). Cells were cultured in E14 standard medium for either 1 passage (pools from gates low 1 and low 2) or 2 passages (pools from gate high) and genomic DNA from individual pools was extracted using the Quick-DNA Miniprep Plus Kit (Zymo Research, D4069) according to the manufacturer's instructions. The Tn5 transposon was produced as described in ref. [54]. The tagmentation reaction was performed as follows. The primers TAC0101 & TAC0102 (45 µl of 100 µM) each were mixed with 10 µl 10× Tris-EDTA (pH 8) and annealed by heating to 95 °C followed by a slow ramp down (0.1 °C s⁻¹) until 4 °C. The transposome is obtained by combining the adapters (1 µl of 1:2 diluted adapters) and the Tn5 transposon (1.5 µl of 2.7 mg ml⁻¹ stock) in 18.7 µl Tn5 dilution buffer (20 mM HEPES, 500 mM NaCl, 25% glycerol) and incubating the mix for 1 h at 37 °C. The tagmentation was performed by mixing 100 ng of genomic DNA with 1 µl of assembled transposome, 4 µl 5× TAPS-PEG buffer (50 mM TAPS-NAOH, 25 mM MgCl₂, 8% (v/v) PEG8000) in a final volume of 20 µl. The reaction was incubated at 55 °C for 10 min and quenched with 0.2% SDS afterwards. For the best mapping results, both sides of the PiggyBac transposon were processed to obtain 5' ITR- and 3' ITR-specific libraries. First, we enriched our target region by linear amplification PCR with 3' ITR-specific (TAC0006) and 5' ITR-specific (TAC0099) primers. The PCR mix was 3 µl of tagmented DNA, 1 µl of 1 µM enrichment primer, 2 µl dNTPs (10 mM), 4 µl 5× Phusion HF Buffer (NEB), 0.25 µl Phusion HS Flex polymerase (2 U µl⁻¹, NEB), in a final volume of 20 µl and amplified as follows: 30 s at 98 °C; 45 cycles of 10 s at 98 °C, 20 s at 62 °C and 30 s at 72 °C; then 20 s at 72 °C. PCR 1 of the library preparation was performed using TAC0161 (3' ITR) and TAC0110 (5' ITR) in combination with N5xx (Illumina, Nextera Index Kit).

The PCR mix was 5 µl of enrichment PCR, 1 µl of 10 µM primers, 2 µl dNTPs (10 mM), 4 µl 5× Phusion HF Buffer and 0.25 µl Phusion HS Flex polymerase (NEB), in a final volume of 25 µl and amplified as follows: 30 s at 98 °C; 3 cycles of 10 s at 98 °C, 20 s at 62 °C and 30 s at 72 °C; and 8 cycles of 10 s at 98 °C, 50 s at 72 °C. In PCR 2 the N7xx (Illumina, Nextera Index Kit) adapters were added to the PiggyBac specific locations as follows. PCR was performed with TAC0103 (both ITRs) and N7xx. The PCR mix was 2 µl of PCR1, 1 µl of 10 µM primers, 2 µl dNTPs (10 mM), 4 µl 5× Phusion HF Buffer and 0.25 µl Phusion polymerase (Thermo Fisher Scientific), in a final volume of 22 µl and amplified as follows: 30 s at 98 °C; 10 cycles of 10 s at 98 °C, 20 s at 63 °C and 30 s at 72 °C. Then, 5 µl of library was checked on a 1% agarose gel and different samples were pooled according to smear intensity. Finally, the library was purified by bead purification using CleanPCR (CleanNA) beads at a ratio 1:0.8 sample:beads. The final library was sequenced using the Illumina MiSeq (150 bp, paired-end) system. Mapping of genome-wide insertions was performed as described in the 'Mapping of piggyBac-enhancer insertion sites by tagmentation' section.

## Deletion of genomic regions containing CTCF-binding sites

gRNA sequences for depletion of the genomic regions containing the CTCF-binding sites were designed using the online tool (https://eu.idtdna.com/site/order/designtool/index/CRISPR_SEQUENCE) and purchased from Microsynth AG (Supplementary Table 1). gRNA sequences were cloned into the PX459 plasmid (Addgene) using the BsaI restriction site. To remove the first forward CTCF-binding site (chromosome 15: 11520474–11520491), $3 \times 10^5$ cells were transfected with 0.5 µg of PX459 CTCF_KO_gRNA3/Cas9 and 1 µg of PX459 CTCF_KO_gRNA10/Cas9 plasmids using Lipofectamine 2000 (Thermo Fisher Scientific, 11668019) according to the manufacturer's instructions. To remove the second forward CTCF-binding sites (chromosome 15: 11683162–11683179), $1 \times 10^6$ cells were transfected with 1 µg of PX459 gRNA2_CTCF_KO/Cas9 and 1 µg of PX459 gRNA6_CTCF_KO/Cas9 plasmids using Lipofectamine 2000 (Thermo Fisher Scientific, 11668019) according to the manufacturer's instructions. Then, 24 h after transfection, 1 µg ml⁻¹ of puromycin was added to the medium for 3 days. Cells were then cultured in standard E14 medium for an additional 4 days. To select cell lines with homozygous deletion, single cells were isolated by FACS on 96-well plate. Sorted cells were kept for 2 days in E14 standard medium supplemented with 100 µg ml⁻¹ primocin (InvivoGen, ant-pm-1) and 10 µM ROCK inhibitor (STEMCELL Technologies, Y-27632). Cells were then cultured in standard E14 medium. Genomic DNA was extracted by lysing cells with lysis buffer (100 mM Tris-HCl pH 8.0, 5 mM EDTA, 0.2% SDS, 50 mM NaCl, proteinase K and RNase) and subsequent isopropanol precipitation. Individual cell lines were analysed by genotyping PCR to determine homozygous deletion of the genomic regions containing the CTCF-binding sites. Cell lines showing the corrected genotyping pattern were selected and expanded. A list of the primers used for genotyping is provided in Supplementary Table 1.

## smRNA-FISH

Cells were collected with accutase (Sigma-Aldrich, A6964) and adsorbed on poly-L-lysine (Sigma-Aldrich, P8920) precoated coverslips. Cells were then fixed with 3% PFA (EMS, 15710) in PBS for 10 min at room temperature, washed with PBS and kept in 70% ethanol at −20 °C. After at least 24 h incubation in 70% ethanol, the coverslips were incubated for 10 min with freshly prepared wash buffer composed of 10% formamide (Millipore Sigma, S4117) in 2× SSC (Sigma-Aldrich, S6639). The coverslips were hybridized overnight (around 16 h) at 37 °C in freshly prepared hybridization buffer composed of 10% formamide, 10% dextran sulfate (Sigma-Aldrich, D6001) in 2× SSC and containing 125 nM of RNA-FISH probe sets against *Sox2* labelled with Quasar 670 (Stellaris) and against eGFP labelled with Quasar 570 (Stellaris). After hybridization, the coverslips were

washed twice with wash buffer prewarmed to 37 °C for 30 min at 37 °C with shaking, followed by 5 min incubation with 500 ng ml⁻¹ DAPI solution (Sigma-Aldrich, D9564) in PBS (Sigma-Aldrich, D8537). The coverslips were then washed twice in PBS and mounted on slides with Prolong Gold medium (Invitrogen, P36934) and cured at room temperature for 24 h. The coverslips were then sealed and imaged within 24 h.

## RNA-FISH image acquisition

Images were acquired on a Zeiss Axion Observer Z1 microscope equipped with 100 mW 561 nm and 100 mW 642 nm HR diode solid-state lasers, an Andor iXion 885 EMCCD camera, and an α Plan-Fluar ×100/1.45 NA oil-immersion objective. Quasar 570 signal was collected with the DsRed ET filter set (AHF Analysentechnik, F46-005), Quasar 670 with Cy5 HC mFISH filter set (AHF Analysentechnik, F36-760) and DAPI with the Sp. Aqua HC-mFISH filter set (AHF Analysentechnik, F36-710). The typical exposure time for RNA-FISH probes was set to around 300–500 ms with 15–20 EM gain and 100% laser intensity. DAPI signal was typically imaged with an exposure time of 20 ms with EM gain 3 and 50% laser intensity. The pixel size of the images was 0.080 × 0.080 µm with a z-step of 0.25 µm for around 55–70 z-planes.

## Image processing and quantification of mRNA numbers

Raw images were processed in KNIME, python and Fiji to extract the numbers of RNAs per cell. The KNIME workflow described below is based on a previously published workflow[55]. z-stacks were first projected to a maximal projection for each fluorescence channel. Individual cells were then segmented using the DAPI channel using Gaussian convolution (σ = 3), followed by filtering using global threshold with Otsu filter, watershed and connected component analysis for nuclei segmentation. Cytoplasmic areas were then estimated with seeded watershed. Cells with nuclei partially outside the frame of view were automatically excluded. Cells containing obvious artifacts, wrongly segmented or not fully captured in *xyz* dimensions were manually excluded from the final analysis. Spot detection is based on the Laplacian of Gaussian method implemented in TrackMate[56]. For the channels containing RNA-FISH probes signal, RNAs spots were detected after background subtraction (rolling ball radius 20–25 pixels) by selecting spot size 0.2 µm and threshold for spot detection based on visual inspection of multiple representative images. Spot detection is based on the Laplacian of Gaussian method from TrackMate. Subpixel localization of RNA spots was detected for RNA channels and a list of spots per cell for each experimental condition and replicate was generated. Spots in each channel were then aggregated by cell in python to extract the number of RNAs per cell.

## Enhancer reporter assays

To generate vectors for the enhancer reporter assay, the *Sox2* promoter, SCR and the truncated versions of the SCR (Ei and Eii) were amplified from E14 mES cell genomic DNA with Phusion High-Fidelity DNA Polymerase (Thermo Fisher Scientific, F549) using primers compatible with Gibson assembly strategy. The *Sox2* promoter was cloned into the 3-SB-EF1-PBBAR-SB vector as described above. The SCR and the truncated versions Ei and Eii were cloned in front of the *Sox2* promoter by linearizing the vector with AgeI (NEB, R3552) and subsequently using Gibson assembly cloning. A transcriptional pause sequence from the human α2-globin gene and an SV40 poly(A) sequence was inserted at both the 5′ and 3′ ends of the enhancers. To test enhancers activity, $3 \times 10^5$ cells were co-transfected with 0.5 µg of the different versions piggyBac vectors and 0.5 µg of pBroad3_hyPBase_IRES_tagRFPt using Lipofectamine 2000 (Thermo Fisher Scientific, 11668019) according to the manufacturer's instructions. As a control, only 0.5 µg of the piggyBac vector carrying the Sox2 promoter was transfected. 24 h after transfection, cells were collected and analysed by flow cytometry.

## Capture-C sample preparation

Cells ($20 \times 10^6$) were cross-linked with 1% formaldehyde (EMS, 15710) for 10 min at room temperature and quenched with glycine (final concentration, 0.125 M). Cells were lysed in 1 M Tris-HCl pH 8.0, 5 M NaCl and 10% NP40 and complete protease inhibitor (Sigma-Aldrich, 11836170001) and enzymatically digested using 1,000 U of MboI (NEB, R0147). Digested chromatin was then ligated at 16 °C with 10,000 U of T4 DNA ligase (NEB, M0202) in ligase buffer supplemented with 10% Triton X-100 (Sigma-Aldrich, T8787) and 240 µg of BSA (NEB, B9000). Ligated samples were de-cross-linked with 400 µg proteinase K (Macherey Nagel, 740506) at 65 °C and phenol–chloroform purified. 3C library preparation and target enrichment using a custom-designed collection of 6,979 biotinylated RNA 'baits' targeting single MboI restriction fragments chromosome 15: 10283500–13195800 (mm9) (Supplementary Table 2; Agilent Technologies; designed as in ref. [57]) were performed according to the SureSelectXT Target Enrichment System for Illumina Paired-End Multiplexed Sequencing Library protocol. The only exceptions were the use of 9 µg of 3C input material (instead of 3 µg) and shearing of DNA using Covaris sonication with the following settings: duty factor: 10%; peak incident power: 175; cycles per burst: 200; treatment time: 480 s; bath temperature: 4 °C to 8 °C.

## Targeted nCATS analysis

gRNA sequences targeting specific genomic regions of chromosome 15 external to the homology arms of the transgene were designed using the online tool (https://eu.idtdna.com/site/order/designtool/index/CRISPR_SEQUENCE) (Supplementary Table 1). Custom-designed Alt-R CRISPR–Cas9 crRNAs (5 crRNAs targeting the region upstream and 5 crRNAs targeting the region downstream the integrated transgene), Alt-R CRISPR–Cas9 tracrRNA (IDT, 1072532) and Alt-R *Sp*Cas9 enzyme (IDT, 1081060) were purchased from IDT. Sample preparation and Cas9 enrichment were performed according to a previously described protocol[51] with a few modifications. Genomic DNA from mES cell founder lines was extracted using the Gentra Puregene Cell Kit (Qiagen, 158745) according to the manufacturer's instructions. The quality of the high molecular mass DNA was checked using the TapeStation (Agilent) system. Typically, 5 µg of high molecular mass DNA was processed for incubation using shrimp alkaline phosphatase (rSAP; NEB, M0371) for 30 min at 37 °C followed by 5 min at 65 °C to dephosphorylate DNA-free ends. For Cas9 enrichment of the target region, all ten Alt-R CRISPR-Cas9 crRNAs were first pooled at an equimolar amount (100 µM) and subsequently incubated with 100 µM of Alt-R CRISPR–Cas9 tracrRNA at 95 °C for 5 min to assemble the Alt-R guide RNA duplex (crRNA:tracrRNA). To assemble the RNP complex, 4 pmol of Alt-R *Sp*Cas9 enzyme was incubated with 8 pmol Alt-R guide RNA (crRNA:tracrRNA) at room temperature for 20 min. In vitro digestion and A-tailing of the DNA were performed by adding 10 µl of the RNP complex, 10 mM of dATP (NEB, N0440) and 5 U of Taq Polymerase (NEB, M0267) and incubating the samples for 30 min at 37 °C followed by 5 min at 72 °C. Adapter ligation for Nanopore sequencing was performed using the Ligation Sequencing Kit (Nanopore, SQK-CAS109) according to the manufacturer's instructions. After purification with AMPure PB beads (Witec, 100-265-900), the samples were loaded into the MniION system, selecting the SQK-CAS109 protocol.

## Nanopore sequencing analysis

To map Nanopore sequencing reads, we first built a custom genome consisting of the transgene sequence flanked by ~10 kb mouse genomic sequence upstream and downstream of the target integration site. The custom genome can be found at GitHub (https://github.com/zhanyinx/Zuin_Roth_2021/blob/main/Nanopore/cassette/cassette.fa). Reads were mapped to the custom genome using minimap2 (v.2.17-r941) with the '-x map-ont' parameter. Nanopore sequencing analysis has been implemented using Snakemake workflow (v.3.13.3). Reads were visualized using IGV (v.2.9.4). The full workflow can be found at GitHub (https://github.com/zhanyinx/Zuin_Roth_2021).

## RNA-sequencing sample preparation and analysis

Mouse embryonic stem cells were collected with accutase (5 min, 37 °C) and counted. Cells ($3 \times 10^5$) were lysed with 300 µl TRIzol reagent. RNA was extracted using the Direct-Zol RNA extraction kit from Zymo. Library preparation was performed after Illumina TruSeq Stranded mRNA-seq according to the manufacturer protocol. Reads were mapped to the *Mus musculus* genome (build mm9) using STAR[58], using the following options: --outSJfilterReadsUnique --outFilterType BySJout --outFilterMultimapNmax 10 --alignSJoverhangMin 6 --alignSJDBoverhangMin 2 --outFilterMismatchNoverLmax 0.04 --alignIntronMin 20 --alignIntronMax 1000000 --outSAM strandField intronMotif --outFilterIntronMotifs RemoveNoncanonicalUnannotated --outSAMtype BAM SortedByCoordinate --seedSearchStartLmax 50 --twopassMode basic. Gene expression was quantified using qCount from QuasR package[59] using the 'TxDb.Mmusculus.UCSC.mm9.knownGene' database for gene annotation (Bioconductor package: Carlson M and Maintainer BP. TxDb.Mmusculus.UCSC. mm9.knownGene: Annotation package for TxDb object(s); R package v.3.2.2). Active promoters were defined as genes with $\log_2[RPKM + 0.1]$ higher than 1.5.

## Capture-C analysis

Capture-C data were analysed using HiC-Pro[60] (v.2.11.4); the parameters can be found at GitHub (https://github.com/zhanyinx/Zuin_Roth_2021). In brief, read pairs were mapped to the mouse genome (build mm9). Chimeric reads were recovered after recognition of the ligation site. Only unique valid pairs mapping to the target regions were used to build contact maps. Iterative correction[61] was then applied to the binned data. The target regions can be found at GitHub (https://github.com/zhanyinx/Zuin_Roth_2021). For SCR_ΔΔCTCF, SCR_ΔCTCF and the derived clonal lines, data from replicate one were used to make the quantification and plots throughout the manuscript.

## Differential capture-C maps

To evaluate the structural perturbation induced by the insertion of the transgene and the mobilization of the enhancer (ectopic sequences), we accounted for differences in genomic distances due to the presence of the ectopic sequence. In the founder cell line (for example, SCR_ΔΔCTCF), insertion of the transgene modifies the genomic distance between loci upstream and downstream the insertion site. To account for these differences, we generated distance-normalized capture-C maps in which each entry corresponds to the interaction normalized to the corrected genomic distance between the interacting bins. Outliers (defined using the interquartile rule) or bins with no reported interactions from capture-C were treated as noise and filtered out. Singletons, defined as the top 0.1 percentile of *Z*-score, were also filtered out. The *Z*-score is defined as (obs − exp)/stdev, where obs is the capture-C signal for a given interaction and exp and stdev are the genome-wide average and standard deviation, respectively, of capture-C signals at the genomic distance separating the two loci. We next calculated the ratios between distance normalized and noise-filtered capture-C maps. A bilinear smoothing with a window of 2 bins was applied to the ratio maps to evaluate the structural perturbation induced by the insertion of the ectopic sequence.

## Chromatin state calling with ChromHMM

Chromatin states were called using ChromHMM[28] with four states. The list of histone modification datasets used is provided in Supplementary Table 3. States with enrichment in H3K9me3 and H3K27me3 were merged, therefore resulting in three chromatin states: active (enriched

in H3K27ac, H3K36me3, H3K4me1 and H3K9ac), repressive (enriched in H3K9me3 and H3K27me3) and neutral (no enrichment).

## Mapping of piggyBac-enhancer insertion sites in population-based splinkerette PCR

To identify true-positive enhancer re-insertion sites, we first filtered out reads containing eGFP fragments. We then retained only read pairs for which one side mapped to the ITR sequence and the other side mapped to the splinkerette adapter sequence. We mapped separately the ITR/splinkerette sides of the read pair to the mouse genome (build mm9) using BWA mem[62] with the default parameters. Only integration sites that had more than 20 reads from both ITR and splinkerette sides were retained.

## Mapping of piggyBac-enhancer insertion sites in individual cell lines

To map the enhancer position in individual cell lines, Sanger sequencing (Microsynth) without the adapter sequences were filtered out. The first 24 bp of each read after the adapter was then mapped to the mouse genome (mm9) using vmatchPattern (Biostrings v.2.58.0). The script used to map Sanger sequencing can be found at GitHub (https://github.com/zhanyinx/Zuin_Roth_2021).

## Mapping of piggyBac-enhancer insertion sites by tagmentation

Before aligning paired-end sequencing reads, reads were filtered using an adaptation of cutadapt[63], processing each read pair in multiple steps. Sequence patterns originating from Tn5 and each ITR were removed. The paired-end reads coming from both ITRs were treated the same. First, the presence of the unique part of the 5′ ITR and 3′ ITR sequence was detected at the start of the second read of the pair and, if present, this sequence was trimmed. Next, the sequence up to and including the TTAA site that was found on both the 5′ITR and 3′ITR was trimmed off. This sequence only partly contained the respective primers used for each ITR, and was used to filter reads that contained the sequence expected for a correct PCR product starting at the transposon. The sequence up to, but not including, the TTAA was removed. Next, all of the other sequence patterns coming from either Tn5 or the ITR were removed from the 5′ end of the first read in the pair and the 3′ end of both reads.

After filtering and trimming the reads, the reads were aligned to a reference genome with an in silico insertion of the split-GFP construct, but with a single TTAA motif instead of the PiggyBac transposon. This was done by aligning the homology arms found in the plasmid against mm10 reference genome. The complete sequence on the reference matching both arms was replaced by the plasmid sequence inserted.

Alignment was performed using Bowtie2 with the fragment length set to a minimum of 0 bp and maximum of 2,000 bp and the very-sensitive option was used. After reads were aligned to the genome, sambamba[64] was used to remove duplicates and samtools[65] was used to filter out read pairs that were not properly paired. We then designated, for each read pair, the position of the first 4 nucleotides of the second read as a putative insertion site. To calculate the fraction of reads originating from the non-mobilized position, the number of read pairs that overlapped the non-mobilized position (the TTAA replacing the PiggyBac of the in silico insert) was divided over the total number of reads originating from putative insertion sites supported by at least one read pair with a mapping quality higher than 2. Confident insertions were identified as those with at least one read for both 5′ and 3′ ITR.

## Calibration of the mean number of mRNAs per cell with smRNA-FISH

A linear model was used to predict the average number of eGFP mRNAs on the basis of the mean eGFP intensity. The model was fitted on 7 data points corresponding to the average number of eGFP mRNAs obtained using single-molecule RNA fluorescence in situ and the mean eGFP intensity obtained by flow cytometry (Extended Data Fig. 1h; $R^2 = 0.9749$, $P < 0.0001$, $t$-test).

## Mathematical model and parameter fitting

The phenomenological two-state model (Fig. 2) and the apparent two-state model deduced from the mechanistic enhancer–promoter model (Fig. 3) were both fitted simultaneously to the mean eGFP levels measured in individual cell lines and to the distributions of RNA numbers measured by smRNA-FISH in six cell lines where the SCR was located at different distances from the promoter. The mean number of mRNAs was calculated analytically and the steady-state distribution of the number of mRNA per cell was approximated numerically (Supplementary Information, model description). The parameters for the phenomenological two-state model are the minimum on rate $k_{on}^{0}$, the minimum on rate $k_{on}^{1}$, the off rate $k_{off}$, the initiation rate $\mu$ and the constant $c$ and Hill exponent $h$, which together control the nonlinear dependency of $k_{on}$ on contact probability. The parameters for the apparent two-state model are the basal on rate $k_{on}^{basal}$, the enhanced on rate $k_{on}^{enh}$, the off rate $k_{off}$, the initiation rate $\mu$, the ratio between the forward and backward rates of the regulatory steps $\beta$ and the number of regulatory steps $n$. All of these parameters were considered to be free in the fitting procedure. The apparent two-state model was also fitted to the binned mean number of mRNA molecules inferred from the eGFP[+] cell lines with the truncated version of the SCR (Fig. 4). In this case, three versions of the apparent two-state model were fitted to the data using log-transformed likelihood ratios. The parameter $\beta$ (version 1) or $k_{on}^{enh}$ (model 2) or both (model 3) were considered to be free parameters, whereas the other parameters were fixed to the best fit values obtained for the full-length SCR dataset. Using log-transformed likelihood ratios, the fit of the three versions was compared to the fit of the model for which all of the parameters were considered to be free. The mathematical description of the enhancer–promoter communication model, the derivation of the apparent two-state model, and the fitting procedures are explained in detail in the Supplementary Information (model description).

## Reporting summary

Further information on research design is available in the Nature Research Reporting Summary linked to this paper.

## Data availability

All capture-C, RNA-seq, Oxford Nanopore, tagmentation and population-based splinkerette PCR sequencing fastq files generated in this study have been uploaded to the Gene Expression Omnibus (GEO) under accession number GSE172257. The following public databases were used: BSgenome.Mmusculus.UCSC.mm9 (https://bioconductor.org/packages/release/data/annotation/html/BSgenome.Mmusculus.UCSC.mm9.html), TxDb.Mmusculus.UCSC.mm9.knownGene (https://bioconductor.org/packages/release/data/annotation/html/TxDb.Mmusculus.UCSC.mm9.knownGene.html).

## Code availability

Custom codes generated in this study are available at GitHub (https://github.com/zhanyinx/Zuin_Roth_2021 (cHiC, Nanopore, Insertion mapping); https://github.com/gregroth/Zuin_Roth_2021 (mathematical model); and https://github.com/vansteensellab/tagmap_hopping/tree/giorgetti (tagmentation-based mapping of PiggyBac insertions)).

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

**Acknowledgements** We thank R. Mitra for sharing the piggyBac-splitGFP vector; A. Boettiger and J.Y. Xiao for discussions on modelling; M. Michalski and S. Andrews for capture-C primer design; L. Gelman and J. Eglinger for help with microscopy and image analysis; and G. Natoli, D. Schübeler, H. Grosshans and E. Nora for discussions and comments on the manuscript. J.Z. was supported by a Marie Skłodowska-Curie grant (748091 '3DQuant'). Research in the Giorgetti laboratory is funded by the Novartis Foundation, the European Research Council (ERC) (759366 'BioMeTre') and Marie Skłodowska-Curie Innovative Training Networks (813327 'ChromDesign' and 813282 'PEP-NET') under the European Union's Horizon 2020 research and innovation program, and the Swiss National Science Foundation (310030_192642). Research in the Meister laboratory is supported by the Swiss National Science Foundation (IZCOZ0_189884/31003A_176226 to P. Meister.). Research in the van Steensel laboratory is supported by ERC Advanced Grant 694466 'GoCADiSC'.

**Author contributions** L.G. and J.Z. conceived and designed the study. J.Z., J.C., E.P., M.K. and G.T. performed the experiments. G.R. wrote and analysed the mathematical model. H.K. provided assistance with flow cytometry. J.C., P. Meister. and S.S. contributed to setting up nCATS. S.S. also provided assistance with high-throughput sequence experiments. J.Z., J.C., J.R. and P. Mach. generated cell lines. M.E., C.L. and B.v.S. assisted with tagmentation-based mapping of insertions. G.R. and Y.Z. analysed the data, except for flow cytometry and single-clone insertion mapping (J.Z.) and smRNA-FISH (E.P.). L.G. wrote the paper with G.R., J.Z. and Y.Z., and input from all of the authors.

**Competing interests** The authors declare no competing interests.

**Additional information**
**Correspondence and requests for materials** should be addressed to Luca Giorgetti.

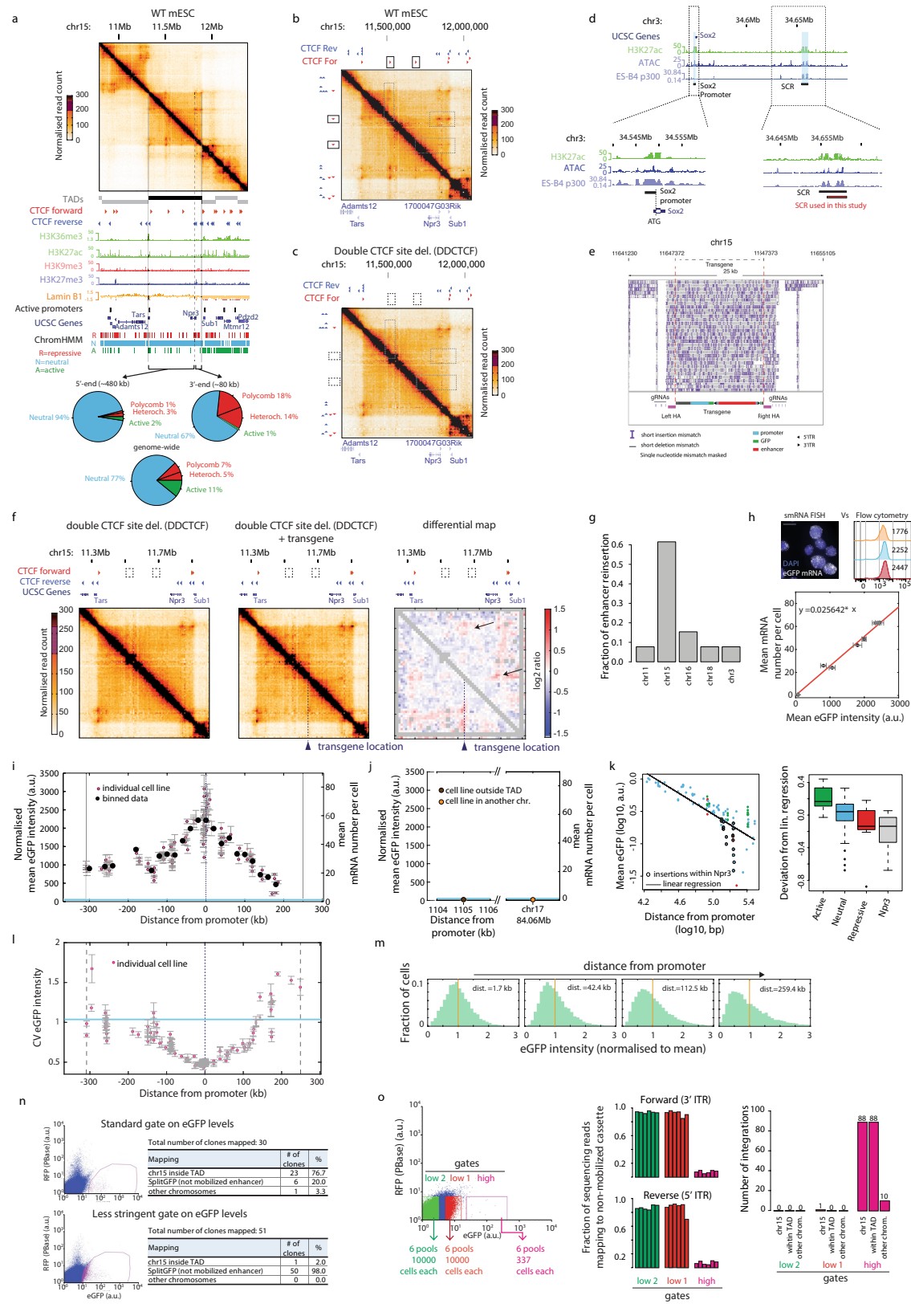

**Extended Data Fig. 1** | See next page for caption.

**Extended Data Fig. 1 | Enhancer action is modulated by genomic distance from the target promoter and constrained by TAD boundaries.** a. Top: capture-C contact map at 6.4 kb resolution in wild-type (WT) mES cells in a 2.6 Mb region centred around the neutral TAD on chromosome 15 we used for the experiments. Vertical grey lines: TAD boundaries. Bottom: genomic datasets and ChromHMM analysis showing that the chosen TAD is devoid of active and repressive chromatin states, with the exception of 80 kb at the 3b at t which is enriched in repressive chromatin states. b. Close-up view of panel **a**, highlighting the presence of CTCF-mediated chromatin loops (dotted boxes) in WT mES cells. c. capture-C contact map at 6.4 kb resolution for the same region as panel b in the cell line with double CTCF site deletions. CTCF deletions lead to loss of CTCF-mediated chromatin loops (dotted boxes). d. Top: UCSC snapshot of the endogenous *Sox2* locus and *Sox2* control region (SCR). Bottom: close-up views showing the regions of the *Sox2* promoter, the SCR region found in ref. [29] and the SCR used in the transgene construct. e. IGV snapshot showing nanopore sequencing reads mapped to a modified mouse genome including the transgene integration. Reads spanning from genomic DNA upstream the left homology arm to genomic DNA downstream the right homology arm confirmed single insertion of the transgene. f. capture-C maps at 6.4 kb resolution of the mES cell line with double CTCF sites deletion (left) and the founder mES cell line with transgene insertion (centre). Right: differential contact map. Grey pixels correspond to 'noisy' interactions that did not satisfy our quality control filters (see Methods). Transgene insertion induces new mild interactions with CTCF sites at the 3. and 5a extremities of the TAD (arrows). g. Barplot showing the fraction of piggyBac-SCR reinsertions genome-wide determined by Illumina sequencing of splinkerette PCR products from a pool of cells after PBase expression. See Methods for a detailed description of the protocol. h. Top: Representative smRNA-FISH image and flow cytometry profiles over different passages in a cell line where the SCR was mobilized in the immediate vicinity of the ectopic *Sox2* promoter. Scale bar, 10 $\mu m$. Bottom: Linear relationship between the mean eGFP intensity and the average number of eGFP mRNAs measured using smRNA-FISH for seven single cell lines ($R^2 = 0.9749$, $p < 0.0001$, t-test). Error bars on the x-axis: standard deviation of three measurements performed on different days, as in Fig. 1h. Error bars on the y-axis: standard deviation of three technical replicates. i. Normalized mean eGFP intensities levels in individual eGFP+ cell lines are plotted as a function of the genomic position of the SCR in individual eGFP+ lines. Data from 127 individual cell lines (light red dots) from a single experiment are presented as mean +\- standard deviation (n=3 measurements performed in different days, as in Fig. 1g). Average eGFP values calculated within equally spaced 20 kb bins

(black dots) are shown. Mean mRNA numbers per cell were inferred from eGFP counts using calibration with smRNA-FISH, see Extended Data Fig. 1h. Shaded light blue area indicates the interval between mean +/- standard deviation of eGFP levels in three promoter-only cell lines. j. Same plot as Fig. 1h showing the only two SCR insertions we detected outside the TAD boundaries (brown dot) and on another chromosome (yellow dot). k. Left: Log10 average eGFP expression (from Fig. 1h) as a function of log10 absolute genomic distance between transgene position and SCR reinsertion. Points are colour-coded as in panel A (chromHMM active, neutral, and repressive states). Black line denotes linear regression. Black circles denote SCR reinsertions within the Npr3 gene body. Right: deviations of eGFP expression levels from the linear regression correlate with chromatin states called using ChromHMM (n: active = 16; neutral = 83; Npr3 = 17; repressive = 7). Reinsertion of SCR within active or repressive regions respectively increases or decreases enhancer activity compared to neutral regions. Box plot: centre line denotes the median; boxes denote lower and upper quartiles (Q1 and Q3, respectively); whiskers denote 1.5x the interquartile region (IQR) below Q1 and above Q3; points denote outliers. l. Coefficients of variation (CV) of eGFP levels measured by flow cytometry plotted against SCR insertion locations in eGFP+ cell lines (light red dots). Data are presented as mean +/- standard deviation (n = 3 measurements in different days). Shaded light blue area indicates the interval between mean +/- standard deviation of eGFP level CVs in three promoter-only cell lines. m. Representative eGFP distributions (normalized to mean eGFP level) in clones with increasing absolute genomic distance (1.7 kb, 42.4 kb, 112.5 kb, and 259.43 kb) between the mobilized enhancer and the ectopic *Sox2* promoter. Vertical line indicates normalized mean eGFP levels. n. FACS plot showing standard (top) and less stringent (bottom) gates on eGFP levels used for single cells sort and insertion analysis of corresponding clonal cell lines. o. Left: FACS plot showing the gates used to sort pools of cells for tagmentation-based mapping of PiggyBac-enhancer insertions. For gates "low1" and "low 2", six pools of 10000 cells were sorted while for gate "high", six pools of 337 cells were sorted. Gate "high" corresponds to the standard gate used to isolate eGFP positive cell lines for the mobilization experiments. Centre: Barplot showing the fraction of sequencing reads mapping to non-mobilized enhancer cassette determined by tagmentation-based mapping from the different pools sorted in gates "low 1", "low 2" and "high". See Methods for a detailed description of the protocol. Right: Numbers and genomic locations of confident insertion sites (identified as those with at least one read for both 5'oth 5 mapping from the different pools sorted in gates "low1", "low 2" and "higeGFP gates.

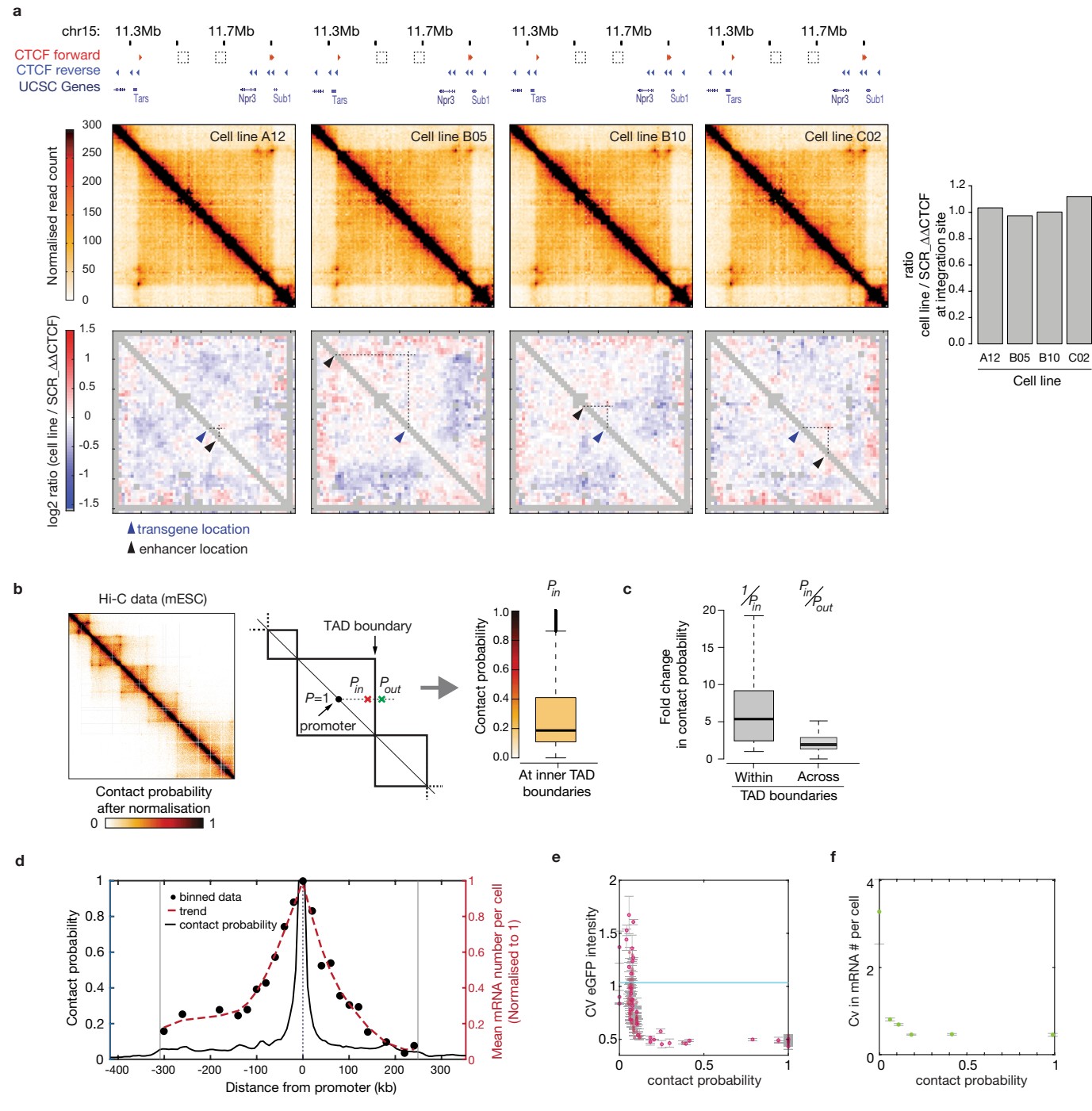

**Extended Data Fig. 2** | See next page for caption.

**Extended Data Fig. 2 | Analysis of chromosome structure around the transgenic locus and genome-wide in mES cells.** a. Top: capture-C maps (6.4 kb resolution) of four cell lines where the SCR (black arrow) has been reinserted at different distances from the promoter (blue arrow). Bottom: differential contact map between individual cell lines and the founder line. Grey pixels: correspond to 'noisy' interactions that did not satisfy quality control filters (see Methods). Right: barplot showing the change in average interaction probabilities between the SCR reinsertion and the cassette, calculated using a square of 5 bins (6.4 kb resolution) centred at the cassette SCR reinsertion interaction. b. Left: example of Hi-C heatmap in mES cells at 6.4 kb resolution. Centre: scheme depicting how the probability of interaction between a promoter and the region immediately before the nearest TAD boundary ($P_{in}$, 12.8 kb i.e. two 6.4 kb bins before the boundary called using CaTCH[66]) and after the nearest TAD boundary ($P_{out}$) are calculated. Right: distribution of contact probability between all active promoters in mES cellss and the closest inner TAD boundary ($P_{in}$) (n = 9655). Box plot description as in Extended Data Fig. 1k. c. Box plots showing the distribution of contact probability changes within the TAD and across the closest TADs boundary for all active promoters in mES cells (n = 9655) whose contact probability outside the TAD is higher than 0.001 (n = 834). Box plot description as in Extended Data Fig. 1k; outliers not shown. d. Contact probabilities of the founder line from the location of the ectopic *Sox2* transgene (black line) and normalized averaged mean number of mRNAs per cell (highest value = 1) generated in individual eGFP+ lines by the SCR mobilization are plotted as a function of its genomic position (dashed red line). The average is calculated within equally spaced 20 kb bins as in Fig. 1h (black dots). e. Coefficients of variation (CV) of eGFP levels measured by flow cytometry plotted against contact probabilities between the ectopic *Sox2* promoter and the locations of SCR insertions. Data are presented as mean values +/- standard deviation (n = 3 measurements in different days). Shaded light blue area indicates the interval between mean +/- standard deviation of eGFP level CVs in three promoter-only cell lines. f. Coefficients of variation (CV) of mRNA number per cell measured by smRNA-FISH plotted against contact probabilities between the ectopic *Sox2* promoter and the locations of SCR in the cell the lines shown in Fig. 2c, d. Data are presented as mean values +/- standard deviation (n = 3 technical replicates).

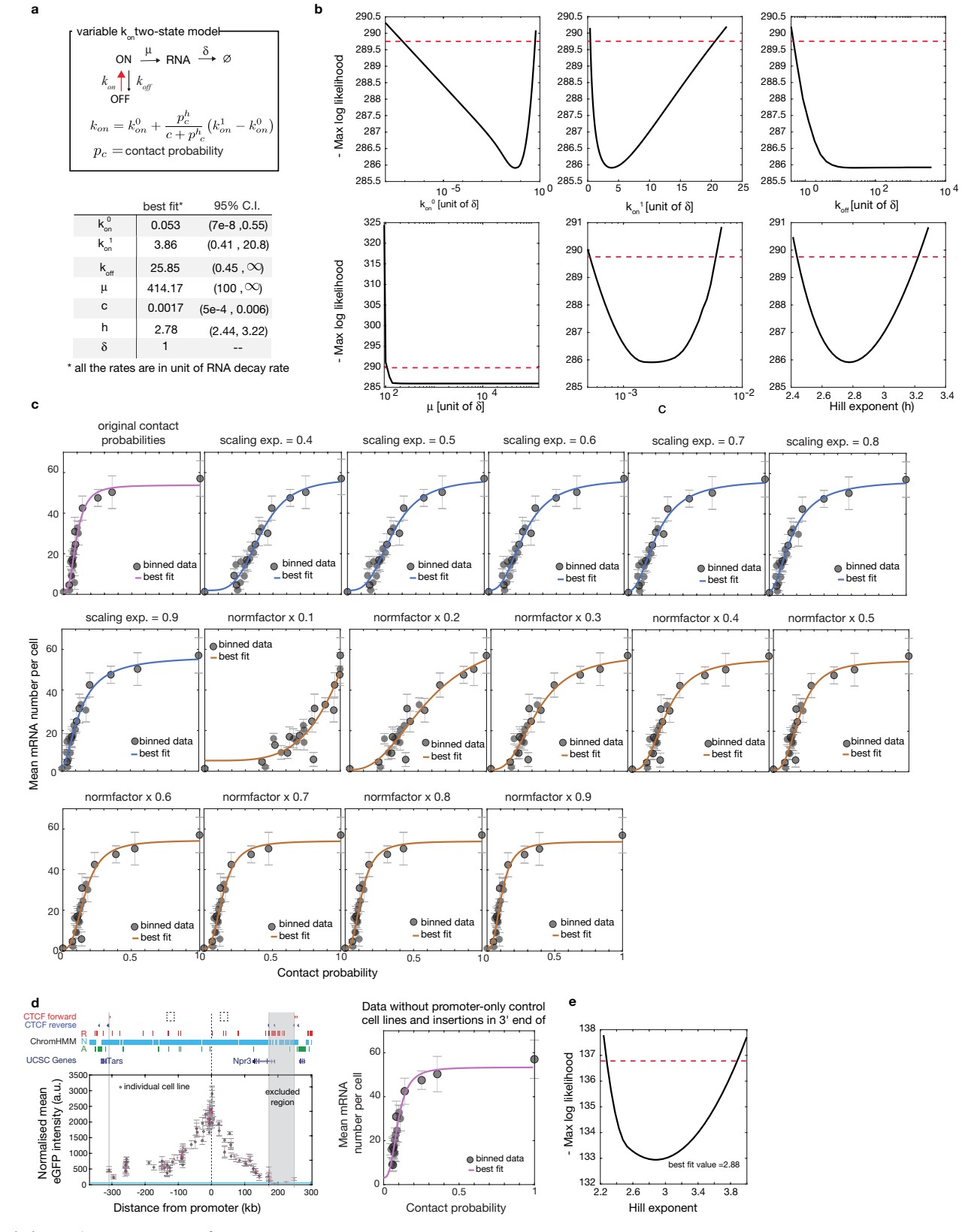

**Extended Data Fig. 3** | See next page for caption.

**Extended Data Fig. 3 | Phenomenological two-state model fitting and robustness analysis.** a. Parameter values and 95% confidence intervals for the best fitting phenomenological two-state model. The rates are in the unit of RNA decay rate ($\delta$). b. Profile likelihood functions for all the parameters of the phenomenological two-state model. The red dashed line shows the threshold used to calculate the 95% confidence intervals (see Supplementary Model description for more details). c. Best fit of the phenomenological two-state model under different perturbations of the contact probabilities. Panels with blue curves show the best fit transcriptional responses when the scaling exponent of the contact probabilities was artificially set to 0.4, 0.5, 0.6, 0.7, 0.8, and 0.9. The scaling exponent of the original contact probabilities is 0.77. Panels with orange curves show the best fit transcriptional responses when contact probabilities were artificially increased by a factor 1/x with x = 0.1,...,0.9 with step of 0.1. Data are presented as average eGFP values calculated within equally spaced 20 kb bins +/- standard deviation (n = number of cell lines per bin), as in Fig. 1h. d. Left: Normalized mean eGFP intensities in individual eGFP+ cell lines are plotted as a function of the genomic position of the SCR. Data from 135 individual cell lines (light red dots) are presented as mean +/- standard deviation (n = 3 measurements performed on different days, as in panel g). Shaded grey area indicates the genomic regions that were excluded from the fit shown in the right panel. Right: Best fit of the phenomenological two-state model in the absence of the promoter-only control cell line and the cell lines with insertions that landed beyond the first CTCF site at the 3′ of the TAD (region highlighted in the left panel). Data are presented as average eGFP values calculated within equally spaced 20 kb bins +/- standard deviation (n = number of cell lines per bin). e. Profile likelihood function for the Hill coefficient for the fit described in panel **d**.

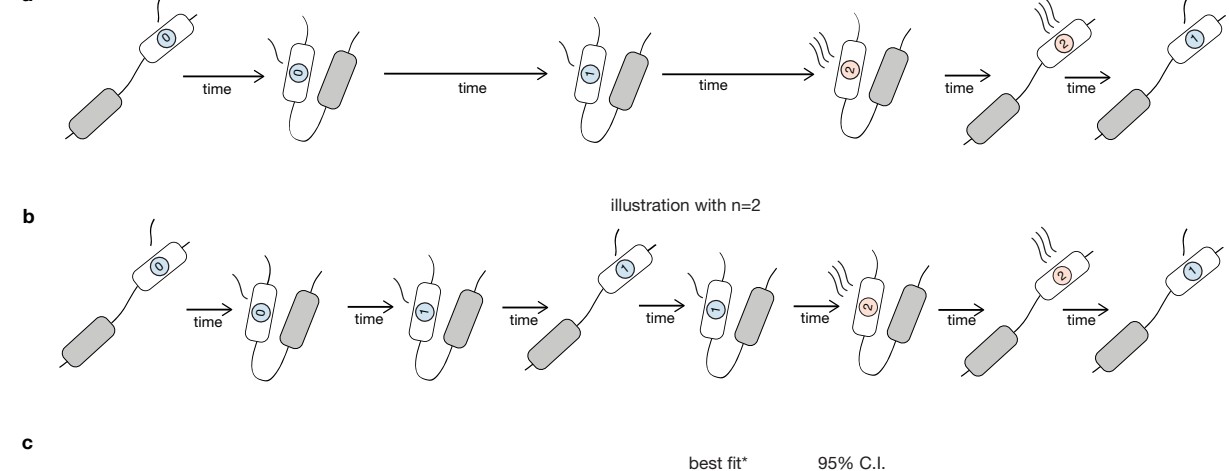

**a** illustration with n=2

**b** illustration with n=2

**c**

Apparent 2-state model

$$ON \xrightarrow{\mu} RNA \xrightarrow{\delta} \varnothing$$

$$k_{on} \uparrow \downarrow k_{off}$$

$$OFF$$

$$k_{on}^{app} = k_{on}^{basal} + \frac{(1-p_c\beta)(p_c\beta)^n}{1-(p_c\beta)^{n+1}}\left(k_{on}^{enh} - k_{on}^{basal}\right)$$

$$\beta = \frac{k_{forward}}{k_{back}}$$

$p_c = $ contact probability

$n = $ # of regulatory steps

| | best fit* | 95% C.I. |
|---|---|---|
| n | 5 | (3 ,7) |
| $k_{on}^{basal}$ | 0.066 | (4e-6 , 0.45) |
| $k_{on}^{enh}$ | 3.66 | (0.4 ,17) |
| $k_{off}$ | 3.51 | (0.32 ,48) |
| $\mu$ | 113.97 | (103 , 575) |
| $\beta$ | 13.62 | (10, 17.5) |
| $\delta$ | 1 | -- |

\* all the rates are in unit of RNA decay rate

**d**

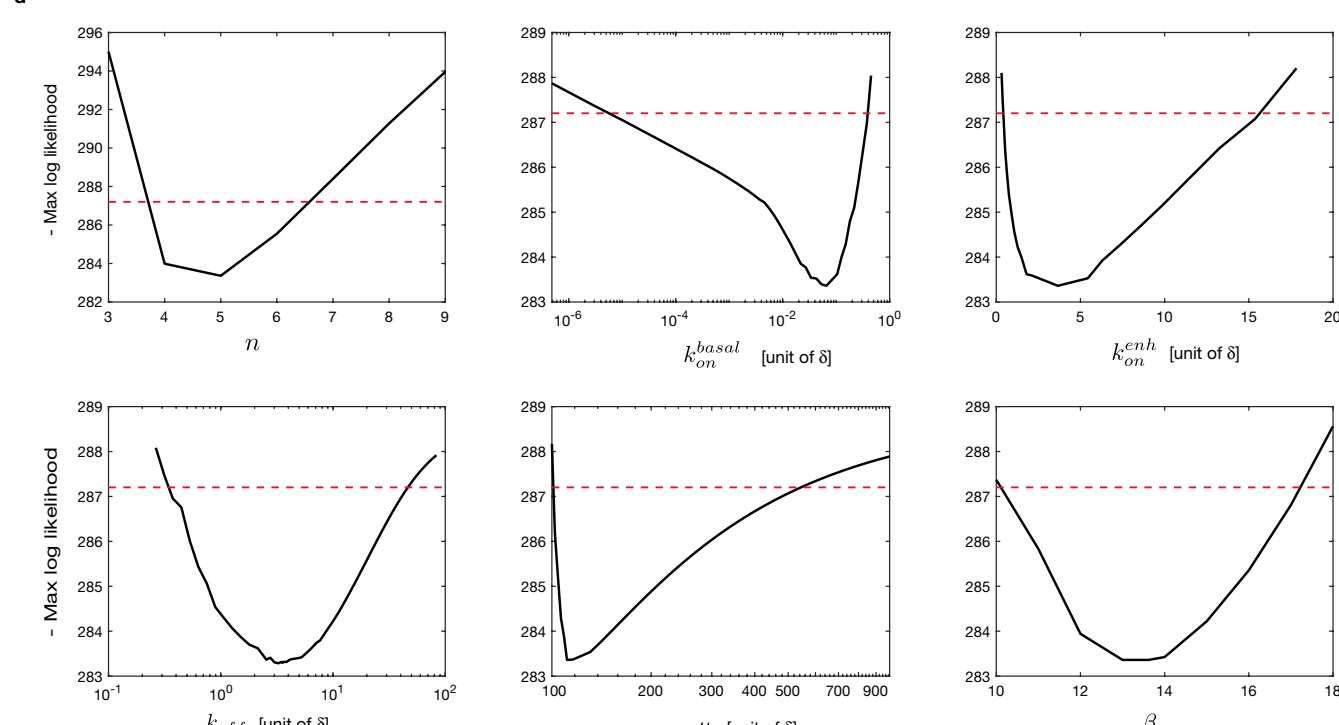

**Extended Data Fig. 4 | Fit of the mechanistic enhancer–promoter model and robustness analysis.** a. Schematic description of the dynamics of the mechanistic model (here with two regulatory steps (*n*=2) for illustration). This case illustrates a scenario where, the enhancer–promoter interaction is long enough to allow the completion of the *2* regulatory steps and transiently increases the promoter activity. b. In an alternative scenario, the interactions are shorter but frequent enough to allow the completion of the *2* regulatory steps and transiently increase the promoter activity. c. Parameter values and 95% confidence intervals for the best fitting apparent two-state model. The rates are in the unit of RNA decay rate (*δ*). d. Profile likelihood functions for all the parameters of the apparent two-state model. Red dashed lines show the threshold used to calculate the 95% confidence intervals (see Supplementary Model description for more details).

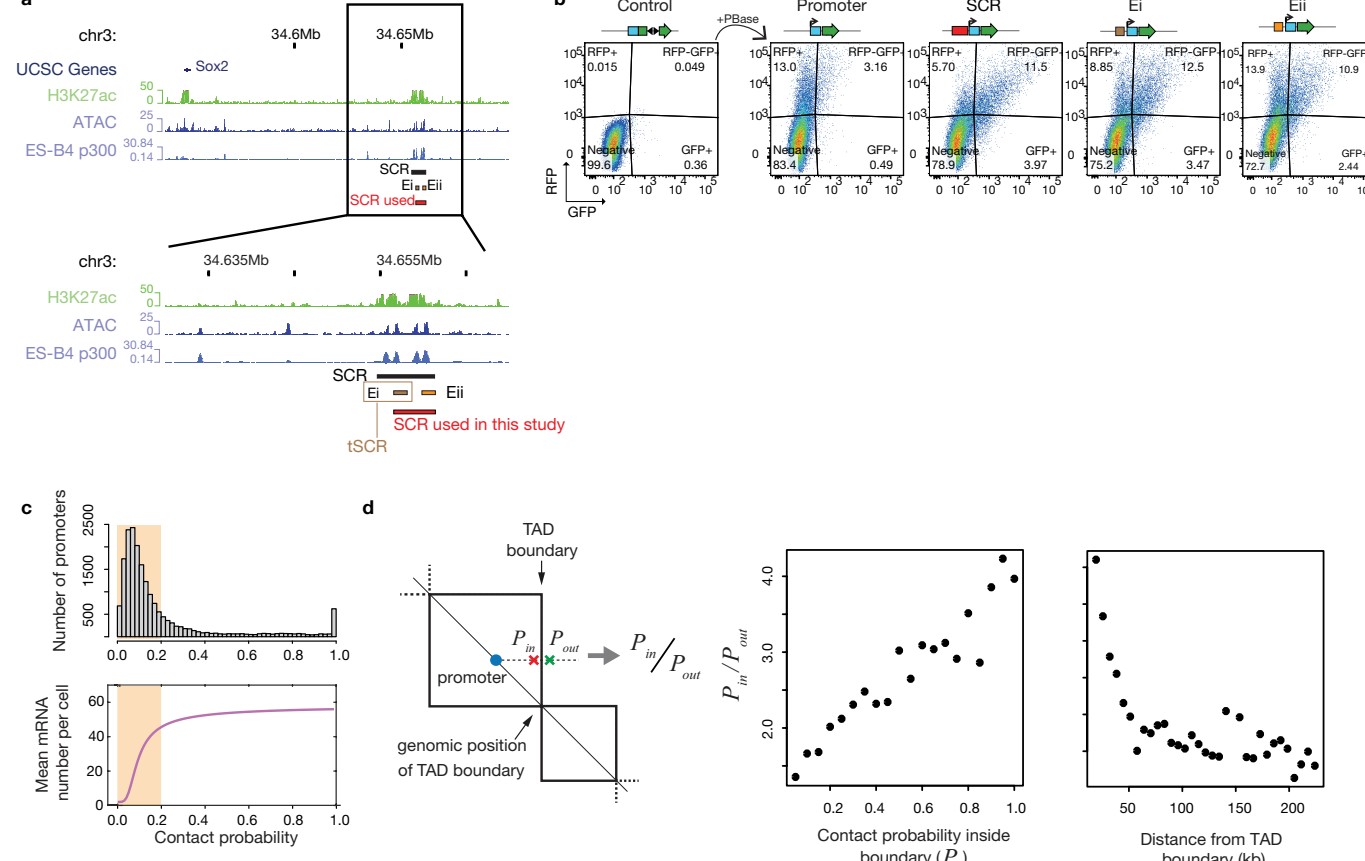

**Extended Data Fig. 5 | Dependence of transcription levels and insulation on enhancer strength.** a. Top: UCSC genome browser snapshot of the endogenous *Sox2* locus and Sox2 control region (SCR). Bottom: close-up view showing the SCR (black) identified in ref.[29] and the enhancer regions used in the transient reporter assays shown in panel **b**. Full-length enhancer is in red (same as in Fig. 1); truncated versions are in brown (Ei) and orange (Eii). Experiments in Fig. 5 were performed with Ei. b. Flow cytometry analysis of mES cells transiently transfected with PBase-RFP and different versions of split eGFP plasmids carry either no enhancer, or the full-length SCR (red, see panel **a**), or the first (brown-Ei) or second (orange-Eii) SCR subregions in front of the Sox2 promoter. Transcription levels generated upon co-transfection with PBase are higher in the presence of the full-length SCR compared to truncated versions. Numbers in each quadrant represent the % of cells either negative or RFP, GFP and RFP-GFP positive. c. Top: distribution of contact probabilities between all active promoters in mES cells and the nearest inner TAD boundaries, calculated as in Extended Data Fig. 2b. Bottom panel: Model prediction for the mean eGFP mRNA numbers per cell plotted against contact probabilities shown as a comparison (same as Fig. 2e). Shaded areas correspond to promoters with contact probability with the closest TAD boundary below 0.2. d. Left panel: scheme of how the probabilities of interaction between promoter and the region before ($P_{in}$) and after the TAD boundary ($P_{out}$) are calculated, same criteria as in Extended Data Fig. 2b. Central panel: promoters with higher contact probabilities with TAD boundaries experience stronger drops of contact probability across boundaries. Right panel: promoters closer to TAD boundaries experience a stronger drop of contact probability across boundaries.

# Reporting Summary

Nature Research wishes to improve the reproducibility of the work that we publish. This form provides structure for consistency and transparency in reporting. For further information on Nature Research policies, see our Editorial Policies and the Editorial Policy Checklist.

## Statistics

For all statistical analyses, confirm that the following items are present in the figure legend, table legend, main text, or Methods section.

| n/a | Confirmed | |
|---|---|---|
| ☐ | ☒ | The exact sample size (*n*) for each experimental group/condition, given as a discrete number and unit of measurement |
| ☐ | ☒ | A statement on whether measurements were taken from distinct samples or whether the same sample was measured repeatedly |
| ☐ | ☒ | The statistical test(s) used AND whether they are one- or two-sided *Only common tests should be described solely by name; describe more complex techniques in the Methods section.* |
| ☐ | ☒ | A description of all covariates tested |
| ☐ | ☒ | A description of any assumptions or corrections, such as tests of normality and adjustment for multiple comparisons |
| ☐ | ☒ | A full description of the statistical parameters including central tendency (e.g. means) or other basic estimates (e.g. regression coefficient) AND variation (e.g. standard deviation) or associated estimates of uncertainty (e.g. confidence intervals) |
| ☐ | ☒ | For null hypothesis testing, the test statistic (e.g. *F*, *t*, *r*) with confidence intervals, effect sizes, degrees of freedom and *P* value noted *Give P values as exact values whenever suitable.* |
| ☒ | ☐ | For Bayesian analysis, information on the choice of priors and Markov chain Monte Carlo settings |
| ☒ | ☐ | For hierarchical and complex designs, identification of the appropriate level for tests and full reporting of outcomes |
| ☒ | ☐ | Estimates of effect sizes (e.g. Cohen's *d*, Pearson's *r*), indicating how they were calculated |

*Our web collection on statistics for biologists contains articles on many of the points above.*

## Software and code

Policy information about availability of computer code

| Data collection | Zeiss Axion Observer Z1 was used for microscopy image collection (RNA FISH), BD LSRII SORP Analyser + HTS was used for acquiring GFP intensity, BD Influx cell sorter was used for the FACS |
|---|---|
| Data analysis | Matlab (version 2019b), global optimisation toolbox (Matlab), symbolic toolbox (Matlab), minimap2 (v. 2.17-r941), Snakemake (v. 3.13.3), IGV (v. 2.9.4), HiC-Pro (v. 2.11.4), ChromHMM (v.1.14), bwa (v. 0.7.17), Biostrings (v. 2.58.0), FlowJo (v. 10.6.2), Knime (3.7.2), Fiji (v. 2.0), TrackMate (v. 6.0.0), Pandas (v. 1.1.0), python 2.7, QuasR 1.34.0, STAR 2.5.0a. Flow Cytometry: BD FACSDiva™ Software. FACS: BD FACS™ Software 1.2.0.142. Custom codes can be found in https://github.com/zhanyinx/Zuin_Roth_2021, https://github.com/gregroth/Zuin_Roth_2021 and https://github.com/vansteensellab/tagmap_hopping/tree/giorgetti |

For manuscripts utilizing custom algorithms or software that are central to the research but not yet described in published literature, software must be made available to editors and reviewers. We strongly encourage code deposition in a community repository (e.g. GitHub). See the Nature Research guidelines for submitting code & software for further information.

## Data

Policy information about availability of data

All manuscripts must include a data availability statement. This statement should provide the following information, where applicable:

- Accession codes, unique identifiers, or web links for publicly available datasets
- A list of figures that have associated raw data
- A description of any restrictions on data availability

All cHi-C, Oxford Nanopore, tagmentation and population-based splinkerette PCR sequencing fastq files generated in this study have been uploaded to the Gene Expression Omnibus (GEO) under accession GSE172257 (https://www.ncbi.nlm.nih.gov/geo/query/acc.cgi?acc=GSE172257). The following public databases were

used: BSgenome.Mmusculus.UCSC.mm9, TxDb.Mmusculus.UCSC.mm9.knownGene.

# Field-specific reporting

Please select the one below that is the best fit for your research. If you are not sure, read the appropriate sections before making your selection.

**☒** Life sciences ☐ Behavioural & social sciences ☐ Ecological, evolutionary & environmental sciences

For a reference copy of the document with all sections, see nature.com/documents/nr-reporting-summary-flat.pdf

# Life sciences study design

All studies must disclose on these points even when the disclosure is negative.

| | |
|---|---|
| Sample size | For cHi-C data we used 2 biological replicates. For RNA FISH we used 3 replicates for each cell line. The flow cytometry measurements were performed in 3 biological replicates. We did not apply statistical methods to pre-determine sample size and followed the general standard practice in the field. Number of replicate experiments is indicated in the legends. |
| Data exclusions | In enhancer mobilisation experiments, enhancer insertions within the transgene itself were omitted as they disrupt the sequence of the transgene and eGFP levels cannot be compared with other enhancer genomic positions. |
| Replication | RNA FISH was performed in triplicates. Flow Cytometry measurements were performed in triplicates. cHi-C was performed in duplicates. Mobilization experiments in ΔΔCTCF background were performed twice. The other mobilisation experiments were performed once. Each mobilisation experiments lead to hundreds different cell lines which can be interpreted as replicates. |
| Randomization | Randomization is not applicable to this study as we used only cell lines and no human or animal subjects were used in this study |
| Blinding | RNA FISH experiments and analysis were performed in a blinded manner. Blinding was not necessary for the other experiments since the results are quantitative and did not require subjective judgment or interpretation. |

# Reporting for specific materials, systems and methods

We require information from authors about some types of materials, experimental systems and methods used in many studies. Here, indicate whether each material, system or method listed is relevant to your study. If you are not sure if a list item applies to your research, read the appropriate section before selecting a response.

## Materials & experimental systems

| n/a | Involved in the study |
|---|---|
| ☒ | ☐ Antibodies |
| ☐ | ☒ Eukaryotic cell lines |
| ☒ | ☐ Palaeontology and archaeology |
| ☒ | ☐ Animals and other organisms |
| ☒ | ☐ Human research participants |
| ☒ | ☐ Clinical data |
| ☒ | ☐ Dual use research of concern |

## Methods

| n/a | Involved in the study |
|---|---|
| ☒ | ☐ ChIP-seq |
| ☐ | ☒ Flow cytometry |
| ☒ | ☐ MRI-based neuroimaging |

# Eukaryotic cell lines

Policy information about cell lines

| | |
|---|---|
| Cell line source(s) | All cell lines are based on E14 mouse embryonic stem cells (mESCs) provided by Edith Heard laboratory, EMBL, Heidelberg |
| Authentication | Cell lines have been recurrently used by the authors in previous studies and therefore have not been authenticated. |
| Mycoplasma contamination | Cells were tested for mycoplasma contamination once a month and no contamination was detected. |
| Commonly misidentified lines (See ICLAC register) | No commonly misidentified lines were used. |

# Flow Cytometry

## Plots

Confirm that:

[✗] The axis labels state the marker and fluorochrome used (e.g. CD4-FITC).

[✗] The axis scales are clearly visible. Include numbers along axes only for bottom left plot of group (a 'group' is an analysis of identical markers).

[✗] All plots are contour plots with outliers or pseudocolor plots.

[✗] A numerical value for number of cells or percentage (with statistics) is provided.

## Methodology

| | |
|---|---|
| Sample preparation | Flow Cytometry: Cells were harvested with Accutase and re-suspend in E14 medium (supplemented with 2i in the case of the remobilization experiment). FACS: cells were harvested with Accutase and re-suspend in E14 medium with only 3% of FCS, 100 μg/m primocin (InvivoGen, ant-pm-1) and 10uM ROCK inhibitor (STEMCELL Technologies, Y-27632). |
| Instrument | Flow Cytometry: BD LSRII SORP Analyser (Becton Dickinson) for transfection efficiency and enhancer reporter assay and BD LSRII SORP Analyser + HTS for enhancer mobilization assay. FACS: BD Influx cell sorter (Becton Dickinson) |
| Software | Flow Cytometry: BD FACSDiva™ Software. FACS: BD FACS™ Software 1.2.0.142 |
| Cell population abundance | Flow Cytometry: 10000 cells were acquired for enhancer mobilization assay and >10000 cells were acquired for transfection efficiency and enhancer reporter assay. FACS: single cells sort was performed by sorting typically six 96-well plates for each founder line. |
| Gating strategy | Flow Cytometry gating: FSC/SSC to discard big cells with high granularity; SCC-W/SCC-H to discard doublets; FSC-A/Dapi to discard dead cells; GFP/histogram to quantify GFP intensity. FACS: FSC/SSC to discard big cells with high granularity; FSC-W/FSC: to discard doublets; SSC-W/SSC: to discard doublets; 530/40[488]/610/20[561]: to discriminate between negative and GFP positive cells. |

[✗] Tick this box to confirm that a figure exemplifying the gating strategy is provided in the Supplementary Information.

