## [Peer Review File · Nature]

Manuscript Title: Nonlinear control of transcription through enhancer-promoter interactions

Reviewer Comments & Author Rebuttals

Reviewer Reports on the Initial Version:

Referees' comments:

Referee #1 (Remarks to the Author):

This study by Zuin, Roth, and colleagues aims to understand how the location of an enhancer influences gene expression. The authors develop a novel genetic system in which they first insert a PiggyBac vector containing a reporter gene and enhancer into a specific location, and then re-mobilize the enhancer to generate a large number of random insertions that are biased toward integrating nearby in the same locus in the genome. Through picking clones, mapping insertion sites, and measuring effects on gene expression, this method provides a means to study how enhancer and promoter positions influence gene regulatory effects. The authors apply this approach to study a Sox2 enhancer and promoter in a locus in mouse embryonic stem cells, profiling hundreds of different insertions. They find that enhancer effects decrease with distance, and that this follows a nonlinear relationship with 3D contacts measured by promoter capture Hi-C. They also show that the coefficient of variation increases with distance, that insertion of a CTCF site results in striking insulation, and that truncation of the enhancer leads to reduced activation. The authors develop a kinetic model to explain these data, which they use to suggest that enhancers act on a large number of rate-limiting steps that help promoters 'memorize' enhancer contacts.

The study represents a heroic experimental and computational effort that I am certain will generate much discussion. I am enthusiastic about the approach and many of the experimental observations, which have the potential to provide (with one caveat described below) an incredibly valuable dataset regarding the influence of distance on enhancer-promoter regulation. The kinetic model for transcription is also thought-provoking, and seems to match certain experimental observations regarding the lack of correlation in single cells between enhancer-promoter contacts and transcription, and enhancers primarily affecting burst frequency rather than burst size. However, the kinetic model and experimental dataset are combined to make conclusions about transcription (e.g., about the number of rate-limiting steps, or promoter memory) that seem to be only one of several possible explanations compatible with their data (see below).

I would be very enthusiastic about the study if the major comments below were addressed:

Major Comments:

1. Accuracy of measurement of GFP levels as a function of distance.

I have one technical question about the experimental measurements. In particular, my understanding of the experimental procedure is that, after the PiggyBac vector was mobilized, cells were sorted based on a gate on GFP expression shown in Fig. 1F, and GFP+ clones isolated. Is there some bias introduced in this strategy, based on selecting clones that are above some threshold of GFP expression? In particular, based on the distributions drawn in Fig 1F, I expect that many insertion-clones that led to weak activation of GFP would lie below the chosen threshold (i.e., similar to the “Promoter-only” condition shifted slightly right). However, if only insertion-clones above the threshold are selected for further measurement, this would artificially increase the mean estimated expression of clones in this expression regime. Could this consideration explain the observed effect where expression is higher than predicted by contact measurements? How many GFP-negative clones were picked and insertions mapped?

To address this, the authors could pick additional clones that lie below the threshold that have insertions in the TAD and measure their expression levels, or sort cells into bins based on expression levels and do some pooled insertion mapping.

2. Consideration of accuracy of 3D contact measurements.

A major part of the narrative rests on the differences between Hi-C contacts and the observed effects on gene expression.

The authors conduct hybrid capture Hi-C to measure changes in 3D contacts after removal of CTCF sites from the region, and, for a few clones, after insertion of the enhancer. Could the authors add a quantification (e.g. barplot) showing the magnitude of the change in Hi-C contacts in the bin linking the enhancer to the gene? It is difficult to tell from the heatmaps in Fig. S2A.

Different 3D contact mapping methods have found differences in the decay as a function of distance (e.g., see SPRITE by Quinodoz et al., Cell, 2018), as well as differences in the strengths of enhancer-promoter contacts (e.g., see Micro-C by Krietenstein et al., Mol. Cell, 2020). Could the authors perform a sensitivity analysis to determine how robust their model and conclusions are to changes in these 2 parameters (decay with distance, and focal signal at enhancer-promoter loci)? Would the magnitude of changes in 3D contact measurements observed in these previous studies affect the authors' conclusions, for example regarding nonlinearity, effect of the CTCF site, or appropriateness of various models for gene regulation?

3. Comparison of alternative models

It is difficult to evaluate how the authors' model performs relative to other, potentially simpler, alternatives. The authors should describe alternative models and compare their performance at predicting the mean and variance of gene expression, and capturing the non-linearity with respect to contact. For example, would a two-state bursting model whose parameters were simple functions of

the contact frequency manage to reproduce the observed nonlinearity?

Relatedly, the marginal increase in R^2 with the $N=4-6$ vs $N=1$ model is used to support the idea that there must exist a large number rate-limiting steps in transcription. However, in Fig. 2B, it is not evident that the data exhibit a sigmoid response. The increase in fit seems to be a result of constraining models to pass through (0,0). In light of the authors' later claims that CTCF sites have additional insulating functions beyond simply modulating contact frequency, another simple model of good fit could be a piecewise function that is identically 0 outside the TAD and inside follows a Langmuir isotherm-type saturation curve (equivalent to their $N=1$ model with an added x-intercept).

In comparing models, the authors could also apply one of the commonly used tests for goodness-of-fit (e.g. chi-squared per degree of freedom, AIC), e.g. going from the $N=1$ model to the $N=4-6$ model.

If the authors' model is not clearly better than simpler models, the authors could improve the paper by toning down conclusions about enhancer function, by thoroughly describing the alternatives, and by describing their current model as "one possible model consistent with the data".

4. Model description

The presentation of the kinetic model was a bit hard to follow, and it took some time to unpack all of the layers. Some of the key details are in figure legends or supplemental text. It could be helpful to state all of the model components in the main text and walk through in slightly more detail the components and their rationale. It would also be helpful to compare where relevant this model to prior theoretical/mechanistic work, and describe which components are based on prior biological knowledge, which were chosen for mathematical convenience, and which were chosen because they fit the data better.

Specific suggestions about model presentation:

Clarify the model components and more simply describe how their model works, in particular the distinctions between promoter "state" and promoter "regime", and the two regime vs. $N+1$ regime models.

The term 'memory', first introduced on page 5, is also confusing and potentially misleading. It would be helpful to define it in the main text (not only in the legend of Fig. 2G).

Minor Comments:

- In addition to the scatterplot comparing activation and contact, the authors could include a plot that shows activation and contact on same graph as function of distance (distance on x-axis, activation and contact overlaid on 2 y-axes)
- In the supplemental text (page 5), I believe the authors confuse the sign of the second derivative at the end points for the sigmoid case (although not in the figures).
- On page 4, I would suggest changing the statement 'These data show that the range of activity of

the SCR enhancer is delimited by TAD boundaries.’ to ‘These data suggest that the range of activity of the SCR enhancer is affected by this/a CTCF site’.

- The authors are able to demonstrate that some of their parameter fits recapitulate certain experimental observations (e.g. the relative time scale of E-P contacts, page 5). Can the authors comment on the extent to which the other parameter values in Fig. S3A might match previous observations? For example, the $k_{\text{forward}}/k_{\text{reverse}}$ rates for the promoter “regimes”: do these parameters make sense biologically (given what we know about bursting kinetics from live-cell imaging experiments)? Similarly, I wonder what molecular mechanisms could in principle explain the strong insulation effect of the deleted CTCF site.
- The authors reference the Alexander et al. study regarding the lack of temporal correlation between enhancer-promoter contacts and transcriptional bursts. In my opinion, the observations in that study are confounded by the fact that the arrays targeting for imaging lie 5-10Kb away from the SCR and Sox2 promoter — a distance that would make it very difficult to know when a contact has actually occurred.

Referee #2 (Remarks to the Author):

In this impressive paper, Zuin, Roth and colleagues use a PiggyBack system to “scatter” an enhancer in the vicinity of a reporter gene promoter to systematically assess the relationship between the position of the enhancer and its effect on gene expression. The main conclusion of the paper is that this relationship is highly dependent on the linear distance (within a TAD, with the TAD boundaries, as expected, providing strong insulation in a single TAD-spanning enhancer translocation event). However, this dependence is non-linear, which argues against a direct relationship between enhancer-promoter contacts and transcriptional bursts. Instead, ODE modelling suggests that these contacts likely switch the promoter from an inactive (lowly-bursting) to a highly-bursting state.

Major comments

1a. Contact probabilities are measured only in the founder cell line. This is reasonable if indeed, no specific loops formed between the translocated enhancer and the promoter. However, I am somewhat surprised that this is the case, since previous *C-based studies did detect SCR-Sox2 loops in the natural context (as discussed in Alexander et al., eLife).

1b. The transcriptional signal is measured at a single-cell level in each cell line, while contact probabilities are measured using a bulk biochemical technique. This assumes that there is little transcriptional and contact heterogeneity within a given clone. While the authors observe no bimodality in transcription, they show that transcriptional heterogeneity does exist and increases with enhancer-promoter distance. Therefore, there is a possibility that at longer distances at least, specific enhancer-promoter loops are formed in a subset of cells, and it is in these cells that these contacts trigger a transcriptional response.

If either 1a or 1b are the case, then the contact probabilities that went into the mathematical model are underestimated, which may affect its conclusions. Could the authors address this possibility - for example, by modelling these scenarios explicitly?

2. The model is based on a single promoter-enhancer pair, and it's not fully clear how unusual this pair is with respect to the relationship between contacts and transcription - particularly if it's true that no specific enhancer-promoter loops are formed. It would be great if the authors could test their model on other loci. The easiest way to do so seems to be taking advantage of the CRISPRi-FISH-flow data from Engreitz lab (Fulco et al., 2019; Nasser et al., 2021).

Minor comments

1. It is not fully clear from the text what is the relationship between the authors' mathematical model and that by Xiao and Boettiger. Were they developed in parallel or one inspired the other? Neither scenario affects the impact of this paper, but it would be good to discuss this model explicitly.

2. The authors claim to have used "capture Hi-C" to detect contact probabilities. However, the method presented is for 3C with capture enrichment; this is "capture C" not "capture Hi-C". Can the authors please clarify in the text which method was used. It would also be helpful to the reader to present the strategy (i.e. tiled capture probes across the locus) in the text on page 4, line 10.

3. The authors deleted a CTCF site within the SCR. Were there any other motifs of note; e.g. binding sites for other factors with known architectural roles?

4. Page 7 line 36 onwards: The source of the genome-wide data presented here and in Suppl. Fig. 5 is not clear.

Referee #3 (Remarks to the Author):

Summary: This paper, 'Nonlinear control of transcription through enhancer-promoter interactions,' sets out to address the important question of how chromosome interactions between enhancers and cognate promoters translate into transcriptional outputs. To this end, this work combined elegant genetic approaches, quantitative single-cell transcriptional measurements and computational modeling to dissect the quantitative relationship between enhancer-promoter contact probability and transcriptional output. With transposon-based mobilization of a genetic reporter, the author derived hundreds of isogenic cell lines with reporters of the same architecture stably integrated in well-mapped genomic positions, something rarely done in the field of chromatin research. This set of cell lines provides a firm foundation for rigorous quantitative measurements, which is leveraged in this work. It also potentially serves as a great resource for the chromatin research community. Using these measurements, the authors uncovered a non-linear dependence of transcriptional output on enhancer-promoter communication frequency. They further provided potential explanations for this nonlinearity with a phenomenological computational model.

Overall, this is an elegant, rigorous and stimulating piece of experimental work that should have substantial impact in the field. By insulating synthetic E-P pairs within a TAD with minimal regulatory and structural complexity, this experimental system is the closest, among many recent works in the field, to decouple the effect of enhancer-promoter interaction probability per se from other potentially confounding variables. Our major concerns revolved around the role of the model in the paper and the strength of the conclusions that one can draw from it.

Major comments:

The role of the modeling initially is to explain the striking ultrasensitive dependence of gene expression on distance and contact frequency. However, it was unclear whether the model represents “a” mechanism of generating ultrasensitivity or “the” mechanism. Because the hypothetical intermediate states do not have specific molecular interpretations, one cannot directly probe their existence and dynamics experimentally. More generally, it wasn’t clear what alternative models should also be considered. Are there other mechanisms that could equally well generate the observed ultrasensitivity?

More generally, what value do we get from the model beyond simply describing the dependence of expression on distance (or contact frequency) as ultrasensitive with a particular Hill coefficient or other parameterization? The authors highlight agreement between the model and experiments in the dependence of CV on expression level and contact frequency (Figure 3). However, it was not at all clear if this is a unique feature of the model or something would observe in any process that regulated expression via modulation of burst frequency. The conclusion on line 31 of page 6 is entirely empirical.

In recent years, numerous models based on multi-step kinetic frameworks have been proposed to account for behavior of transcription bursting, while taking enhancer-promoter communications into considerations (see Lammers et al. 2020) and references therein). The model presented in the current work seems to relate to these published models, frameworks, but the precise relationship was not clearly articulated in the paper.

The conclusion that variability in eGFP is intrinsically generated would require a way to rule out other inputs that would not show up in the flow cytometry gating and cell cycle, e.g. by using a “two color” approach and analyzing correlations between two copies of the same construct in the same cell. Without an approach like this, it is not possible to rule out the many other “extrinsic” variables (e.g. Transcription factor or mediator concentrations) that do not show up in flow but could nevertheless modulate expression.

Because it possesses many steps, the model necessarily includes many free parameters. In the supplement, the authors systematically vary the number of steps and show how it affects the potential ultrasensitivity. However, it was not clear to me how well constrained the parameters are in the fits described in section 8.3 of the supplement. What are the confidence intervals for these parameters?

Along the same lines, how much does the model suffer from overfitting? It was not clear what if

anything constrains the number of the modeling parameters. Take the intermediary step # as an example. The paper claims that 4-11 steps yields the best agreement between the model and experiments. However, this seems to be quite a large range; in fact, according to the supplementary model description, 11 is actually the upper bound of simulation, which is equivalent to stating that any number larger than 4 yields good fit. This makes the number 4-11 of little practical value.

On page 5, line 16-31, and in Figure 2G, the authors discuss the appearance of “memory” (long-lived states) in the model. Again, it was not clear to what extent such long-lived states are unique to this model or might also appear in others. More importantly, there is no direct experimental evidence for these memory states beyond the burst-like expression dynamics, which have been generally observed across many transcriptional systems from prokaryotes to mammals.

In the introduction and abstract, the author emphasizes that one important aspect of the paper concerns the context-dependent dependency of transcriptional output on the E-P interaction probability. However, there seems to be little data focusing on this particular aspect.

Figure 5: The authors show two possible ways, in the model, to modulate the expression-vs-contact relationship (Figure 5A,B), and then show that the experimental perturbation predominantly modulates amplitude (Figure 5D). Is it possible to predict an experimental perturbation that would show the other type of modulation? Such a testable prediction would strengthen the connection between model and experiment.

Minor points:

Figures 2K and 3C: Is there any possible explanation for the apparent periodicity in the simulated results? Does this match experimental measurement? Or does it indicate an overfitting issue of the model?

Figure 2G: is it a simulated result or just a diagram?

Figures 2I, J: These seem to be the same panel as 2B and it appear to be a bit redundant

Figures 3A and C: are these the same data on different y scales?

Figure 3B: might be interesting to see how well the model fits the experimental, maybe via a gamma/negative binomial dist? This might be relevant to the bursting behavior.

Line 1-3 of page 7: the transgene seems to be very close to the rescued CTCF motif, confounded by the limited resolution of Hi-C map (claimed to be 6.4kb), the validity of the claim is questionable.

Line 43-45 of page 7, logic was not clear.

Author Rebuttals to Initial Comments:

We thank the referees for their very constructive and helpful criticism. We have addressed their comments with new experiments and analyses. New and modified sentences in the main text have been marked in red in the revised manuscript.

In summary, these are the main changes:

1. We provide additional mapping of integration sites in clonal cell lines, as well as next-generation sequencing of piggyBac-genome junctions in bulk populations of cells with different eGFP levels. These new experiments validate our choice for the gate used in sorting eGFP+ cells.
2. Motivated by the Reviewer's comments, we present a substantially improved modeling strategy. Briefly: thanks to new smRNA FISH analyses, we found that the Sox2 promoter can be well approximated by an apparent two-state model whose *on* rate depends sigmoidally on contact probabilities (**new Figure 2**). This allows us to constrain the mechanistic model of enhancer-promoter communication to a well-defined parameter regime, and to derive general predictions that do not strictly depend on parameter fitting (**new Figure 3**).
3. We clarify the experimental bases, rationale and operation of our mechanistic model, as well as the predictions it generates. We also clarify that alternative explanations of the experimental data might exist, and cite relevant literature.
4. We provide stronger evidence that the transcriptional response is sigmoidal and does not depend on potential experimental errors and confounding effects.
5. We provide rigorous assessment of parameter robustness by analyzing the profile of maximum likelihood whenever model fitting is required.

Taken together, although the conclusions of the manuscript remain largely unchanged, we feel that our paper has been greatly improved thanks to the Reviewers' insightful suggestions.

Point-by-point responses to reviewers's comments

Referee #1 (Remarks to the Author):

This study by Zuin, Roth, and colleagues aims to understand how the location of an enhancer influences gene expression. The authors develop a novel genetic system in which they first insert a PiggyBac vector containing a reporter gene and enhancer into a specific location, and then re-mobilize the enhancer to generate a large number of random insertions that are biased toward integrating nearby in the same locus in the genome. Through picking clones, mapping insertion sites, and measuring effects on gene expression, this method provides a means to study how enhancer and promoter positions influence gene regulatory effects. The authors apply this approach to study a Sox2 enhancer and promoter in a locus in mouse embryonic stem cells, profiling hundreds of different insertions. They find that enhancer effects decrease with distance, and that this follows a nonlinear relationship with 3D contacts measured by promoter capture Hi-C. They also show that the coefficient of variation increases with distance, that insertion of a CTCF site results in striking insulation, and that truncation of the enhancer leads to reduced activation. The authors develop a kinetic model to explain these data, which they use to suggest that enhancers act on a large number of rate-limiting steps that help promoters 'memorize' enhancer contacts.

The study represents a heroic experimental and computational effort that I am certain will generate much discussion. I am enthusiastic about the approach and many of the experimental observations, which have the potential to provide (with one caveat described below) an incredibly valuable dataset regarding the influence of distance on enhancer-promoter regulation. The kinetic model for transcription is also thought-

provoking, and seems to match certain experimental observations regarding the lack of correlation in single cells between enhancer-promoter contacts and transcription, and enhancers primarily affecting burst frequency rather than burst size. However, the kinetic model and experimental dataset are combined to make conclusions about transcription (e.g., about the number of rate-limiting steps, or promoter memory) that seem to be only one of several possible explanations compatible with their data (see below).

We thank the referee for their enthusiastic appreciation of our work.

I would be very enthusiastic about the study if the major comments below were addressed:

Major Comments:

1. Accuracy of measurement of GFP levels as a function of distance.

I have one technical question about the experimental measurements. In particular, my understanding of the experimental procedure is that, after the PiggyBac vector was mobilized, cells were sorted based on a gate on GFP expression shown in Fig. 1F, and GFP+ clones isolated. Is there some bias introduced in this strategy, based on selecting clones that are above some threshold of GFP expression? In particular, based on the distributions drawn in Fig 1F, I expect that many insertion-clones that led to weak activation of GFP would lie below the chosen threshold (i.e., similar to the “Promoter-only” condition shifted slightly right). However, if only insertion-clones above the threshold are selected for further measurement, this would artificially increase the mean estimated expression of clones in this expression regime. Could this consideration explain the observed effect where expression is higher than predicted by contact measurements? How many GFP-negative clones were picked and insertions mapped?

To address this, the authors could pick additional clones that lie below the threshold that have insertions in the TAD and measure their expression levels, or sort cells into bins based on expression levels and do some pooled insertion mapping.

This is indeed a crucial point, which we were able to address with new experiments. We would first like to draw the attention of the Reviewer to the fact that the vertical line in Fig. 1F does not correspond to the eGFP+ gate, but rather to the average eGFP level of promoter-only cell lines. We apologize for the misunderstanding and hope that FACS gates are now more clearly shown in the **new Suppl. Fig. 1L-M**, in addition to the Supplementary Gating strategy file.

We realize that the initial version of the manuscript did not provide an adequate explanation of how eGFP+ cells were gated. As the revised text related to **Figure 1** hopefully clarifies (page 3-4), the gate we used to define eGFP+ cells was not stringent -- in fact, it was so loose that we routinely found a substantial fraction (~15%) of clones where the piggyBac-enhancer cassette had actually not been mobilised and thus generated ‘negative’ eGFP levels as the founder cell line (see **Fig. 1F**, top panel). Our expectation was thus that we would be able to detect any eGFP fluorescence levels that would exceed this value. Indeed, as reported in **Suppl. Fig. 1L**, we occasionally observed clones (2 out of 264 of those where the enhancer had actually been mobilised) where the piggyBac-enhancer cassette had reinserted outside chr15 and generated, as expected, eGFP levels similar to those of promoter-only control cell lines. Taken together, these facts support the notion that this gate enables the detection of potential additional cell lines with enhancer positions generating very low eGFP expression levels. To address the Reviewer’s question directly, we nevertheless sorted cells using an even less stringent gate on eGFP levels (see **Suppl. Fig. 1L**, reproduced below in Rebuttal Figure 1, bottom panel). Mapping of enhancer integration locations with splinkerette PCR following clonal expansion revealed that of the 51 cell lines generated in this experiment, 50 had a non-mobilized enhancer cassette. In the remaining clone, the

enhancer had reinserted within the original TAD and generated eGFP levels in the range predicted by its genomic distance from the promoter (green point in Rebuttal Figure 2 below). These numbers should be compared to those obtained in a control experiment performed in parallel using the original eGFP gate instead (upper panel in **Suppl. Fig. 1L** and Rebuttal Figure 1 below). Here, in 23/30 cell lines the enhancer had instead reinserted within the original TAD; in one case it had reinserted on a different chromosome; and 6/30 cell lines (20%) contained a non-mobilized enhancer cassette.

Rebuttal Figure 1. Standard (top) and less stringent (bottom) gates on eGFP levels and insertion analysis of corresponding clonal cell lines. Single cells FACS sort was performed 7 days after PBase-IRES-RFP transfection. For this particular experiment, we noticed that in a small fraction of cells, PBase-IRES-RFP was still active, contrary to the other sorts (see for example Rebuttal Figure 3 below).

These results thus confirm our expectation that lowering the eGFP gate only leads to a severe increase in the number of cell lines carrying a non-mobilized enhancer, without enabling the detection of any additional 'low-level' eGFP+ insertions. We note that the cell lines generated in these two additional experiments were now added to the dataset shown in Figure 1H and related analyses in the manuscript.

Rebuttal Figure 2. Additional eGFP+ clonal cell lines identified using the standard (pink) or less stringent (green) eGFP gates indicated in Rebuttal Figure 1. These data were added to Fig. 1H and related analyses in the manuscript.

To further support our choice of the eGFP gate, we additionally performed a tagmentation-based mapping of PiggyBac insertion sites in bulk populations of cells that were sorted either from the original eGFP+ gate (6 pools of 337 sorted cells each), or from two lower eGFP gates (6 pools of 1'0000 sorted cells each) (left panel in **Suppl. Fig. 1M**, reproduced below in Rebuttal Figure 3). In line with single-cell sorting experiments, we found that, after quality control and trimming of reads (see Methods), ~90% of the mapped fragments originate from the non-mobilized enhancer cassette in cells sorted from both lower eGFP gate regions, compared to ~10% in cells sorted from the original eGFP+ gate (center panel).

Rebuttal Figure 3. Left: FACS plot showing the gates used to sort pools of cells for tagmentation-based high-throughput detection of enhancer insertions. For gates “low 1” and “low 2”, six pools of 10000 cells were sorted while for gate “high”, six pools of 337 cells were sorted. Gate “high” corresponds to the standard gate used to isolate eGFP positive cell lines for the mobilization experiments. Right: Barplot showing the fraction of sequencing reads mapping to non-mobilized enhancer cassette determined by Tn5 transposase-mediated high-throughput mapping from the different pools sorted in gates “low 1”, “low 2” and “high”.

Taken together, these new experiments strongly support the notion that the original eGFP gate represented an optimal compromise between efficiently detecting ‘true’ eGFP+ cell lines (including those where eGFP levels are low, e.g. because the SCR reintegrates on other chromosomes) and excluding uninformative, non-mobilised piggyBac-enhancer alleles. They also indicate that our gating strategy did not lead to significant biases in the detection of eGFP levels in single clones. These experiments are now referred to in the main text at page 4 (lines 23-27).

2. Consideration of accuracy of 3D contact measurements.

A major part of the narrative rests on the differences between Hi-C contacts and the observed effects on gene expression.

The authors conduct hybrid capture Hi-C to measure changes in 3D contacts after removal of CTCF sites from the region, and, for a few clones, after insertion of the enhancer. Could the authors add a quantification (e.g. barplot) showing the magnitude of the change in Hi-C contacts in the bin linking the enhancer to the gene? It is difficult to tell from the heatmaps in Fig. S2A.

We now provide a barplot with the relative changes in contact probabilities inferred from cHi-C in **Supplementary Fig. 2A**, confirming that they do not exceed 15% of the ‘wild-type’ contacts in any of the clones that we investigated.

Different 3D contact mapping methods have found differences in the decay as a function of distance (e.g., see SPRITE by Quinodoz et al., Cell, 2018), as well as differences in the strengths of enhancer-promoter contacts (e.g., see Micro-C by Krietenstein et al., Mol. Cell, 2020). Could the authors perform a sensitivity analysis to determine how robust their model and conclusions are to changes in these 2 parameters (decay with distance, and focal signal at enhancer-promoter loci)? Would the magnitude of changes in 3D contact measurements observed in these previous studies affect the authors’ conclusions, for example regarding nonlinearity, effect of the CTCF site, or appropriateness of various models for gene regulation?

We agree with the Reviewer that this is an important point. We have now verified that the detected nonlinear (sigmoidal) character of the transcriptional response is robust to changes in the scaling and absolute magnitude of contact probabilities, which we both varied in a wide range around the experimentally observed values. To provide quantitative support to this analysis, we analyzed the resulting transcription vs. contact probability curves using a simple two-state model whose *on* rate depends on contact probabilities through a Hill function (another insightful suggestion from the Reviewer, which we discuss below in our response to point 3). We found that the numerical value of the Hill coefficient, which is directly related to the shape of the transcriptional response, is well constrained (**Suppl. Figure 3A-B**) and systematically larger than 1, no matter the changes in scaling or magnitude of enhancer-promoter contact frequencies. This is now reported in the new **Suppl. Figure 3C** and main text (page 4) and together with additional tests (see our response to the Reviewers' point 3 below) strongly supports a sigmoidal interpretation of the transcriptional response.

This presents an important result because (as we hope the revised manuscript clarifies) it allowed us to constrain the revised mechanistic model of promoter-enhancer communication to a specific parameter regime (see point 3 below and the new **Figure 3**). This in turn led to new predictions of the mechanistic model that do *not* depend on the fit of the full model, but rather on a theoretical analysis of its general behavior. We are thus confident that these predictions do not depend on potential artefacts in the detection of contact probabilities.

We would finally like to mention that the hypothesis that artifactual changes in contact probabilities could explain the blocking effect of CTCF sites is unlikely. This could only be the case if local enhancer-promoter interactions or scaling properties changed across the CTCF site, a case for which we could not find evidence in our data nor in published Micro-C / SPRITE data.

3. Comparison of alternative models

It is difficult to evaluate how the authors' model performs relative to other, potentially simpler, alternatives. The authors should describe alternative models and compare their performance at predicting the mean and variance of gene expression, and capturing the non-linearity with respect to contact. For example, would a two-state bursting model whose parameters were simple functions of the contact frequency manage to reproduce the observed nonlinearity?

We thank the Reviewer for raising this point, which gave us an opportunity to clarify, refine and improve our model analysis. In response to the Reviewer's request, we found that the data are indeed compatible with a two-state model whose *on* rate (but not *off* or initiation rates) depends on contact probabilities in a sigmoidal manner. This simple model approximates well the average transcriptional response and the single-cell mRNA distributions that we measured using smRNA FISH in a selected panel of cell lines, now reported in **Figure 2**.

We acknowledge that this is an important result, which has become central in the revised version of the manuscript. Indeed although it does not provide mechanistic insight on how enhancer-promoter contacts could actually be translated into transcription levels, this phenomenological 'variable' two-state description provides fundamental constraints on the parameters of the mechanistic model of enhancer-promoter communication. This is now described in detail in the **new Figure 3** and pages 6-7 of the main text. In summary, the fact that the *Sox2* promoter is well approximated by a 'variable' two-state model with a sigmoidal k_{on} implies that in our mechanistic model both enhancer-promoter contacts and the mechanisms that transmit information to the promoter following every contact must be faster than the promoter's *on/off* transitions. This allowed us to derive general conclusions and predictions from the mechanistic model that do *not* rely on fitting the model to the experimental curve anymore, but rather depend on its ability to produce an apparent two-state behavior with sigmoidal k_{on} .

Relatedly, the marginal increase in R^2 with the $N=4-6$ vs $N=1$ model is used to support the idea that there must exist a large number rate-limiting steps in transcription. However, in Fig. 2B, it is not evident that the data exhibit a sigmoid response. The increase in fit seems to be a result of constraining models to pass through (0,0). In light of the authors' later claims that CTCF sites have additional insulating functions beyond simply modulating contact frequency, another simple model of good fit could be a piecewise function that is identically 0 outside the TAD and inside follows a Langmuir isotherm-type saturation curve (equivalent to their $N=1$ model with an added x-intercept).

We agree with the Reviewer on the need to provide further evidence that the transcriptional response is sigmoidal. Instead of fitting the full mechanistic model and evaluating its R^2 , we now characterized the amount of sigmoidality in the experimental data using the phenomenological two-state model with variable *on* rate described in **Figure 2**. This greatly reduced the number of fitting parameters, enabled an analysis of the profile of maximum likelihood and the derivation of 95% confidence intervals. We found that the numerical value of the Hill coefficient for k_{on} , which is related to the sigmoidality of the transcriptional response, was particularly well constrained (new **Suppl. Fig. 3A**). This new rigorous analysis returned a best estimate for the Hill coefficient of 2.8 with a 95% c.i. of 2.4-3.2, corresponding to a sigmoidal transcriptional response.

We nevertheless agree with the reviewer that a stepwise function cannot be excluded *a priori* given the 'extra' insulating effect that CTCF sites might provide at least in some cases. To test this hypothesis, and to simultaneously rule out that the repressive environment outside the CTCF site at the 3' of the TAD also distorts our interpretation of the transcriptional response, we repeated the fit of the variable two-state model but this time excluding from the analysis promoter-only control cell lines, as well as insertions beyond the first reverse CTCF site (see **Rebuttal Figure 4** below).

Rebuttal Figure 4. Highlighting the region excluded from the fit to test the sigmoidality of the transcriptional response in the absence of interference by CTCF sites.

We then fitted the remaining data points and found that the transcriptional response remained sigmoidal (Hill coefficient=2.88, 95% c.i. of 2.25 - 3.9). Thus even inside the TAD and in the absence of extra insulation by CTCF sites or confounding repressive effects, the best fit to the transcriptional response is robustly sigmoidal. This new analysis is provided in the **new Suppl. Fig. 3D-E**.

We however agree with the Reviewer that given the ‘extra’ insulation provided by CTCF, we cannot formally exclude that the role played by sigmoidality in insulation at CTCF-associated TAD boundaries might be only partial. We have accordingly toned down the related statements in the abstract and the manuscript (notably page 7, lines 18-25 and page 10, first paragraph), and hope that this now conveys a more balanced presentation of our results and their implications.

We would finally like to note that we actually did not force the model to pass through the origin of coordinates. When contact probability is zero, the apparent two-state model behaves as a basal two-state model with *on* rate $k_{on}^0 > 0$ and nonzero mRNA production, thus matching observations in promoter-only cell lines (left panel in **Fig. 2D**). We hope that the revised text related to **Figure 2** (page 5) clarifies this point.

In comparing models, the authors could also apply one of the commonly used tests for goodness-of-fit (e.g. chi-squared per degree of freedom, AIC), e.g. going from the N=1 model to the N=4-6 model.

We invite the Reviewer to refer to the previous point for a detailed description of how fits were performed in the revised manuscript. We hope that the new strategy based on evaluating the maximum likelihood profile of the simpler and more robust variable two-state model addresses the Reviewer’s criticism.

If the authors’ model is not clearly better than simpler models, the authors could improve the paper by toning down conclusions about enhancer function, by thoroughly describing the alternatives, and by describing their current model as “one possible model consistent with the data”.

We agree with the Reviewer that alternative explanations of the data cannot be formally excluded, and now acknowledge this fact in the discussion part of the revised manuscript (page 10). We also toned down or deleted statements in the main text and abstract that might have suggested that different interpretations should be ruled out. In the discussion part we additionally refer to alternative models that have been proposed to describe hypersensitive transcriptional responses.

We would nevertheless like to respectfully draw the Reviewer’s attention to the fact that we are not aware of any previously developed model able to explain the entire dataset we generated, namely the mean and cell-to-cell distributions of mRNA numbers as a function of contact probabilities, which to our knowledge had actually never been measured before. For example, the recent ‘futile cycle promoter’ model proposed by the Boettiger lab (Xiao et al., eLife 2021) can generate a sigmoidal dependence of mean transcription levels to contact probabilities through a cooperative mechanism induced e.g. by a biomolecular condensate. However, this model does not account for the fact that without the enhancer, the promoter itself behaves as a two-state model and thus fails to predict cell-to-cell variability.

From a purely formal point of view, any model that would be able to generate a two-state behavior with a k_{on} that depends sigmoidally on contact probabilities would be compatible with our data. Although there might be other model architectures that generate this behavior, we believe that ours is a simple implementation relying on a limited number of realistic hypotheses based on experimental evidence:

- 1) Enhancer-promoter contacts are stochastic in single cells, which is supported by a large number of DNA FISH studies, e.g. Amano et al. Dev. Cell 2009, Giorgetti et al. Cell 2014, Bintu et al. Science 2018 and many others)
- 2) Although evidence in mammals is still lacking (we agree with the Reviewer that the Alexander et al. paper has important caveats), enhancer-promoter contacts might be dynamic and transiently change the transcriptional activity of the promoter, as recently observed in a live-imaging study in *Drosophila* (Chen et al. Nat. Gen. 2018).
- 3) The molecular mechanisms that transmit information from the enhancer to the promoter, whatever their exact nature, involve (probably several) regulatory processes that are inherently stochastic and thus potentially rate-limiting, such as transcription factor binding/unbinding, assembly of regulatory complexes,

or enzymatic reactions required for the onset of transcription. Recent live-cell single-molecule imaging experiments have provided initial insights into the kinetics of such processes and place their timescales in the order of (tens of) seconds (see for example Cho et al., Science 2018, Nguyen et al. Mol. Cell 2021; timescales are also reviewed in Lammers et al., Curr Op Cell Biol 2020).

We hope that the entirely rewritten and remodeled modeling part of the manuscript (see point 4 below) gives a clearer explanation of the rationale of the model, its implementation and its relationship with the data.

4. Model description

The presentation of the kinetic model was a bit hard to follow, and it took some time to unpack all of the layers. Some of the key details are in figure legends or supplemental text. It could be helpful to state all of the model components in the main text and walk through in slightly more detail the components and their rationale. It would also be helpful to compare where relevant this model to prior theoretical/mechanistic work, and describe which components are based on prior biological knowledge, which were chosen for mathematical convenience, and which were chosen because they fit the data better.

Specific suggestions about model presentation:

Clarify the model components and more simply describe how their model works, in particular the distinctions between promoter “state” and promoter “regime”, and the two regime vs. N+1 regime models.

The term ‘memory’, first introduced on page 5, is also confusing and potentially misleading. It would be helpful to define it in the main text (not only in the legend of Fig. 2G).

We thank the reviewer for these suggestions and have extensively revised the modeling part of the manuscript. This allowed us to accommodate the new two-state model with variable k_{on} (**new Figure 2**), and to better explain the rationale, hypothesis and structure of the mechanistic model (**new Figure 3**) (pages 6-7). We have also renamed the ‘low’ and ‘high’ promoter regimes into ‘basal’ and ‘enhanced’, which we hope are more suggestive of the processes they describe. We hope that the descriptions of the (now two) models are easier to follow and more balanced. We have also removed references to transcriptional ‘memory’, which as also pointed out by Reviewer #3 could indeed be misleading. We replaced this part with a more extensive description of model dynamics and timescale decoupling in the range of parameters where the transcriptional response is sigmoidal (see **Figure 3F** and related text in page 7).

Minor Comments:

- In addition to the scatterplot comparing activation and contact, the authors could include a plot that shows activation and contact on same graph as function of distance (distance on x-axis, activation and contact overlaid on 2 y-axes)

This graph was actually already provided (**Suppl. Fig. 2D**). It can be found in **Suppl. Fig. 2D** in the revised manuscript.

- In the supplemental text (page 5), I believe the authors confuse the sign of the second derivative at the end points for the sigmoid case (although not in the figures).

Thank you. Yes, the signs were inverted. We note however that this part of the supplementary text has been removed in our revised analysis of the model.

- On page 4, I would suggest changing the statement ‘These data show that the range of activity of the SCR enhancer is delimited by TAD boundaries.’ to ‘These data suggest that the range of activity of the SCR enhancer is affected by this/a CTCF site’.

We thank the Reviewer for pointing at this inconsistency. Thanks to the additional experiment described in our response to their point 1 above, we now isolated several more eGFP+ clones from the original gate (see also Rebuttal Figure 2), including some with insertions between the CTCF site the Reviewer refers to and the 3’ end of the TAD. Interestingly, either because of the insulating effect of the CTCF site, or because of the predominantly repressive chromatin state in this region, these clones have relatively low expression levels. This further confirms our ability to detect even lowly-expressing clones using the original eGFP gate. Unless the Reviewer feels strongly that this sentence should be removed, we would thus prefer to keep it in the manuscript.

- The authors are able to demonstrate that some of their parameter fits recapitulate certain experimental observations (e.g. the relative time scale of E-P contacts, page 5). Can the authors comment on the extent to which the other parameter values in Fig. S3A might match previous observations? For example, the $k_{\text{forward}}/k_{\text{reverse}}$ rates for the promoter “regimes”: do these parameters make sense biologically (given what we know about bursting kinetics from live-cell imaging experiments)? Similarly, I wonder what molecular mechanisms could in principle explain the strong insulation effect of the deleted CTCF site.

We have inserted a paragraph at the end of the results section (page 7, line 26 on) to explain that model fitting to the experimental data resulted in promoter bursting kinetics that are in line with previous experimental observations (in the minutes to tens of minutes range, values are reported in the new **Suppl. Fig. 4B**). Based on the requirement that $k_{\text{close, far}} \gg k_{\text{forward, back}} \gg k_{\text{on}}^{\text{basal, enh}}$, k_{off} , μ , we predict enhancer-promoter interactions to be in the range of seconds (or less), which will be exciting to test in future live-cell experiments. Rates through intermediate steps should therefore be in an intermediate scale - possibly in the range of seconds to tens of seconds, in line with recent estimations of assembly times of pre-initiation complexes in vivo (Nguyen et al Mol Cell 2021).

It is difficult to speculate on the mechanisms underlying the insulation effect of CTCF based on the data we collected. One intriguing possibility is that while actual contact probabilities do not change across a CTCF site, the kinetics of enhancer-promoter contacts do. In other words, the *far* and *close* rates might change individually without changing the contact probability, which is given by the ratio $k_{\text{close}}/(k_{\text{close}} + k_{\text{far}})$. This could possibly emerge as an effect of CTCF providing a barrier to loop extrusion by cohesin, which could introduce asymmetries in the dynamics of chromatin loops. It will be extremely interesting to address this question in the future using live-cell imaging approaches, but we trust that the Reviewer agrees that this goes well beyond the scope of this study.

- The authors reference the Alexander et al. study regarding the lack of temporal correlation between enhancer-promoter contacts and transcriptional bursts. In my opinion, the observations in that study are confounded by the fact that the arrays targeting for imaging lie 5-10Kb away from the SCR and Sox2 promoter — a distance that would make it very difficult to know when a contact has actually occurred.

We have nuanced the comparison with Alexander et al. in the main text as follows: “The model also predicts that because of the decoupling of their timescales (**Fig. 3G**), enhancer-promoter contacts should be uncorrelated to transcription bursts as recently suggested by a study where locations flanking the endogenous Sox2 gene and SCR were imaged simultaneously with Sox2 nascent RNA” (page 7).

Referee #2 (Remarks to the Author):

In this impressive paper, Zuin, Roth and colleagues use a PiggyBack system to “scatter” an enhancer in the vicinity of a reporter gene promoter to systematically assess the relationship between the position of the enhancer and its effect on gene expression. The main conclusion of the paper is that this relationship is highly dependent on the linear distance (within a TAD, with the TAD boundaries, as expected, providing strong insulation in a single TAD-spanning enhancer translocation event). However, this dependence is non-linear, which argues against a direct relationship between enhancer-promoter contacts and transcriptional bursts. Instead, ODE modelling suggests that these contacts likely switch the promoter from an inactive (lowly-bursting) to a highly-bursting state.

We thank the Reviewer for their appreciation of our manuscript.

We would like to point out that the model does not rely on ODEs but is rather fully stochastic. The mean, the variance and the distribution of the number of mRNA molecules per cell at steady-state were calculated numerically as well as analytically for specific limit cases as described in detail in the Supplemental Model Description.

Major comments

1a. Contact probabilities are measured only in the founder cell line. This is reasonable if indeed, no specific loops formed between the translocated enhancer and the promoter. However, I am somewhat surprised that this is the case, since previous *C-based studies did detect SCR-Sox2 loops in the natural context (as discussed in Alexander et al., eLife).

We actually did measure contact probabilities in a panel of clonal lines where the SCR had been mobilised in different positions around the promoter (**Suppl. Fig. 2A**). We hope that the new histogram provided in **Suppl. Fig. 2A** clarifies that minor changes in contact frequencies can indeed be detected in these clones. However, changes are systematically smaller than 15% with respect to cell lines where the enhancer has not been mobilised. Even accounting for the confounding effect of the second, wild-type allele in capture Hi-C experiments, actual changes in contact probabilities introduced by ‘focal’ SCR-promoter interactions should not be higher than 30%.

Interestingly, as now reported in **Suppl. Fig. 3C**, the sigmoidal character of the transcriptional response is robust to even much larger changes in contact probabilities (see also our response to Reviewer #1’s point 1 above). Since the general predictions of the mechanistic model (now shown in **Figure 3**) only depend on the generic sigmoidal behavior of the transcriptional response, rather than on parameter fitting to a specific experimental curve, small changes in contact probabilities such as those that are introduced by mobilizing the enhancer are not expected to significantly impact the any conclusions of our work.

1b. The transcriptional signal is measured at a single-cell level in each cell line, while contact probabilities are measured using a bulk biochemical technique. This assumes that there is little transcriptional and contact heterogeneity within a given clone. While the authors observe no bimodality in transcription, they show that transcriptional heterogeneity does exist and increases with enhancer-promoter distance. Therefore, there is a possibility that at longer distances at least, specific enhancer-promoter loops are formed in a subset of cells, and it is in these cells that these contacts trigger a transcriptional response.

We fully agree with the Reviewer that cell-to-cell and temporal variability in promoter-enhancer interactions are a central feature and in fact, our stochastic model is entirely based on this notion. We apologize if this did not emerge clearly from the previous version of the manuscript. We hope that the entirely rewritten modeling section and new **Figure 3** will now give a better account of how the model works.

In summary, contact probability in the model is defined as the population-averaged fraction of 'cells' where the enhancer and promoter are physically 'close' at any given moment in time (**Fig. 3A**). When the *genomic* distance between the enhancer and the promoter is small, their contact probabilities are high - which means that at any given time, the enhancer is physically close to the promoter in many cells. Conversely when genomic distances are longer and contact probabilities lower, the enhancer is close to the promoter in fewer cells. No matter the actual value of the contact probability, enhancer-promoter contacts within a cell are dynamic in time. Anytime the enhancer is close in a single cell it can contribute to transcriptional activation by transiently increasing the *on* rate of the promoter by 'pushing' it through the intermediate regulatory steps, as we hope it should now be clearer from the new **Figure 3** and related text in pages 7-8.

If either 1a or 1b are the case, then the contact probabilities that went into the mathematical model are underestimated, which may affect its conclusions. Could the authors address this possibility - for example, by modelling these scenarios explicitly?

We hope based on the previous responses that the Reviewer agrees that we have addressed both scenarios (1a in the new **Suppl. Figure 3C** and 1b directly in the model).

2. The model is based on a single promoter-enhancer pair, and it's not fully clear how unusual this pair is with respect to the relationship between contacts and transcription - particularly if it's true that no specific enhancer-promoter loops are formed. It would be great if the authors could test their model on other loci. The easiest way to do so seems to be taking advantage of the CRISPRi-FISH-flow data from Engreitz lab (Fulco et al., 2019; Nasser et al., 2021).

We agree with the Reviewer that it would be extremely interesting to test if the nonlinear relationship we measured for the *Sox2* enhancer-promoter pair holds true for other regulatory pairs. The only formally equivalent manner to do so would be to replace the SCR and *Sox2* promoter with additional enhancer-promoter pairs in our assay, and measure their communication in the same 'neutral' chromatin environment in the absence of confounding effects. While we are definitely keen to develop this in the future, we trust that the Reviewer agrees that due to the complexity of such multiplexed experiments (which would have to rely on some kind of library delivery and barcode readout), this exceeds the scope of the current study. We have nonetheless tested if the sigmoidal transcriptional response we detected for *Sox2* can predict the CRISPRi data from Fulco et al. 2019, as suggested by the Reviewer. This is based on the following assumptions:

- 1) The transcriptional response of any regulatory pair has the same shape as the one we measured for the *Sox2*-SCR pair;
- 2) As in Fulco et al., the contribution of each regulatory element is independent from the others in each considered 5-Mb genomic region;
- 3) The fraction of regulatory input due to a single regulatory element is given by a modified 'ABC' score, which we indicate here has 'ABfC', where instead of being multiplied by normalised Hi-C contact frequencies, the 'activity' (A) score (which in Fulco et al. is proportional to the element's amount of H3K27ac and chromatin accessibility) is multiplied by the sigmoidal function of contact probabilities we observed for *Sox2* (normalized to 1).

Under this set of hypotheses, we found that our model performs similarly to the simpler ABC model proposed by Fulco et al., in which contact probabilities contribute linearly to transcription levels. Both the correlation with CRISPRi transcriptional changes and precision/recall curves are on par with those generated by the ABC model (see Rebuttal Figure 5 below).

Unless the Reviewer argues that these data should be shown in the revised version of the manuscript, we would prefer not to include them because the assumptions they rely on are arbitrary and cannot be tested rigorously. This is at odds with the strategy presented in our paper, where we minimized regulatory and

structural complexity in an engineered genomic region in order to extract quantitative information and ensure that perturbations are meaningful and interpretable.

Rebuttal Figure 5. Left and center panels: comparison of the ability of ABC and ABfC scores to predict observed changes in gene expression following CRISPRi of individual regulatory elements (see Fulco et al. Figure 3). Each dot represents one tested regulatory element / promoter pair. Red dots: connections for which perturbation resulted in a significant decrease in the expression of the tested gene. Blue dots: connections for which perturbation resulted in a significant increase in the expression of the tested gene or in no significant effect. Right panel: precision-recall plot for classifiers of regulatory element / promoter pairs. As in Fulco et al., positive pairs are those for which perturbation of the distal element significantly decreases expression of the gene.

Minor comments

1. It is not fully clear from the text what is the relationship between the authors' mathematical model and that by Xiao and Boettiger. Were they developed in parallel or one inspired the other? Neither scenario affects the impact of this paper, but it would be good to discuss this model explicitly.

The two models were developed independently. We openly discussed the results with the Boettiger lab and evaluated possible overlap, compatibility and co-submission options. We eventually consensually decided in favor of two separate submissions.

We now explicitly discuss alternative explanations of our data and refer to Xiao et al. in the discussion section (page 10). However we would like to point out that we are not aware of any previously developed models, including Xiao et al., which can account for the whole range of experimental results provided in our study, namely both the population-averaged (mean FACS eGFP values) and single-cell behavior (mRNA distributions) of transcription as a function of hundreds of enhancer positions.

2. The authors claim to have used "capture Hi-C" to detect contact probabilities. However, the method presented is for 3C with capture enrichment; this is "capture C" not "capture Hi-C". Can the authors please clarify in the text which method was used. It would also be helpful to the reader to present the strategy (i.e. tiled capture probes across the locus) in the text on page 4, line 10.

We apologize for the confusion. We refer to the method now as 'capture-C' and explain the tiling strategy in the results section: "as revealed by capture-C with tiled oligonucleotides spanning 2.9 megabases around the transgene insertion site".

3. The authors deleted a CTCF site within the SCR. Were there any other motifs of note; e.g. binding sites for other factors with known architectural roles?

We performed motif enrichment analysis in the surrounding sequence using CentriMo from Meme Suite (Bailey and Machanick, "Inferring direct DNA binding from ChIP-seq", *Nucleic Acids Research*, 40:e128, 2012). We did not detect any known additional motifs within the deleted sequence of 18 bp containing the CTCF motif.

4. Page 7 line 36 onwards: The source of the genome-wide data presented here and in Suppl. Fig. 5 is not clear.

We apologize for the missing information. We used the TxDb.Mmusculus.UCSC.mm9.knownGene database and in-house RNA-seq in wild-type mESC to identify active promoters. We now uploaded the RNA-seq data to the GEO repository. Detection of TAD boundaries as well as estimation of contact probabilities were performed using our Hi-C experiments in the same mESC line (Redolfi et al., *Nat Struct Mol Biol* 2019).

Referee #3 (Remarks to the Author):

Summary: This paper, 'Nonlinear control of transcription through enhancer-promoter interactions,' sets out to address the important question of how chromosome interactions between enhancers and cognate promoters translate into transcriptional outputs. To this end, this work combined elegant genetic approaches, quantitative single-cell transcriptional measurements and computational modeling to dissect the quantitative relationship between enhancer-promoter contact probability and transcriptional output. With transposon-based mobilization of a genetic reporter, the author derived hundreds of isogenic cell lines with reporters of the same architecture stably integrated in well-mapped genomic positions, something rarely done in the field of chromatin research. This set of cell lines provides a firm foundation for rigorous quantitative measurements, which is leveraged in this work. It also potentially serves as a great resource for the chromatin research community. Using these measurements, the authors uncovered a non-linear dependence of transcriptional output on enhancer-promoter communication frequency. They further provided potential explanations for this nonlinearity with a phenomenological computational model.

Overall, this is an elegant, rigorous and stimulating piece of experimental work that should have substantial impact in the field. By insulating synthetic E-P pairs within a TAD with minimal regulatory and structural complexity, this experimental system is the closest, among many recent works in the field, to decouple the effect of enhancer-promoter interaction probability per se from other potentially confounding variables. Our major concerns revolved around the role of the model in the paper and the strength of the conclusions that one can draw from it.

We thank the Reviewer for their appreciation of our manuscript and their careful analysis of our modeling results. Together with the issues raised by Reviewer #1, their insightful comments prompted us to substantially refine and rewrite our modeling strategy.

In summary: analysis of smRNA FISH data (now provided in **Fig. 2D**) revealed that the steady-state behavior of the ectopic Sox2 promoter can be approximated by an effective two-state model whose *on* rate depends sigmoidally on contact probabilities (now presented in **Figure 2**). Although this does not provide mechanistic insight on how enhancer-promoter contacts could actually be translated into transcription levels, it nevertheless provides fundamental constraints on the mechanistic model of enhancer-promoter communication. Indeed as now described in **Figure 3** and in the revised Supplementary Model Description, the mechanistic enhancer-promoter model reduces to the observed two-state behavior with sigmoidal *on*

rate when enhancer-promoter interactions are faster than intermediate regulatory processes, which are themselves faster than promoter bursting dynamics. Importantly, this is not anymore based on fitting the full model to the experimental data, but rather on an analysis of the model's general behavior. These findings address many of the Reviewer's comments and, in our view, substantially improve the modeling part of the manuscript.

Major comments:

The role of the modeling initially is to explain the striking ultrasensitive dependence of gene expression on distance and contact frequency. However, it was unclear whether the model represents “a” mechanism of generating ultrasensitivity or “the” mechanism. Because the hypothetical intermediate states do not have specific molecular interpretations, one cannot directly probe their existence and dynamics experimentally. More generally, it wasn't clear what alternative models should also be considered. Are there other mechanisms that could equally well generate the observed ultrasensitivity?

This indeed is a very important question, also raised by Reviewer #1. We would like to mention first that we are not aware of any available models that can account for the whole range of experimental results provided by our study (namely the dependence of the mean and cell-to-cell distribution of expression levels on contact probabilities with a distal enhancer, as a function of its contact probability with the promoter). From a formal point of view however, any mechanistic model where the interplay between interactions and promoter operation results in an effective two-state behavior with apparent k_{on} depending sigmoidally on contact probabilities (as now presented in **Fig. 2**) would describe our data equally well. It is certainly possible to imagine other model architectures that would generate the required behavior. However we believe that the one presented in the manuscript is a simple implementation relying on a limited number of realistic hypotheses:

1) Enhancer-promoter contacts are stochastic in single cells. This is supported by a large number of DNA FISH studies, e.g. Amano et al. Dev. Cell 2009, Giorgetti et al. Cell 2014, Bintu et al. Science 2018 and many others).

2) Although evidence in mammals is still lacking, enhancer-promoter contacts might be dynamic events that transiently change the transcriptional activity of the promoter, as recently observed in live-imaging study in *Drosophila* (Chen et al. Nat. Gen. 2018).

3) The molecular mechanisms that transmit information from the enhancer to the promoter, whatever their exact nature, involve (probably more than one) regulatory processes that are inherently stochastic and thus potentially rate-limiting, such as transcription factor binding/unbinding, assembly of regulatory complexes, and enzymatic reactions required for the establishment of productive transcription. Recent live-cell single-molecule imaging experiments have provided initial insights into the kinetics of such processes and place their timescales in the order of (tens of) seconds (see e.g. Mediator/PoIII dynamics in Cho et al., Science 2018; TF binding/unbinding in Chen et al., Cell 2014 and others; PIC assembly/disassembly in Nguyen et al. Mol. Cell 2021; timescales are also reviewed in Lammers et al., Curr Op Cell Biol 2020). We believe that the ‘intermediate steps’ in our model, which a simplified and more robust fitting procedure now estimates to be in a range of 4 to 7 (based on 95% confidence intervals, see below) represent a reasonable abstract representation of such stochastic regulatory processes.

We however agree with the Reviewer that alternative schemes cannot be ruled out *a priori* and have now toned down or removed any statement in the main text and abstract that might previously have suggested that any alternatives should be excluded. We have also discussed (see page 10) potential different explanations (such as the ‘futile cycle promoter’ model proposed in Xiao et al., eLife 2021), although they would require substantial modification to be tested against our experimental dataset (e.g. they should incorporate the intrinsic two-state dynamics we observed in the absence of an enhancer, see **Fig. 2D**).

More generally, what value do we get from the model beyond simply describing the dependence of expression on distance (or contact frequency) as ultrasensitive with a particular Hill coefficient or other parameterization? The authors highlight agreement between the model and experiments in the dependence of CV on expression level and contact frequency (Figure 3). However, it was not at all clear if this is a unique feature of the model or something would observe in any process that regulated expression via modulation of burst frequency. The conclusion on line 31 of page 6 is entirely empirical.

We agree that our presentation of the dependence of protein CVs on expression levels could be misleading. The Reviewer is correct in that any process modulating burst frequency would lead to the same result. In order to provide a more balanced presentation of these data and reduce the impression that they constitute a validation of the model, we now present them in **Suppl. Fig. 1N-O** as part of the empirical findings context of enhancer mobilisation experiments (page 4). To better characterize cell-to-cell variability in transcription levels we now provide additional smRNA FISH quantifications in a panel of cell lines (**Fig. 2D**), which indeed are compatible with a modulation of k_{on} and thus of burst frequency.

We hope that the Reviewer agrees that our model:

- 1) provides a possible mechanistic explanation of how interactions with an enhancer determine transcription levels, based on a minimal set of general and plausible hypotheses;
- 2) makes the experimentally testable prediction that the dynamics of enhancer-promoter interactions should be much faster than the bursting kinetics of the promoter (and should thus be on the timescale of seconds or less). Importantly, this prediction does not rely on parameter fitting but rather on reducing the full mechanistic model to an apparent two-state regime (see **Figure 3**). Equally important, the same experimental constraint on model parameters places the n regulatory steps in our model in an intermediate timescale of (tens of) seconds, in line with estimated rates of regulatory processes at the enhancer/promoter interface.

In recent years, numerous models based on multi-step kinetic frameworks have been proposed to account for behavior of transcription bursting, while taking enhancer-promoter communications into considerations (see Lammers et al. 2020) and references therein). The model presented in the current work seems to relate to these published models, frameworks, but the precise relationship was not clearly articulated in the paper.

We apologize for the lack of references to previous theoretical literature. We thank the Reviewer for pointing us to the very insightful Lammers et al., Curr Op Cell Biol 2020 review, which we now refer to in the modeling part of the revised manuscript (page 6: "We reasoned that the interplay between the timescales over which all these stochastic processes occur might generate nonlinear effects, as previously hypothesized to explain the intrinsic bursty activity of a promoter (Ref. 46)"). Indeed one of the hypotheses presented there (namely that rate-limiting steps in transcription factor binding can generate nonlinear effects and bursty promoter behavior) is very much in line with - although formally not equivalent to - the mechanism we propose to transform interactions in transcription levels.

In Lammers et al., rate-limiting steps describe transitions between favorable and unfavorable transcription factor binding states within an enhancer located genomically close to the core promoter (rather than a distal enhancer looping on the promoter). The resulting nonlinearity leads to the appearance of bursting behavior at the promoter itself. In our model, these 'microscopic' processes operating at the promoter are not described explicitly but rather implicitly integrated in the promoter's two-state on/off dynamics, which occurs both in its basal and in enhanced regimes (**Fig. 3C**). Rate-limiting regulatory steps instead operate upstream of the two-state promoter and mediate the transmission of regulatory information following enhancer looping events.

We now also more extensively refer to another recently proposed model (the ‘futile cycle promoter’ model of Xiao et al., eLife 2021) which can produce sigmoidal transcriptional responses to contact probabilities.

The conclusion that variability in eGFP is intrinsically generated would require a way to rule out other inputs that would not show up in the flow cytometry gating and cell cycle, e.g. by using a “two color” approach and analyzing correlations between two copies of the same construct in the same cell. Without an approach like this, it is not possible to rule out the many other “extrinsic” variables (e.g. Transcription factor or mediator concentrations) that do not show up in flow but could nevertheless modulate expression.

We thank the Reviewer for this comment and agree that in the absence of additional analysis of extrinsic vs. intrinsic fluctuations it is not possible to identify further sources of extrinsic variability. To provide a more balanced presentation of cell-to-cell variability in our study, we removed this analysis from the manuscript and now present the data on protein CV as an empirical finding in the context of enhancer mobilisation experiments (**Suppl. Fig. 1N-O**). We also provide smRNA FISH data in a panel of cell lines (**Fig. 2D**), which indeed supports the notion that contacts with the SCR modulate burst frequency as shown in the new **Figure 2**.

Because it possesses many steps, the model necessarily includes many free parameters. In the supplement, the authors systematically vary the number of steps and show how it affects the potential ultrasensitivity. However, it was not clear to me how well constrained the parameters are in the fits described in section 8.3 of the supplement. What are the confidence intervals for these parameters?

In the revised model we were able to reduce the number of parameters thanks to the discovery that the promoter behaves as an effective two-state model with a k_{on} that sigmoidally depends on contact probabilities. This enabled to constrain the mechanistic model to the region in the parameter space where it produces such behavior (**Figure 3F-H**) and to fit the asymptotic behavior of the model rather than its full version, reducing the number of parameters and enabling us to calculate confidence intervals for each of them (now provided in **Suppl. Fig. 4A-B**). In this new context the model’s general predictions, notably that there are more than one regulatory step and that enhancer-promoter contacts should be much faster than promoter bursting kinetics, do not depend on fitting but rather on a qualitative study of the parameters and on the fit of the variable two-state model.

Along the same lines, how much does the model suffer from overfitting? It was not clear what if anything constrains the number of the modeling parameters. Take the intermediary step # as an example. The paper claims that 4-11 steps yields the best agreement between the model and experiments. However, this seems to be quite a large range; in fact, according to the supplementary model description, 11 is actually the upper bound of simulation, which is equivalent to stating that any number larger than 4 yields good fit. This makes the number 4-11 of little practical value.

The Reviewer raises an important and valid point. As explained in the previous response, we have now fitted a reduced version of the model with a reduced number of parameters and assessed 95% confidence intervals for each of them. This new approach also allowed us to fit the six smRNA FISH distributions in **Fig. 2D** simultaneously with the mean transcriptional response, which contributed to further constraining the parameters. The number of intermediate steps is particularly well defined and is estimated at five intermediate steps (95% confidence interval: 4-7) (**Suppl. Fig. 4A-B**). As also mentioned in the response to the previous point, the qualitative prediction that there are more than one regulatory step does not depend on the fit but rather on the parameter study and on the fit of the variable two-state model.

On page 5, line 16-31, and in Figure 2G, the authors discuss the appearance of “memory” (long-lived states) in the model. Again, it was not clear to what extent such long-lived states are unique to this model or might also appear in others. More importantly, there is no direct experimental evidence for these memory states beyond the burst-like expression dynamics, which have been generally observed across many transcriptional systems from prokaryotes to mammals.

We realize that our previous description of ‘memory’ might have generated some confusion. We thus decided to de-emphasize it in the revised manuscript, focusing instead on a hopefully clearer discussion that for the mechanistic model to produce the behavior observed in our experiments, the timescales of bursting and of promoter-enhancer interactions must be uncoupled (page 7). We agree with the Reviewer that there is currently no experimental evidence for this and in fact, we hope the new version of the manuscript clarifies how this is a general prediction of our model, which will be exciting to test in future live-cell experiments.

In the introduction and abstract, the author emphasizes that one important aspect of the paper concerns the context-dependent dependency of transcriptional output on the E-P interaction probability. However, there seems to be little data focusing on this particular aspect.

We believe that our own data provide a clear illustration of the widespread and often cited intuition in the field that ‘every locus is different’. First, as shown in **Suppl. Fig. 1K**, even the small deviations from chromatin state neutrality occurring within the extremely simplified genomic region we considered correlate with increased or decreased communication with the promoter, if the enhancer is inserted in neighboring sequences. Second, **Figure 4** clearly shows that the presence of a single CTCF site can have a very strong effect on enhancer-promoter communication. Third, as indicated by experiments with the truncated SCR (**Figure 5**), different (although in this case related) enhancers elicit different transcription levels from the same genomic distance. Together this argues that in ectopic, gene-dense and structurally complex locations it would be extremely difficult to detect general trends linking distance or contact probabilities to enhancer-promoter communication, especially considering that it is unclear if different enhancers act additively, cooperatively or exclusively on single promoters, and vice versa.

Figure 5: The authors show two possible ways, in the model, to modulate the expression-vs-contact relationship (Figure 5A,B), and then show that the experimental perturbation predominantly modulates amplitude (Figure 5D). Is it possible to predict an experimental perturbation that would show the other type of modulation? Such a testable prediction would strengthen the connection between model and experiment.

We agree with the reviewer that it would be desirable to test our revised model with new data connecting it with different enhancer-promoter pairs or perturbations of regulatory pathways in the future. At the same time, as also mentioned in our response to Reviewer #2 (point 2), we believe that the strength of our study resides in the careful minimization of regulatory and structural confounding effects with the aim of extracting quantitative information and ensuring that perturbations are meaningful and interpretable. In the absence of a clear understanding in the field of the identity of the molecular players that transmit regulatory information from an enhancer to a promoter and determine their compatibility (see e.g. Martinez-Ara et al., *bioRxiv* 2021, Bergman et al., *bioRxiv* 2021 and references therein), it is unfortunately very difficult to predict *a priori* which perturbations might elicit a change in the rates of intermediate regulatory steps.

In principle, these perturbations could target transcription factors (TFs), coactivators (e.g. Mediator subunits), or complexes involved in promoter operation (e.g. the pre-initiation complex). However, global impairment of trans-acting factors results by definition also in indirect effects as they are not limited to the

locus under study. As a consequence such approaches would make the assignment of cause/effect relationships difficult, at odds with our approach.

Another exciting approach would be to introduce sequence modifications in the enhancer or the promoter impairing or replacing TF binding. Targeted approaches to sequence modification would require precise knowledge of the mechanisms that link TF binding to transmission of regulatory information, and specifically of which DNA binding factor should be targeted to result in a modified effective reaction rate. We hope that the Reviewer agrees that this is far from being well established.

As also discussed in our response to Reviewer #2, we believe that the only viable approach to identify (if not predict) mutations and factors involved in changing the shape of the transcriptional response would be to replace the wild-type SCR sequence used in this study with a very large number of modified enhancer sequences where transcription factor binding sites are systematically swapped, inserted or changed. This would be an extremely interesting endeavour which would quickly impinge on fundamental open questions such as convening on an actual definition of what sequences constitute an active enhancer; which additional properties determine minimal enhancer activity; how to identify motifs that would a priori lead to efficient transcription factor binding; etc. While we are very much committed to developing such a method in the near future, we trust that the Reviewer agrees that this requires a long-term theoretical and experimental commitment, and that the validity and broad significance of our study does not rely on such additional developments. We would nonetheless be happy to reconsider our position, should the Reviewer wish to provide additional suggestions.

Minor points:

Figures 2K and 3C: Is there any possible explanation for the apparent periodicity in the simulated results? Does this match experimental measurement? Or does it indicate an overfitting issue of the model?

The graphs the Reviewer referred to showed the predictions of the model for transcriptional levels as a function of the genomic distance from the promoter. Fluctuations in those graphs reflected fluctuations in contact probabilities that were amplified into larger transcriptional changes through the hypersensitive part of the transcriptional response. We have however removed those panels to accommodate the new interpretation of the data in terms of a variable two-state model (**Figure 2**).

Figure 2G: is it a simulated result or just a diagram?

This was actually a simulated result; it has however been replaced by a new simulated trajectory in **Figure 3G**.

Figures 2I, J: These seem to be the same panel as 2B and it appear to be a bit redundant

These plots are now in **Figure 2E** and **G**, with the addition of a few data points that were obtained in experiments we performed in response to Reviewer #1. Unless the Reviewer feels strongly against it, we would prefer to keep them both because panel G provides a close-up view of the behavior of the best-fitting model across the TAD boundary regions.

Figures 3A and C: are these the same data on different y scales?

We apologize for the confusion. Only Figure 3A with the correct y scale has been maintained in the revised version of the manuscript, as described above, and is now in **Suppl. Fig. 1N**.

Figure 3B: might be interesting to see how well the model fits the experimental, maybe via a gamma/negative binomial dist? This might be relevant to the bursting behavior.

We thank the Reviewer for this suggestion; however as noted above, we have now used smRNA FISH measurements (**Figure 2D**) to better characterize bursting behavior, which we found to be compatible with frequency modulation by the enhancer through a k_{on} that depends sigmoidally on contact probabilities (**Figure 2D-F**).

Line 1-3 of page 7: the transgene seems to be very close to the rescued CTCF motif, confounded by the limited resolution of Hi-C map (claimed to be 6.4kb), the validity of the claim is questionable.

We modified the sentence (now in page 8) as follows: “Strikingly, transcriptional insulation across the CTCF site occurred in the absence of changes in the promoter’s interaction probabilities with the region downstream of the CTCF site, at least in the current experimental setup (tiled capture-C with 6.4 kb binning level)”. We agree with the Reviewer that changes might become detectable using more sophisticated C techniques in the future (e.g. capture approaches to MicroC such as the one proposed in Aljahani et al., biorXiv 2021).

Line 43-45 of page 7, logic was not clear

We thank the Reviewer for pointing this out. We have now modified these sentences (now in page 9) and hope that the message is now clearer.

Reviewer Reports on the First Revision:

Referees' comments:

Referee #1 (Remarks to the Author):

The authors have addressed the issues raised in my original review. The revised manuscript is very strong. The exploration of a two-state model with k_{on} on a function of contact frequency is a great addition. The text explaining the model is now much clearer, and I appreciate the separation of the analysis of the two-state model versus the more complex model.

I would highly recommend publication without any further major changes.

Additional minor suggestions and questions:

1. I appreciate the additional experiments the authors performed to address the question of whether changing the FACS sort gate could affect their measurements — specifically, by mapping the locations of additional insertion events to determine whether there are insertions in the TAD that do not lead to detectable increases in gene expression. I think these data do strengthen the experimental dataset. However, I do not believe that this concern is fully eliminated. The authors have reported $P(\text{mobilized element is in TAD} \mid \text{low expression gate})$, when the quantity that is more relevant is $P(\text{low expression gate} \mid \text{mobilized element is in TAD})$. By my rough estimation, if $>1\%$ of low-expression insertions are located in the TAD, this would affect the estimate of the expression level of constructs at a given distance. Can the authors report this quantity? E.g., Fig S1M shows that $\sim 90\%$ of sequencing reads map to non-mobilized cassettes, and I did not see any text reporting where the remaining 10% map.
2. Wording suggestion: The authors describe two categories of models — the “two-state model” / “apparent two-state model” and a more complex model that is referred to as “the model”. This language is a bit confusing. Can the authors give the more complex model some name (“3-step model”?) and use that consistently throughout the text?
3. Page 8, in considering how a different enhancer might affect the model, the authors consider whether $k_{\text{forward}}/k_{\text{back}}$ or $k_{\text{on_enh}}$ might be affected. Could a change in μ also be a possibility?
4. Page 8 line 15. “enhancer strength modulates the ability of a promoter to turn on”. Could you briefly speculate about how mechanistically different enhancers might change $k_{\text{on_enh}}$?
5. Fig 3F references k_{lo} which I believe is a remnant of the previous version of the model
6. The same graphical motif is reused to describe the two-state model (2F) and a subcomponent of the more complex model (3F, 5B). Consider making these distinct graphics for clarity.
7. Could you add the 3D contact data for the two cell lines directly to Fig 4C?

Referee #2 (Remarks to the Author):

I thank the authors for the additional analyses and explanations in response to my and other reviewers' comments. I believe the already impressive paper has improved even further as a result, particularly owing to the updated modelling section.

I agree that the analysis of Fulco et al data does not have to be included, but it was interesting to see that it did not change the predictions either way. I also note a slight misunderstanding regarding my Point 1b, but on rereading I concur that was entirely my fault in phrasing it unclearly. Either way, it is generally addressed by the new analysis in Figure S3.

One outstanding question is whether the authors' prediction that enhancer-promoter contacts are much faster than the "regulatory steps" at the promoter is biologically plausible. Also, what if this prediction is not valid – could something else be behind the poor fit of the model in Figure 3 to the two-state model without this assumption? Could it be, for example, that the two-state model itself is a simplification? I would invite the authors to discuss these possibilities at the top of p.8.

Finally, the new Figure 3 could do with some minor polishing. For example, in panel 3C I would suggest illustrating more clearly that the rate of transcription in the basal regime is lower. Also, panel 3DE are a little confusing (and it's unclear if the caption above D applies to both D and E).

Overall though, I wish to take this opportunity to congratulate the authors on this exciting and rigorous work.

Referee #3 (Remarks to the Author):

We appreciate the authors extensive revisions and the thoughtful responses to our previous comments. As before, we continue to believe that the experimental data set is interesting and valuable for the whole field of gene regulation. Our previous review tried to clarify the connections between data, model, and interpretation. The revision significantly improved the modeling framework, and critically showed how many aspects of the experimental data could be explained by a phenomenological model in which enhancer-promoter contact frequency non-linearly controls the "on" rate of a two-state model of bursty gene expression. (Here, we will refer to this model, shown in Figure 2, as the high-level model). This conclusion alone is extremely useful, as it drastically simplifies the potential complexity. The manuscript goes beyond this statement to further explain how control of the effective on-rate in the high-level model could arise in a particular regime of a more detailed underlying "low level" model, more similar to that in the original manuscript. While there is no additional direct experimental support for this low level model, the revision uses previous work in the literature to better support some of its key assumptions. (However, see comment below about one citation.)

Overall, we continue to find the paper to be an important contribution to a quantitative understanding of this crucial aspect of mammalian gene regulation. It will undoubtedly stimulate a

lot of further research and open up new questions. In our view, the high level model is already valuable, because it has relatively few parameters, and could be further tested by other researchers in the future, even without a direct molecular interpretation for each parameter. The revision devotes a great deal of space to the second, low-level, model. They also note that it is consistent with previous observations of a lack of correlation between transcription bursts and enhancer-promoter contacts (page 7). However, because it has more parameters, most of which cannot be directly measured or controlled, it was still less clear to us how useful it would be to other researchers. It remains “a model” (albeit with additional support) and not necessarily “the model.” Given the limited conclusions one can draw without additional experimental validation, we would overall advocate condensing it considerably to focus more on the high level model.

Additionally, we found the description of the model in the text to be somewhat hard to follow. We would advise getting feedback from colleagues to help make it as streamlined and readable as possible.

The author points out that their mechanistic model can be reduced to a 2-state model when $k_{close, far} \gg k_{forward, back} \gg k_{onbasal, enh}, k_{off}, \mu, n > 1$ and $k_{forward} > k_{back}$, which is a set of critical qualitative constraints of potentially great biological relevance. However, except that the observed decoupling of E-P proximity and transcriptional activation (Alexander et al. 2019) is consistent with a consequence of the model constraints, no other experimental evidence has been provided to support these theoretical constraints. Moreover, one reference cited by the author (Chen, H. et al. 2018) has experimental data indicating the timescale of E-P interactions to be on the order of minutes. Such a timescale is longer than the indicated timescales of molecular regulatory processes (on the order of seconds to tens of seconds), which is at odds with the constraint that $k_{close, far} \gg k_{forward, back}$. It would be helpful if the authors could comment on the apparent contradiction.

Minor comments regarding the revised manuscript:

In terms of figure presentation, the papers often shows the same data multiple times (e.g, Figure 2B and E contain the same data set; 2F and 3H depict the same model). Perhaps better to just show one panel and explicitly refer to different parts of the panel when discussing respective data in the article.

Fonts can be quite small in many figure panels. For example, figure panels with genome browser tracks are generally not legible.

The insertion site of transgene is so close to the deleted CTCF site, making the claim from line 17-18 on page 8 hard to judge.

Xiao et al. has been cited twice.

Author Rebuttals to First Revision:

We would like to thank the Referees for their appreciation of our manuscript and for a very constructive review process, which we think has substantially improved our study. We have now addressed their remaining comments and suggestions as described in the following point-to-point response. Substantially modified passages are marked in red in the (substantially shortened) main text and figure legends.

In summary, these are the main new changes:

1. We provide a reduced and simplified description of the mechanistic model (Fig. 3), which should also clarify its connection to the simpler phenomenological model of Fig. 2; see for example new text at p. 6.
2. We highlight how the mechanistic model provides “a” possible mechanism rather than “the” mechanism of enhancer-promoter communication, and how future live-cell imaging experiments with higher spatial resolution will allow testing its model predictions, and thus its founding hypotheses (notably through a new sentence in the Discussion paragraph, p. 8).
3. We speculate on which mechanisms could allow enhancers to modulate promoter *on* rates (p. 7).

Point-by-point responses to reviewers’s comments

Referees' comments:

Referee #1 (Remarks to the Author):

The authors have addressed the issues raised in my original review. The revised manuscript is very strong. The exploration of a two-state model with k_{on} a function of contact frequency is a great addition. The text explaining the model is now much clearer, and I appreciate the separation of the analysis of the two-state model versus the more complex model.

I would highly recommend publication without any further major changes.

Additional minor suggestions and questions:

1. I appreciate the additional experiments the authors performed to address the question of whether changing the FACS sort gate could affect their measurements — specifically, by mapping the locations of additional insertion events to determine whether there are insertions in the TAD that do not lead to detectable increases in gene expression. I think these data do strengthen the experimental dataset. However, I do not believe that this concern is fully eliminated. The authors have reported $P(\text{mobilized element is in TAD} \mid \text{low expression gate})$, when the quantity that is more relevant is $P(\text{low expression}$

gate | mobilized element is in TAD). By my rough estimation, if >1% of low-expression insertions are located in the TAD, this would affect the estimate of the expression level of constructs at a given distance. Can the authors report this quantity? E.g., Fig S1M shows that ~90% of sequencing reads map to non-mobilized cassettes, and I did not see any text reporting where the remaining 10% map.

We thank the Referee for their positive assessment of the previous additions and apologize for not including this quantity in the previous version of the manuscript. In the lowest eGFP gates (“low 2”), none of the remaining ~10% of reads could be confidently assigned to any specific genomic insertion except within the transgene itself. These insertions perturb the structure of the transgene and likely lead to loss of fluorescent signal, which explains why they are found in this gate. The same holds for the “low 1” gate, where we additionally identified one insertion outside the TAD on Chr15. These numbers are now presented in the revised legend to **Extended Data Fig. 1o**. As specified in the Methods section, confident insertion sites were defined as those with at least one ITR-genome read on both the 5' and 3' of the piggyBac cassette. Thus our in-depth analysis of insertion sites in bulk populations from low and negative eGFP gates supports the notion that these gating regions do not contain eGFP-negative insertions within the TAD.

2. Wording suggestion: The authors describe two categories of models — the “two-state model” / “apparent two-state model” and a more complex model that is referred to as “the model”. This language is a bit confusing. Can the authors give the more complex model some name (“3-step model”?) and use that consistently throughout the text?

We agree with the Reviewer and now refer to the two-state model of **Fig. 2** as “phenomenological two-state model” and to the more detailed model of **Fig. 3** as “mechanistic model”. We believe these new nomenclature more closely reflects each model’s description level and epistemic value and hope that they clarify which model we refer to in each passage of the text.

3. Page 8, in considering how a different enhancer might affect the model, the authors consider whether $k_{\text{forward}}/k_{\text{back}}$ or $k_{\text{on_enh}}$ might be affected. Could a change in μ also be a possibility?

The Referee is right in that the initiation rate of the promoter could be increased by the enhancer and thus could change with different enhancer strength. However, in the case of the *Sox2* promoter, we know that the effect of the full-length SCR is to increase the *on* rate of the promoter (and not its initiation rate μ) compared to its basal transcription, as shown in **Fig. 2**. Since in the absence of an enhancer there is no transmission of regulatory information, presence of a full-length SCR by definition also modulates $k_{\text{forward}}/k_{\text{back}}$. Thus the most natural assumption is that a truncated version of the SCR (which shares largely similar, although less numerous, transcription factor binding sites with the full-length version) should also affect the promoter’s *on* rate and/or the transmission of regulatory information. We hope that the Referee agrees that modulation of initiation rates, although potentially intriguing, is not justified by our data, at least in the specific context of the SCR and its truncated version. It will be however exciting in the future to determine if other enhancers rather modulate initiation (or regulatory) rates using modified versions of our assay with additional pairs of enhancers and promoters.

4. Page 8 line 15. “enhancer strength modulates the ability of a promoter to turn on”. Could you briefly speculate about how mechanistically different enhancers might change k_{on_enh} ?

The Referee raises an important point. Our analysis indeed reflects the wider state of the field in remaining agnostic of the molecular mechanisms that control promoter bursting itself. As explained in detail in the excellent Coulon et al. and Lammers et al. reviews cited in the manuscript, these mechanisms are incompletely understood and likely involve stochastic molecular reactions at the promoter and might involve (combinations of) chromatin state modifications, TF and PIC recruitment, PolII binding, post-translational modifications licensing, pausing... We therefore added a sentence to the passage indicated by the reviewer (now at page 6) pointing at how interactions with the enhancer might lead to modulation of some of these molecular processes.

5. Fig 3F references k_{lo} which I believe is a remnant of the previous version of the model

Thank you, the rates have now been removed from this panel.

6. The same graphical motif is reused to describe the two-state model (2F) and a subcomponent of the more complex model (3F, 5B). Consider making these distinct graphics for clarity.

To avoid any possible confusion, we have now removed the scheme from the new version of **Fig. 5a** and recolored the two schemes in **Fig. 2** and **Fig. 3** in order to emphasize that although formally they are in fact the same model, they derive from different types of analysis (one phenomenological, in **Fig. 2**, and one mechanistic, in **Fig. 3**).

7. Could you add the 3D contact data for the two cell lines directly to Fig 4C?

We have now added the virtual 4C profiles using the promoter location as a viewpoint to **Fig. 4c** as requested by the Referee.

Referee #2 (Remarks to the Author):

I thank the authors for the additional analyses and explanations in response to my and other reviewers' comments. I believe the already impressive paper has improved even further as a result, particularly owing to the updated modelling section.

I agree that the analysis of Fulco et al data does not have to be included, but it was interesting to see that it did not change the predictions either way. I also note a slight misunderstanding regarding my Point 1b, but on rereading I concur that was entirely my fault in phrasing it unclearly. Either way, it is generally addressed by the new analysis in Figure S3.

One outstanding question is whether the authors' prediction that enhancer-promoter contacts are much faster than the “regulatory steps” at the promoter is biologically plausible. Also, what if this prediction is

not valid – could something else be behind the poor fit of the model in Figure 3 to the two-state model without this assumption? Could it be, for example, that the two-state model itself is a simplification? I would invite the authors to discuss these possibilities at the top of p.8.

We thank the Referee for their appreciation of our revised manuscript and for raising this point, which also relates to Referee #3's comment on the expected duration of enhancer-promoter contacts. Ours is indeed a risky prediction: despite anecdotal measurements of physical distances between regulatory sequences in living cells (Chen et al. 2018 - in *Drosophila* and Alexander et al. 2019 - in mESC) we have no information on the actual duration of their molecular interactions. Short interactions are an essential requisite for our mechanistic model (**Fig. 3**) to reduce to an apparent two-state model and thus reproduce the phenomenological observation that the promoter can be approximated as a two-state model with variable k_{on} (**Fig. 2**). If future experiments contradict this prediction, then the hypotheses of the mechanistic model in **Fig. 3** will have to be discarded as there is no other way it can produce a sigmoidal transcriptional response. We however believe that this is a strength rather than a limitation of our study, as it provides quantitative predictions that can be proved or disproved and not merely qualitative arguments in favor or disfavor. We have hopefully clarified our position in the last paragraph of the manuscript (new text at page 8).

We would also like to take this opportunity to clarify that the phenomenological two-state model with variable on rate described in **Fig. 2** actually provides a very good description of the *Sox2* promoter, considering that it accounts for both the single-cell and population-averaged transcription levels simultaneously and in large numbers of cell lines. The 'apparent' two-state model to which the mechanistic model of **Fig. 3** reduces provides very similar levels of agreement with the experimental data. We hope that the revised text makes this point clearer (see for example page 6, new text in the first paragraph). But the Referee is certainly right in that just like any other models, the two-state model can only provide a simplified and idealized coarse-grained description of promoter behavior and it cannot be excluded that more sophisticated models with marginally better agreements with the data could provide an alternative explanation. This is why in the manuscript we generally privileged expressions such as "can be well approximated as" a two-state model over more assertive versions.

Finally, the new Figure 3 could do with some minor polishing. For example, in panel 3C I would suggest illustrating more clearly that the rate of transcription in the basal regime is lower. Also, panel 3DE are a little confusing (and it's unclear if the caption above D applies to both D and E).

We agree with the Referee with the need to streamline **Fig. 3**. We have now removed the former panels **3d-e** to **Extended Data Fig. 4a-b** and emphasized differences in the arrow sizes corresponding to on rates in the basal and enhanced state in **Fig. 3c**. We hope that these modifications improve the readability of Fig. 3.

Overall though, I wish to take this opportunity to congratulate the authors on this exciting and rigorous work.

Referee #3 (Remarks to the Author):

We appreciate the authors extensive revisions and the thoughtful responses to our previous comments. As before, we continue to believe that the experimental data set is interesting and valuable for the whole field of gene regulation. Our previous review tried to clarify the connections between data, model, and interpretation. The revision significantly improved the modeling framework, and critically showed how many aspects of the experimental data could be explained by a phenomenological model in which enhancer-promoter contact frequency non-linearly controls the “on” rate of a two-state model of bursty gene expression. (Here, we will refer to this model, shown in Figure 2, as the high-level model). This conclusion alone is extremely useful, as it drastically simplifies the potential complexity. The manuscript goes beyond this statement to further explain how control of the effective on-rate in the high-level model could arise in a particular regime of a more detailed underlying “low level” model, more similar to that in the original manuscript. While there is no additional direct experimental support for this low level model, the revision uses previous work in the literature to better support some of its key assumptions. (However, see comment below about one citation.)

Overall, we continue to find the paper to be an important contribution to a quantitative understanding of this crucial aspect of mammalian gene regulation. It will undoubtedly stimulate a lot of further research and open up new questions. In our view, the high level model is already valuable, because it has relatively few parameters, and could be further tested by other researchers in the future, even without a direct molecular interpretation for each parameter. The revision devotes a great deal of space to the second, low-level, model. They also note that it is consistent with previous observations of a lack of correlation between transcription bursts and enhancer-promoter contacts (page 7). However, because it has more parameters, most of which cannot be directly measured or controlled, it was still less clear to us how useful it would be to other researchers. It remains “a model” (albeit with additional support) and not necessarily “the model.” Given the limited conclusions one can draw without additional experimental validation, we would overall advocate condensing it considerably to focus more on the high level model.

We thank the Referee for their insightful comments on the modeling part of our manuscript. As also stated in the previous round of revision, we completely agree with the Referee that the mechanistic model provided in **Fig. 3** is “a model” and not “the model” - yet we disagree on this being a limitation of our study. First, we would like to point out once more that no parameter fitting was required to derive its general predictions. Whenever performed, parameter fitting was done rigorously and confidence intervals were provided for all parameters, as previously requested by the Referee. Second, we strongly believe that the main advantage of using a model to interpret the data is precisely to make quantitative predictions that can be tested and used to prove or disprove the underlying hypotheses. The revised discussion paragraph (page 8) now hopefully better describes how the prediction on enhancer-promoter interaction timescales will allow challenging the hypotheses of our mechanistic model in the near future.

We hope that the Referee agrees that this provides an intellectually honest account of the model's role and epistemic value.

We nevertheless agree with the Referee that the main text related to the mechanistic model could be shortened, which we hope we have achieved with the revised version of the manuscript. We have also reduced **Fig. 3** by moving the illustration in previous panels D and E to **Extended Data Fig. 4**.

Additionally, we found the description of the model in the text to be somewhat hard to follow. We would advise getting feedback from colleagues to help make it as streamlined and readable as possible.

We hope that the new, more compact description of the mechanistic model (p.5-6) is also easier to follow. As the Referee will see in the Acknowledgement section, the manuscript has indeed been read by several colleagues who have provided useful comments including on the modeling section.

The author points out that their mechanistic model can be reduced to a 2-state model when $k_{close, far} \gg k_{forward, back} \gg k_{onbasal, enh}, k_{off}, \mu, n > 1$ and $k_{forward} > k_{back}$, which is a set of critical qualitative constraints of potentially great biological relevance. However, except that the observed decoupling of E-P proximity and transcriptional activation (Alexander et al. 2019) is consistent with a consequence of the model constraints, no other experimental evidence has been provided to support these theoretical constraints. Moreover, one reference cited by the author (Chen, H. et al. 2018) has experimental data indicating the timescale of E-P interactions to be on the order of minutes. Such a timescale is longer than the indicated timescales of molecular regulatory processes (on the order of seconds to tens of seconds), which is at odds with the constraint that $k_{close, far} \gg k_{forward, back}$. It would be helpful if the authors could comment on the apparent contradiction.

The Referee raises an important and very interesting point. As stated above, we believe that the usefulness of our mechanistic model resides in the quantitative and testable prediction it makes, as the new text hopefully clarifies (notably at page 8). It would be certainly reassuring if existing measurements already supported this prediction, but we believe that none of the available studies (including Chen et al 2018) provides a realistic estimate of enhancer-promoter contacts in mammalian cells. First, the measurements by Chen et al were conducted in early *Drosophila* embryos, where chromatin dynamics could be very different from mESC notably due to the very fast cell cycle dynamics and the completely different genome size and nuclear architecture. Second, in Chen et al. stable enhancer-promoter associations were ectopically induced *via* a pair of *homie* insulator/pairing elements. These are known to lead to stable and frequent interactions of genomic locations even when they are located in *trans* (Fujioka et al, PLoS Gen 2016). It is thus not unlikely that physical interactions in this context might not be representative of endogenous (and possibly more unstable) enhancer-promoter contacts. Third, in this and other studies residual chromatic aberrations in multi-color microscopy (~150-200 nm) prevent the detection of smaller distances, and might thus conceal shorter-scale and faster fluctuations of enhancer-promoter distances. This and other limitations of live-cell imaging approaches are extensively reviewed e.g. in Brandão et al., Curr Op. Cell Biol. 2021. The revised discussion paragraph (notably the

new text at page 8) should now hopefully suggest how future tests of our prediction will likely be enabled by microscopy methods with increased spatial (and thus temporal) resolution.

Minor comments regarding the revised manuscript:

In terms of figure presentation, the papers often shows the same data multiple times (e.g, Figure 2B and E contain the same data set; 2F and 3H depict the same model). Perhaps better to just show one panel and explicitly refer to different parts of the panel when discussing respective data in the article.

We understand the Referee's concern but we would prefer to keep these panels separate, unless they strongly argue against it. Although the data points are the same in **Fig. 2b** and **e**, the model curve in **Fig. 2e** can only be introduced *after* showing how the model was derived in **Fig. c-d**. We also agree that the models in **Fig. 2f** and (now) **3f** are identical, but we believe it is important to make the point that the mechanistic model in **Fig. 3** can reduce to a two-state description that is analogous to the one presented in **Fig. 2**, but where the dependence of the (apparent) k_{on} on contact probabilities is linked to the mechanistic model's parameters.

Fonts can be quite small in many figure panels. For example, figure panels with genome browser tracks are generally not legible.

We have now modified all fonts to comply with *Nature* formatting guidelines.

The insertion site of transgene is so close to the deleted CTCF site, making the claim from line 17-18 on page 8 hard to judge.

We added a mention to the fact that no notable effect can be distinguished "at least in the current experimental setup". To support this statement, in **Fig. 4c** we now show virtual 4C profiles from the promoter viewpoint in the region downstream of the CTCF.

Xiao et al. has been cited twice.

Thank you, this has now been corrected.